# Transformer Circuits Can Realize Clustering Algorithms

**Kenneth L. Clarkson**[1]  **Lior Horesh**[1]  **Takuya Ito**[1]  **Charlotte Park**[† 2]  **Parikshit Ram**[1]

## Abstract

Although transformers are most commonly optimized as statistical sequence models, it is unclear to what extent they can implement and learn exact algorithmic computations. Here, we specify a transformer implementation from first principles that executes a fundamental and widely used method for $k$-means clustering: Lloyd's algorithm. We theoretically prove and empirically demonstrate that this implementation of a transformer architecture, which we term the $k$-*means transformer*, exactly implements Lloyd's algorithm for $k$-means clustering using the standard circuit mechanisms of modern transformers: attention block, residual connections, and feed-forward block. In learning experiments, we find that training this base architecture on $k$-means clustering yields a generalizable clustering algorithm that surpasses Lloyd's algorithm in terms of clustering quality. Finally, we demonstrate that interpretable alterations (e.g., inclusion of layer normalizations) to this architecture yields diverse and novel variants of clustering algorithms, including soft $k$-means, spherical $k$-means, trimmed $k$-means. Overall, our results show that transformer circuit mechanisms can instantiate exact algorithmic routines for clustering, while simultaneously providing an effective learnable model.

## 1. Introduction

Clustering is one of the most fundamental and widely studied problems in computer science, dating back to the mid-20th century (Lloyd, 1982; MacQueen, 1967). Classic problems (such as, $k$-means clustering), which aim to group data points into meaningful clusters, have impacted diverse fields ranging from biology and ecology to economics, social studies, and machine learning (Hartigan, 1975; Aggarwal &

Author names in alphabetical order. [†]Work done as an intern at IBM Research in 2024. [1]IBM Research [2]Massachusetts Institute of Technology. Correspondence to: PR <parikshit.ram@ibm.com>.

*Proceedings of the $43^{rd}$ International Conference on Machine Learning*, Seoul, South Korea. PMLR 306, 2026. Copyright 2026 by the author(s).

Reddy, 2014). Despite their widespread utility, clustering problems are computationally challenging, shown to be NP-hard, even when the number of clusters is two (Dasgupta, 2008). The combination of their widespread impact and inherent hardness has driven decades of research into their underlying theory and methodology.

Despite the historical prevalence of clustering algorithms, the rise of deep learning has significantly altered the recent landscape of machine learning research. Deep learning architectures, and the transformer in particular (Vaswani et al., 2017), excel at learning hierarchical features from raw data, reducing the need to explicitly identify clusters from data features. However, exactly how transformer mechanisms, such as attention, normalization, feed-forward layers, residual connections, contribute to its precise algorithmic expressivity remains largely unclear.

In this study, we bridge classically studied discrete clustering algorithms with modern neural architectures (Figure 1), offering a clear and interpretable approach to investigating how transformer circuit mechanisms map onto discrete operations of a widely used classical scheme: Lloyd's algorithm for $k$-means clustering (Lloyd, 1982).

**Contributions.** Specifically, we present the following:

- Given a set of data points and random initial centers, we show that one layer of an encoder-decoder *attention-only* transformer with *Euclidean distance-based attention scores* exactly performs a single iteration of Lloyd's algorithm in its forward pass (Theorem 2.1).
- We show that this base architecture, when trained on a distribution of clustering tasks, outperforms Lloyd's algorithm (by a significant margin) for new clustering tasks.
- We present an alternative transformer architecture that uses both the standard dot-product attention and feed-forward blocks of conventional transformers to exactly execute a single iteration of Lloyd's algorithm in a single forward pass with one layer (Theorem 2.2).
- We show that expressing a clustering algorithm through transformer circuit mechanisms facilitates the design of additional – and in some cases novel – clustering algorithms using transformer components such as token-wise normalization (Ba et al., 2016; Zhang & Sennrich, 2019) or alternate attention activations, like linear or sparse attention (Martins & Astudillo, 2016) (Section 2.5).

**A**

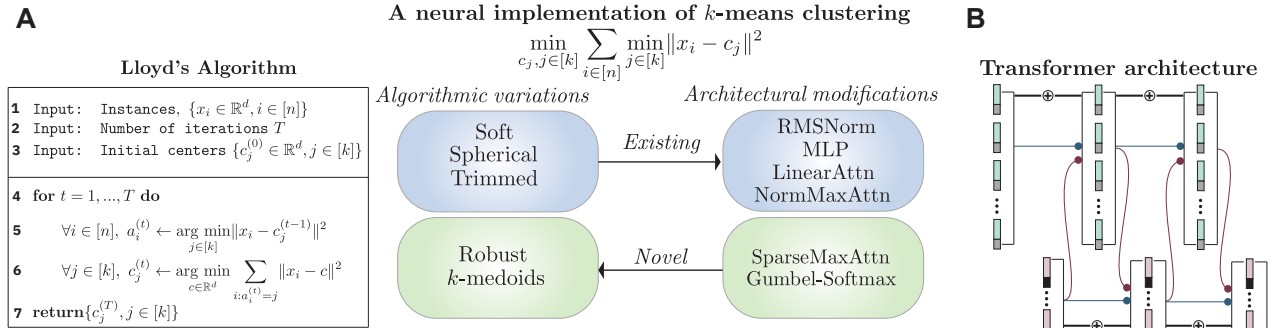

**B**

**Transformer architecture**

*Figure 1.* **Overview.** For the $k$-means clustering problem (**top**; Equation (1)) – a discrete optimization task commonly solved via Lloyd's algorithm (**A**) – we introduce a precise transformer architecture (**B**) that exactly implements Lloyd's iteration using standard components, including self-attention, cross-attention, and residual connections. Specifying precise algorithmic subroutines with transformer circuit mechanisms allows us to (i) implement variations of $k$-means clustering, such as spherical and soft $k$-means (**center left blue box**), by modifying transformer mechanisms (**center right blue box**), and further (ii) devise novel (and interpretable) $k$-means variations (**center left green box**) by incorporating existing transformer mechanisms such as sparse and Gumbel attention (**center right green box**).

## 2. Results

We begin by describing Lloyd's algorithm for $k$-means clustering. Then we introduce the $k$-means transformer, and show that it exactly realizes Lloyd's algorithm both theoretically and empirically, and can be trained to execute a better clustering algorithm than Lloyd's. Finally, we show that variations to the architecture are mathematically equivalent to specific existing and novel clustering algorithms.

**Notation.** We denote the index set as $[m] \triangleq \{1, \ldots, m\}$. Scalars and vectors are denoted by lowercase italics (e.g., $x$, $y$). We will specify when $x$ is a scalar or vector. We assume vectors are column vectors, and their transpose $x^\top$ a row vector. The $i$-th entry in the vector $a$ is denoted by $a^i$. Matrices are denoted as uppercase italics, e.g., $A \in \mathbb{R}^{m \times n}$, with $A^{ij}$ denoting its $(i, j)$-th entry. We use $I_d$ to denote the $(d \times d)$ identity matrix. We specify our attention projection matrices compactly using the following notation: For integers $1 \leq i \leq j \leq d$, $I_d^{i:j}$ denotes a linear operator on vectors $x \in \mathbb{R}^d$ such that $(I_d^{i:j} x)^k = x^k$ for $i \leq k \leq j$, and zero otherwise: $I_d^{i:j} x$ zeroes out the $m$-th entries of $x$ with indices $m \in [1, i) \cup (j, d]$, leaving the rest unchanged. [1] When $j = d$, we use $I_d^{i:}$, and when $i = 1$, we use $I_d^{:j}$.

### 2.1. $k$-means clustering and Lloyd's algorithm

Given a set of points $X = \{x_i \in \mathbb{R}^d, i \in [n]\}$ in a $d$-dimensional Euclidean space, $k$-means clustering involves solving the following discrete optimization problem to find a set of possible centers $C = \{c_1, \ldots, c_k\} \subset \mathbb{R}^d, |C| = k$ by minimizing the following objective:

$$\min_{c_1, c_2, \ldots, c_k \in \mathbb{R}^d} \sum_{i=1}^n \min_{j=1,\ldots,k} \|x_i - c_j\|^2. \qquad (1)$$

This is a well-studied combinatorial optimization problem. Lloyd's algorithm for $k$-means clustering (Figure 1A), often referred to as the "$k$-means clustering algorithm", solves the discrete optimization in Equation (1) (given some initial centers) in an iterative manner (Figure 1A, lines 4-6) by using the conditions mentioned: it alternately (i) assigns the points to clusters for fixed centers (Figure 1A, line 5), and (ii) updates cluster centers given cluster assignments (Figure 1A, line 6) in each iteration. Both these alternating steps minimize the $k$-means objective with respect to one set of optimization variables (assignments or centers) while fixing the other set (centers or assignments respectively). In the following subsection, we present *a transformer that exactly implements Lloyd's algorithm in-context*.

### 2.2. A transformer implementation of Lloyd's algorithm

The original encoder-decoder transformer (Vaswani et al., 2017) takes a sequence of tokens as input and decodes an output sequence of tokens. The main components of a single *transformer block* are (i) the self-attention within the encoder and decoder, (ii) the cross-attention between the two, (iii) residual connections, (iv) per-token normalization (such as LayerNorm (Ba et al., 2016)), and (v) per-token nonlinear transformation via a single hidden-layer feed-forward network (FFN). Here we simplify the original transformer architecture to implement Lloyd's algorithm: the primary ingredients only require residual connections, self-attention, and cross-attention. [2] Given a set of points and initial centers, each layer of our proposed transformer exactly performs a single iteration of Lloyd's algorithm (Figure 1A, lines 5 and 6) in its forward-pass, and $T$ such layers exactly execute Lloyd's algorithm for $T$ iterations; see the schematic of our proposed $k$-means transformer in Figure 2.

---

[1] We can implement $I_d^{i:j}$ as a $(d \times d)$ diagonal matrix with entries $(I_d^{i:j})^{kk} = 1$ for $k \in [i, j]$, and all other entries as zeros.

[2] Token-wise normalization and FFNs are necessary for specific variations of $k$-means clustering, which we detail later.

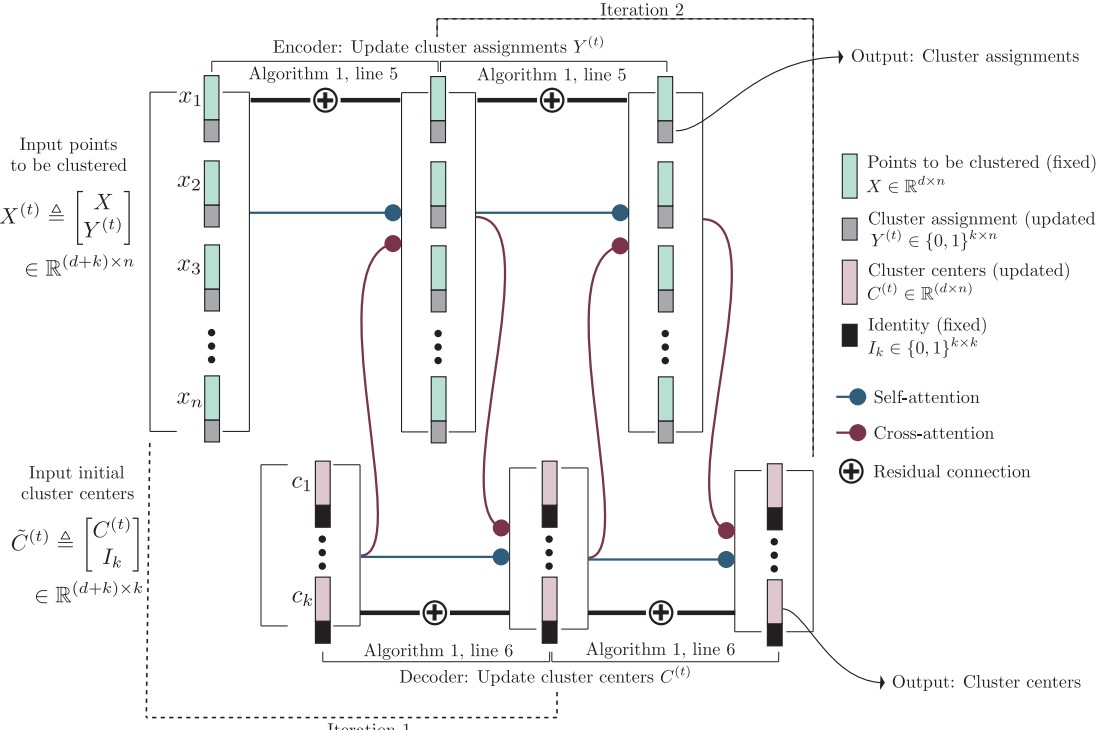

*Figure 2.* **The $k$-means transformer.** We depict the architectural diagram that uses only attention and residual connections. The more general architecture is in Figure 12. The inputs to the transformer are the set of $n$ points in $X \in \mathbb{R}^{d \times n}$ (■), and the initial cluster centers $C^{(0)} \in \mathbb{R}^{d \times k}$ (■) tokenized as $X^{(0)}$ (■) and $\tilde{C}^{(0)}$ (■) respectively; initial cluster assignments $Y^{(0)} \in \{0, 1\}^{k \times n}$ are set to zero. Each layer consists of (i) residual connections (⊕), (ii) encoder-self (–•) and cross-attention (–•), followed by (iii) a decoder self- and cross-attention, and realizes a single iteration of Lloyd's algorithm (see Fig. 1A); we depict two such iterations/layers here. The encoder update step (Equation (4), top) replaces the old cluster assignments $Y^{(t)}$ (■) in the encoder tokens $X^{(t)}$ with the new cluster assignments $Y^{(t+1)}$ to get the next layer encoder tokens $X^{(t+1)} \leftarrow \begin{bmatrix} X \\ Y^{(t+1)} \end{bmatrix}$. The decoder update step (Equation (4), bottom) replaces the old cluster centers $C^{(t)}$ in the token embeddings $\tilde{C}^{(t)}$ with the new cluster centers $C^{(t+1)}$ to get the next layer decoder tokens $\tilde{C}^{(t+1)} \leftarrow \begin{bmatrix} C^{(t+1)} \\ I_k \end{bmatrix}$.

In the $k$-means transformer, the set $X$ of points $x_i$ are processed in the encoder, while the cluster centers $c_1, \ldots, c_k$ are iteratively updated in the decoder. While the standard transformer makes use of dot-product based attention scores (that is, $\langle Qx, Kx' \rangle$ for query vector $x$ and key vector $x'$ with corresponding projection matrices $Q, K$) (Vaswani et al., 2017), below we consider negative squared Euclidean distance based attention scores (that is, $-\|Qx - Kx'\|^2$) (Tsai et al., 2019). One critical step in the proposed methodology is the use of a limiting version of the standard soft-max attention, where the soft-max temperature approaches zero (see Appendix D). This limiting soft-max attention is different from the arg-max or hard-max attention, and has been studied as *averaging hard attention* (Strobl et al., 2024).

Below, we show how to implement Lloyd's algorithm with a precise choice of parameters. For this, we define specific embeddings of points and cluster centers, and make use of the attention mechanism and residual connections. To this end, we present three critical components: (i) the process of embedding the input (points to be clustered and the initial cluster centers), (ii) the form of the selective attention

mechanism necessary for clustering, and (iii) the specific encoder-decoder architecture and model parameters with self/cross-attention and residual connections.

**Input embeddings.** We embed each point $x_i \in \mathbb{R}^d$ into a $(d + k)$-dimensional vector $[x_i^\top, y_i^\top]^\top$ for a $y_i \in \{0, 1\}^k$, where the last $k$ dimensions are placeholders for the (evolving) one-hot cluster assignment of each point. Initially, we set $y_i = 0$ for all $i \in [n]$. Arranging these $n$ embeddings as $(d + k)$-dimensional columns gives us the initial encoder embeddings $X^{(0)} \in \mathbb{R}^{(d+k) \times n}$, which we can also write as $\begin{bmatrix} X \\ Y^{(0)} \end{bmatrix}$, where $X \in \mathbb{R}^{d \times n}$ is the input set of points arranged as a matrix with $x_i \in X$ as the $i$-th column in $X$, and $Y^{(0)} \in \{0, 1\}^{k \times n}$ is the matrix of cluster assignments (as one-hot encodings). We also embed the initial centers $c_j \in \mathbb{R}^d$ into a $(d+k)$-dimensional vector $[c_j^\top, e_j^\top]^\top$, where $e_j \in \{0, 1\}^k$ is the $j$-th characteristic vector (a one-hot vector with the $j$-th entry equal to 1), and the last $k$ dimensions denote the index of the cluster. The first $d$ dimensions are placeholders for the (evolving) cluster centers. Stacking the per-center embeddings gives us the initial de-

coder embeddings $\tilde{C}^{(0)} \in \mathbb{R}^{(d+k)\times k}$, which we can write as $\tilde{C}^{(0)} \triangleq \begin{bmatrix} C^{(0)} \\ I_k \end{bmatrix}$. These embeddings evolve through the $k$-means transformer such that the embedding matrices input to the $(t+1)$-th layer are (also see Figure 2 left):

$$\tilde{C}^{(t)} \triangleq \begin{bmatrix} C^{(t)} \\ I_k \end{bmatrix}, \quad X^{(t)} \triangleq \begin{bmatrix} X \\ Y^{(t)} \end{bmatrix}, \quad (2)$$

where $Y^{(t)ji} = 1$ if $x_i$ is assigned to the $j^{\text{th}}$ cluster and 0 otherwise, and $C^{(t)}$ are the cluster centers computed after $t$ Lloyd's iterations (Figure 1A, lines 5-6). Thus the embedding size $d_{\text{emb}}$ for this transformer is $d_{\text{emb}} = d + k$.

**Attention mechanism for clustering.** The attention operation is applied with $n$ queries $X \in \mathbb{R}^{d_{\text{emb}} \times n}$ to $m$ keys $Z \in \mathbb{R}^{d_{\text{emb}} \times m}$, using (learnable) attention projection matrices $Q, K, V \in \mathbb{R}^{d_{\text{emb}} \times d_{\text{emb}}}$, by computing

$$\mathcal{A}^\gamma_{\langle,\rangle}(X, Z; Q, K, V) \triangleq VZ\sigma_\gamma(A) \in \mathbb{R}^{d_{\text{emb}} \times n}, \quad (3)$$

where $A = (KZ)^\top(QX) \in \mathbb{R}^{m \times n}$ is the usual pre-softmax attention score matrix, and $\sigma_\gamma$ is the softmax function applied column-wise on $A$. Note that in standard transformers, the $(j, i)$-th entry of the attention score matrix has $A^{ji} = \langle Kz_j, Qx_i \rangle$. However, for Euclidean $k$-means, we define the attention score as $A^{ji} = -\|Kz_j - Qx_i\|^2$, which we denote as $\mathcal{A}^\gamma_2(\cdots)$. We denote standard dot product attention as $\mathcal{A}^\gamma_{\langle,\rangle}(\cdots)$.

As $\gamma \to \infty$, the vector $\sigma_\gamma(a)$ becomes concentrated on the indices of the largest coordinates of $a$. We define $\sigma_\infty(a) \triangleq \lim_{\gamma \to \infty} \sigma_\gamma(a)$, the limiting soft-max. When $\gamma = \infty$, we omit it, and have $\mathcal{A}_{\langle,\rangle}(\cdots)$ and $\mathcal{A}_2(\cdots)$. Further details on selective attention for clustering can be found in Appendix D. While we are explicitly discussing the inverse temperature $\gamma$ here, it is easy to see that this $\gamma$ can be easily absorbed into the $Q, K$ projection matrices, for example, by setting $Q \leftarrow \sqrt{\gamma}Q$ and $K \leftarrow \sqrt{\gamma}K$.

**Architectural connections and parameters.** Consider the $(t+1)$-th transformer layer, with $X^{(t)}$ and $\tilde{C}^{(t)}$ denoting the input to the $(t+1)$-th encoder and decoder layer respectively (see Equation (2)). To update the cluster assignments and the cluster centers as in Lloyd's algorithm, that is, to obtain $X^{(t+1)}$ and $\tilde{C}^{(t+1)}$ from $X^{(t)}$ and $\tilde{C}^{(t)}$, we compute

$$\begin{aligned} X^{(t+1)} &\leftarrow X^{(t)} + \mathcal{A}_2(X^{(t)}, \tilde{C}^{(t)}, Q_1, K_1, V_1) \\ &\quad + \mathcal{A}_2(X^{(t)}, X^{(t)}, Q_2, K_2, V_2) \\ \tilde{C}^{(t+1)} &\leftarrow \tilde{C}^{(t)} + \mathcal{A}_{\langle,\rangle}(\tilde{C}^{(t)}, X^{(t+1)}, Q_3, K_3, V_3) \\ &\quad + \mathcal{A}_{\langle,\rangle}(\tilde{C}^{(t)}, \tilde{C}^{(t)}, Q_4, K_4, V_4), \end{aligned} \quad (4)$$

where, for both the encoder and decoder tokens, the first terms on the right-hand side are the residual connections,

the second terms are the cross-attention, and the last terms are the self-attention (Figure 2). Note that only the encoder self- and cross-attention updates make use of the negative squared Euclidean distance based attention scores $\mathcal{A}_2(\cdots)$ [3]; the decoder attention updates utilize the standard dot-product attention scores $\mathcal{A}_{\langle,\rangle}(\cdots)$.

Here, the $\{(Q_\mu, K_\mu, V_\mu), \mu = 1, \dots, 4\}$ correspond to the parameters of the transformer (i.e., the queries, keys, and values projection matrices), and *are shared across all layers* $t \in [T]$. We set them as follows:

$$\begin{aligned} K_1 = Q_1 = K_2 = Q_2 = I^{:d}_{d_{\text{emb}}}, \ V_1 = -V_2 = I^{d+1:}_{d_{\text{emb}}}, \\ Q_3 = K_3 = Q_4 = K_4 = I^{d+1:}_{d_{\text{emb}}}, \ V_3 = -V_4 = I^{:d}_{d_{\text{emb}}}. \end{aligned} \quad (5)$$

The input embeddings, attention mechanism and transformer architecture and parameters are set such that:

1. The first $d$ dimensions of the encoder tokens $X^{(t)}$ remain unchanged, while the last $k$ dimensions are updated after a forward pass to reflect new cluster assignments. These cluster assignments are represented as one-hot vectors.
2. The last $k$ dimensions of the decoder tokens $\tilde{C}^{(t)}$ stay fixed, while the first $d$ dimensions are updated after a forward pass through the transformer to reflect new centers.
3. The encoder cross-attention computes the update to the last $k$ dimensions of $X^{(t)}$, while the decoder cross-attention computes the update to the first $d$ dimensions of $\tilde{C}^{(t)}$ (see Appendix D for further details).
4. The encoder (decoder) self-attention exactly computes the additive inverse of the current last $k$ dimensions (first $d$ dimensions) of $X^{(t)}$ tokens ($\tilde{C}^{(t)}$ tokens), and when combined with the residual connection, resets the current cluster assignments (cluster centers) to zero. This allows the cross-attention to update the encoder (decoder) tokens as per Lloyd's algorithm.

More precisely, we prove the following:

**Theorem 2.1.** *Given as input a set of points* $X = \{x_1, \dots, x_n\}$ *and initial centers* $C = \{c_1^{(0)}, \dots, c_k^{(0)}\}$, *arranged in columns of matrices* $X \in \mathbb{R}^{d \times n}$ *and* $C \in \mathbb{R}^{d \times k}$, *and a $T$-layer transformer parameterized with shared* $\{(Q_\mu, K_\mu, V_\mu), \mu = 1, \dots, 4\}$ *as specified in Equation (5), utilizing the updates in Equation (4), the output of the $T^{th}$ transformer layer exactly matches the output of Lloyd's algorithm (Figure 1A) after $T$ iterations.*

The proof uses the values of $\{(Q_\mu, K_\mu, V_\mu), \mu = 1, \dots, 4\}$ specified in Equation (5), and the properties of limiting soft-max attention to show the equivalence between the transformer update and execution of the Lloyd's algorithm. We provide further technical details and the proof in Appendix E. This $k$-means transformer is able to exactly im-

---

[3]We will show how the encoder updates can be obtained with dot-product attention $\mathcal{A}_{\langle,\rangle}(\cdots)$ in Section 2.4.

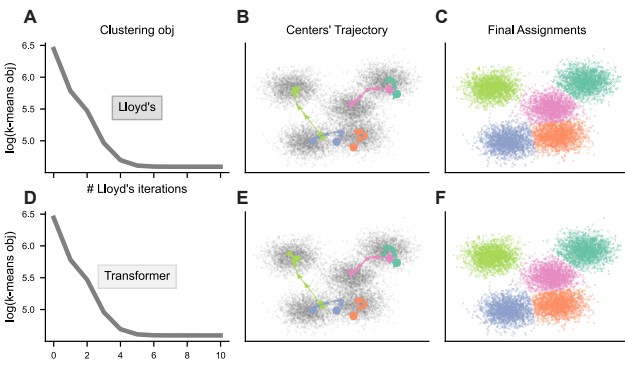

*Figure 3.* **Numerical validation.** We compare the $k$-means transformer to Lloyd's algorithm in $d = 2$ dimensions with $n = 10000$ points and $k = 5$ clusters using $\gamma = 10^4$. We present: **(A-C)** The objective function, evolution of cluster centers, and final assignments after 10 iterations of Lloyd's algorithm. **(D-F)** Same as **(A-C)**, but after forward-pass through 10 layers of the $k$-means transformer. (In panels **B** and **E**, initial cluster centers are the circles ● while the evolving cluster centers are shown as ▲; see a larger version in Figure 8 in Appendix A.)

plement Lloyd's algorithm with a practically-sized model:

**Corollary 2.1.** *The size of this $k$-means transformer architecture implementing Lloyd's algorithm is $O(d + k)$.*

Thus, for a $k$-clustering problem with $n$ points in $d$ dimensions, the size of the transformer is only additive in $d$ and $k$ — that is $O(d+k)$ — which is smaller than the $O(dk)$ size of the output cluster centers and smaller than the $O(d(n + k))$ size of the input consisting of $n$ points and $k$ initial centers.

The transformer implementation of Lloyd's algorithm assumes the theoretical limiting soft-max attention. We therefore study conditions under which exact numerical correspondences can be obtained between the $k$-means transformer and Lloyd's algorithms for a finite $\gamma$. We first illustrate this using points $x_i \in \mathbb{R}^2$ to aid in visualization, and $k = 5$ clusters. Specifically, for very large, yet finite $\gamma = 10^4$, we find that each layer of the $k$-means transformer exactly matches Lloyd's algorithm across 10 iterations (i.e., 10 transformer layers) (Figure 3A,D). In Figure 3B,E we demonstrate that the trajectories of the cluster centers evolve identically across Lloyd's algorithm and the $k$-means transformer, resulting in identical final cluster assignments (Figure 3C,F). We perform a thorough analysis of parity between the transformer and Lloyd's algorithm with varying problem parameters such as number of points, dimensions, number of clusters, and data variance in Appendix A.

### 2.3. Training the transformer to cluster

The previous result presents a transformer architecture with precise weights that exactly realizes Lloyd's algorithm. How would this transformer architecture behave when its weights are learned by optimizing a clustering objective? Here, given a set of clustering tasks, we learn the transformer pa-

rameters $\{(Q_\mu, K_\mu, V_\mu \in \mathbb{R}^{d_{emb} \times d_{emb}}), \mu = 1, \ldots 4\}$ used in Equation (4). We consider clustering tasks consisting of $n$ points $X \in \mathbb{R}^{d \times n}$ sampled from a $d$-dimensional mixture of Gaussians. Each task is generated from a different mixture of Gaussians. We randomly select $k$ of the $n$ points as the initial centers $C^{(0)} \in \mathbb{R}^{d \times k}$. We train a one-transformer layer with randomly initialized weights and $\gamma = 1.0$ by minimizing the (smoothed upperbound of the) $k$-means objective in Equation (1), averaged across all tasks. See Appendix B for further experimental details and results.

We train this one-layer transformer model (for single-step clustering) until convergence, and then utilize it repeatedly to perform multi-step clustering. We evaluate this trained model on 320 new clustering tasks (from the same task distribution), and compare it to Lloyd's algorithm for the same number of clustering steps. Figure 4a shows that the *trained transformer significantly outperforms the $k$-means objective obtained by Lloyd's algorithm*, converging to an average log-$k$-means-objective of $\sim$5 compared to 5.25 obtained by Lloyd's. This highlights the ability of our proposed architecture to learn highly effective clustering algorithms even when using standard soft-max attention with $\gamma = 1$ (as opposed to limiting soft-max attention with $\gamma \to \infty$).

To evaluate whether the transformer learns meaningful clustering mechanisms, we apply this learned transformer to clustering problems of varying sizes. We use the transformer trained on $d = 32$-dimensional clustering tasks with $n = 512$ samples and $k = 10$ clusters. In Figure 4d, we evaluate this trained transformer on unseen clustering tasks, varying the number of samples ranging from 128 (smaller than the training tasks) to 2048 ($4\times$ the size of the training tasks). We find that this trained transformer still outperforms Lloyd's algorithm.

In our proposed model (where we chose the weights), we use $d_{emb} = (d + k)$-dimensional token embeddings to encode points with cluster assignments, and cluster centers with cluster indices. To understand the importance of the choice of $d_{emb} = d + k$ to clustering performance, we consider an alternative transformer model with $d_{emb} = d$ dimensional token embeddings (dropping the additional $k$ dimensions). After training the model on the same set of clustering tasks, we find that the model with reduced $d_{emb}$ is unable to match the performance of Lloyd's algorithm, converging to an average log-$k$-means-objective of $\sim$5.35 compared to 5.25 achieved by Lloyd's. This indicates that the additional embedding dimensions are key to effective clustering performance (Figure 4b).

Lloyd's algorithm is known to be quite sensitive to the initial centers, and various initialization algorithms have been developed such as the farthest-first traversal scheme (Gonzalez, 1985) and $k$-means++ (Arthur & Vassilvitskii, 2007). In Figure 4c, we evaluate the ability of the trained transformer

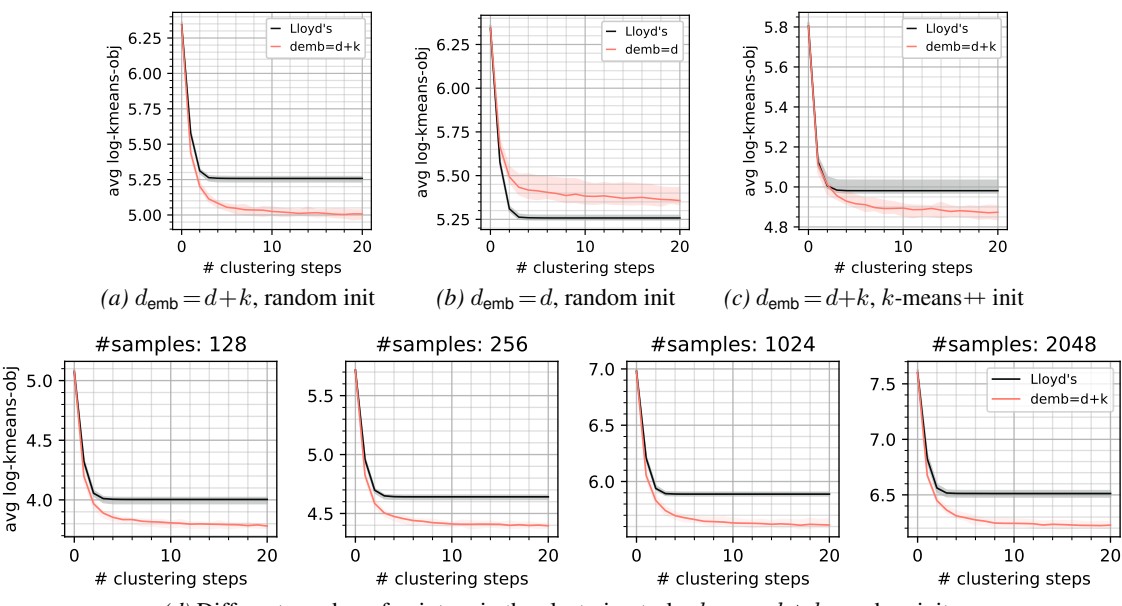

*(a)* $d_{\text{emb}} = d+k$, random init     *(b)* $d_{\text{emb}} = d$, random init     *(c)* $d_{\text{emb}} = d+k$, $k$-means++ init

*(d)* Different number of points $n$ in the clustering task. $d_{\text{emb}} = d + k$, random init

*Figure 4.* **Clustering effectiveness of the trained $k$-means transformer.** We evaluate the trained $k$-means transformer against Lloyd's algorithm for 320 unseen clustering tasks with $d = 32$ dimensions and $k = 10$ clusters for 20 clustering steps, starting from the same initial cluster centers. The lines represent the average log of the $k$-means objective in Equation (1) (*lower is better*), while the ribbons indicate their min-max range over the 320 clustering tasks. Beyond the base comparison in Figure 4a with $n = 512$ points, $d_{\text{emb}} = d + k$, and randomly initialized centers, we consider additional settings such as (i) a different $d_{\text{emb}} = d$ in Figure 4b, (ii) $k$-means++ center initialization in Figure 4c, and (iii) varying number of points $n$ in the clustering task in Figure 4d.

to leverage improved initialization via $k$-means++. We see that the performance of Lloyd's algorithm improves to an average log-$k$-means-objective of less than 5.0 (compared to 5.25 in Figure 4a on the same set of clustering tasks). However, the trained transformer still continues to outperform Lloyd's, achieving an average log-$k$-means-objective of ∼4.85 (improving from ∼5.0 in Figure 4a on the same set of tasks). Note that this transformer was only trained with tasks using random initial centers, but is able to utilize improved initial centers during test time for unseen tasks.

To better understand what the model is learning, we present the learned weights in Figure 5, and make the following observations: (i) The learned $Q_\mu^\top K_\mu$ matrices match the dominant diagonal form of the theoretical constructions in 3/4 cases even though the learned $Q_\mu, K_\mu$ matrices do not look similar to the constructed ones. This aligns with our Remark E.1 in Appendix E. (ii) The learned $Q_3^\top K_3$ for the cross-attention in the cluster center update appears significantly different (more diffuse) than the constructed $I_{d+k}^{d+1:}$ in Theorem 2.1, implying that the learned model updates the centers differently than Lloyd's. (iii) The learned value matrices $(V_3, V_4)$ in the center update (cross- and self- attention, respectively) align with the constructed ones. However, the value matrices for the assignment updates $(V_1, V_2)$ appear to differ from the constructions, especially with the self-attention $V_2$ having a dominant diagonal form (close to $I_{d+k}^{:d}$) that is orthogonal to the constructed $-I_{d+k}^{d+1:}$.

To better compare the behaviour of the learned $k$-means transformer to Lloyd's we present the attention maps for each of the attention modules for 4 of the clustering tasks in Figure 6, and make the following observations: (i) All the learned attention maps are quite sparse even with a softmax attention with $\gamma = 1$, implying that the transformer learned to utilize sparse attention maps (as in Theorem 2.1) without explicitly requiring the limiting softmax attention. (ii) The learned self-attention maps for both the updates in Equation (4) seem to match the constructed one quite well with a diagonal attention map. (iii) The learned and constructed cross-attention maps differ significantly. The learned assignment update attention maps $\mathcal{A}_2^\gamma(X^{(t)}, \tilde{C}^{(t)}, Q_1, K_1)$ visually appear to better match the constructed maps, while the learned center update attention map $\mathcal{A}_2^\gamma(\tilde{C}^{(t)}, X^{(t+1)}, Q_3, K_3)$ differs significantly from the constructed one.

### 2.4. Clustering with dot-product attention and FFN

The proposed transformer in Section 2.2 uses (negative) squared Euclidean distances to exactly emulate a single Lloyd's iteration for Euclidean $k$-means. However, standard transformers architectures use dot-product attention. Here, we will show that transformers with dot-product attention can also exactly execute a single Lloyd's iteration with (i) a modified token embedding scheme, and (ii) a FFN block following the attention block, as is standard in typical trans-

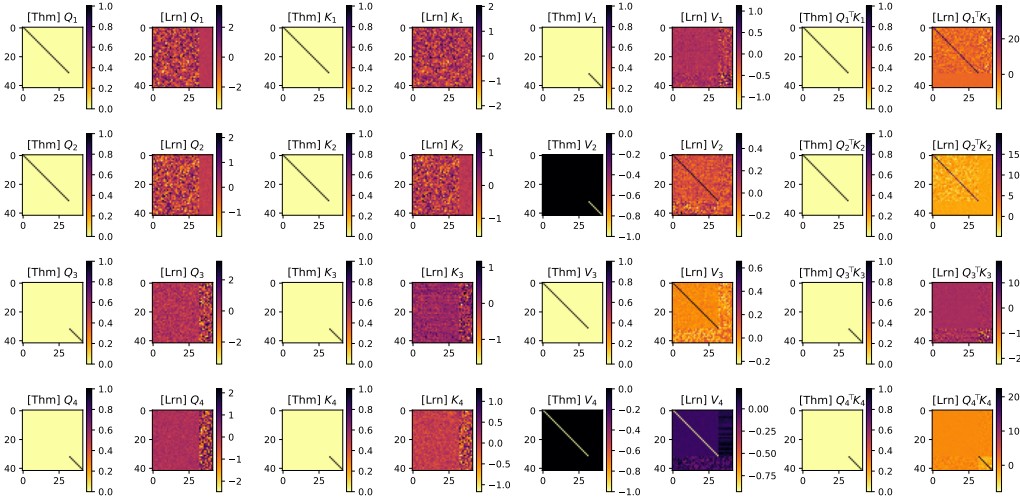

*Figure 5.* **Constructed [Thm] and learned [Lrn] weights in the $k$-means transformer.** We examine the model with $d = 32, k = 10$ using the 4 attention modules in Equation (4), and view the 4 sets of weights $\{(Q_\mu, K_\mu, V_\mu, Q_\mu^\top K_\mu), \mu \in [4]\}$, comparing the learned ones with the constructions in Theorem 2.1. The embedding dimensionality $d_{\mathsf{emb}} = d + k = 42$; all weight matrices are $(42 \times 42)$.

formers. One difference from a standard block is that we will make use of a FFN with rectified quadratic activation instead of the usual rectified linear or ReLU activation.

For any matrix $A \in \mathbb{R}^{m \times p}$, let $q(A) \in \mathbb{R}^p$ be the column-wise squared $\ell_2$ norm, with the $i$-th value in $q(A)$ being the squared $\ell_2$ norm of the $i$-th column of $A$. Given this definition, we modify the $(d + k)$-dimensional embeddings from Equation (2) to $d_{\mathsf{emb}} = (d + k + 2)$-dimensional ones as follows:

$$\tilde{C}^{(t)} \triangleq \begin{bmatrix} C^{(t)} \\ q(C^{(t)})^\top \\ -1/2 \\ I_k \end{bmatrix}, \quad X^{(t)} \triangleq \begin{bmatrix} X \\ q(X)^\top \\ -1/2 \\ Y^{(t)} \end{bmatrix}. \quad (6)$$

Then we use the following dot-product attention updates:

$$
\begin{aligned}
X^{(t+1)} &\leftarrow X^{(t)} + \mathcal{A}_{\langle,\rangle}(X^{(t)}, \tilde{C}^{(t)}, Q_1, K_1, V_1) \\
&\quad + \mathcal{A}_{\langle,\rangle}(X^{(t)}, X^{(t)}, Q_2, K_2, V_2) \\
\tilde{C}^{(t+1/2)} &\leftarrow \tilde{C}^{(t)} + \mathcal{A}_{\langle,\rangle}(\tilde{C}^{(t)}, X^{(t+1)}, Q_3, K_3, V_3) \quad (7) \\
&\quad + \mathcal{A}_{\langle,\rangle}(\tilde{C}^{(t)}, \tilde{C}^{(t)}, Q_4, K_4, V_4) \\
\tilde{C}^{(t+1)} &\leftarrow \tilde{C}^{(t+1/2)} + W_2^\top \sigma_{\mathsf{F}}(W_1 \tilde{C}^{(t+1/2)}),
\end{aligned}
$$

where $\{(Q_\mu, K_\mu, W_\mu), \mu = 1, \dots, 4\}$ are attention parameters, and $W_1, W_2 \in \mathbb{R}^{d \times (d+k+2)}$ are the FFN parameters, shared across all layers $t \in [T]$, with $\sigma_{\mathsf{F}}(z) = [z]_+^2$ as the rectified quadratic activation. Then, we can show that:

**Theorem 2.2** (informal). *Given as input a set of points $X$ and initial centers $C$, there exists a $T$ layer transformer with shared attention parameters $\{(Q_\mu, K_\mu, V_\mu), \mu = 1, \dots, 4\}$ and FFN parameters $(W_1, W_2)$ using the updates in Equation* (7) *such that the output of the $T^{th}$ transformer layer exactly matches the output of Lloyd's algorithm with the same input after $T$ iterations.*

The complete theorem statement and proof are presented in Appendix F. It follows the same technique as in Theorem 2.1 with the following modifications: (i) The $Q_1, K_1, Q_2, K_2$ matrices are set such that the dot-product between the projected query-key tokens computes exactly the negative squared Euclidean distance making use of the squared norms present in $q(X)$ and $q(C^{(t)})$ in Equation (6); for example, in the encoder cross-attention, with the $i$-th encoder to-ken as query $q = [x_i^\top, \|x_i\|^2, -1/2, y_i^\top]^\top$ and $j$-th decoder token as key $k = [c_j^\top, \|c_j\|^2, -1/2, e_j^\top]^\top$, the projection matrices $Q_1, K_1$ are set such that the projected forms are $Q_1 q = [x_i^\top, \|x_i\|^2, -1/2]^\top$ and $K_1 k = [c_j^\top, -1/2, \|c_j\|^2]^\top$, and thus $\langle Q_1 q, K_1 k \rangle = -1/2 \|x_i - c_j\|^2$. (ii) The decoder attention updates compute the cluster centers $c_j^{(t+1)}, j \in [k]$ in $\tilde{C}^{(t+1/2)}$ as done in Theorem 2.1. (iii) The decoder FFN computes the squared norm of the new cluster centers $\left\| c_j^{(t+1)} \right\|^2$, and updates the decoder tokens to get $\tilde{C}^{(t+1)}$.

### 2.5. General architecture and other algorithms

While we provably show that our specific transformer architecture with appropriate parameterization exactly executes Lloyd's algorithm for Euclidean $k$-means clustering, we also consider a broader view of our proposed architecture as an algorithmic skeleton for clustering in Figure 12 (Appendix C). This novel view of *clustering algorithms as a transformer architecture* allows us to instantiate diverse set of algorithms by modifying different transformer circuit mechanisms. Table 3 (Appendix C) shows examples of existing clustering algorithms that are reproduced in our architecture by modifying different circuit mechanisms.

**Soft Euclidean $k$-means clustering.** (Appendix E, Theorem E.1) Starting with the base architecture that implements

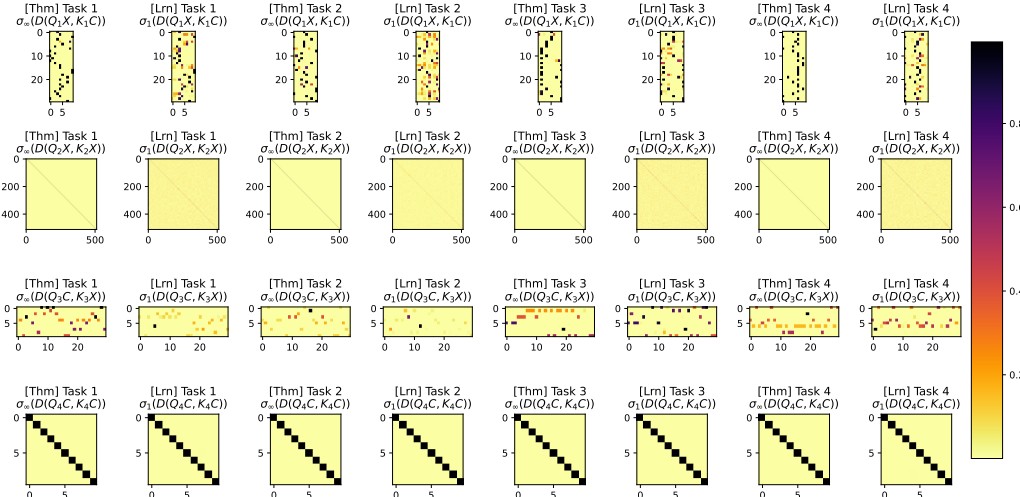

*Figure 6.* **Constructed [Thm] and learned [Lrn] attention maps.** We consider the model with $d = 32$, $k = 10$, and present the 4 sets of attention maps (denoted as $\sigma_\gamma(D(QZ, KZ'))$ with projected queries $QZ$ and projected keys $KZ'$) from the learned model (with $\gamma = 1$) for a single update with 4 different clustering tasks with $n = 512$. We visualize them next to the maps that implement Lloyd's algorithm in Theorem 2.1 (with $\gamma \to \infty$). Note that for the cross-attention maps, the attention maps would be $(512 \times 10)$ or $(10 \times 512)$, making heatmap visualization quite unclear. Thus, we subsample 30 points, and show the cross-attention maps for those 30 points only (leading to more visually clear $(30 \times 10)$ and $(10 \times 30)$ heatmaps).

Lloyd's algorithm (Figure 2, Theorem 2.1), by (i) swapping the limiting soft-max attention with a standard soft-max attention in the encoder cross-attention update, and (ii) changing the limiting soft-max attention with a linear attention in the decoder cross-attention update, we recover the standard soft $k$-means clustering algorithm (Dunn, 1974; Bezdek, 2013) (Algorithm 2 in Appendix E). Note that this is not a discrete clustering problem since each point is allowed to be partially assigned to different clusters (and the partial assignments affect the cluster center updates).

**Discrete spherical $k$-means clustering.** (Appendix G, Theorem G.1) By (i) changing the attention score from negative squared Euclidean distance to dot-product and (ii) including normalization (Ba et al., 2016; Zhang & Sennrich, 2019) in the decoder (such as RMSNorm), we obtain Lloyd's algorithm for spherical clustering (Algorithm 3 in Appendix G), where the points and the centers remain on the unit sphere.

**Trimmed Euclidean $k$-means clustering.** (Appendix H) The original $k$-means problem is known to be extremely sensitive to outliers due to the use of the squared Euclidean distance. To ignore outliers while computing new cluster centers, we replace the soft-max (a negative Shannon entropy regularized arg-max) in the decoder cross-attention with a different regularized arg-max. The *norm $\alpha$-negative entropy* regularized arg-max produces the "norm-max" attention (Blondel et al., 2020), and results in the classical trimmed $k$-means algorithm (Cuesta-Albertos et al., 1997) (Algorithm 4 in Appendix H).

Beyond finding transformer configurations that execute known algorithms in-context, the general view of the trans-former architecture as an algorithmic skeleton allows us to add or substitute architectural components, yielding "novel" clustering algorithms. We present two such algorithms:

**Robust $k$-means clustering.** (Appendix I) While trimmed $k$-means increases the robustness of the $k$-means algorithm by effectively ignoring the outliers based on an estimated distance threshold, we develop an alternate form of robust discrete $k$-means by utilizing a sparse soft-max attention (Gupta et al., 2021; Martins & Astudillo, 2016). Such attention selects points close to the cluster centers, while weighting the contributions of the (selected) points based on their distance to the cluster centers. This dampens the contributions of cluster members that are far away from centers.

**Randomized Euclidean $k$-medoids clustering.** (Appendix I) We consider Euclidean $k$-medoids, defined as:

$$\min_{c_1,\ldots,c_k \in X} \sum_{i=1}^{n} \min_{j=1,\ldots,k} \|x_i - c_j\|^2, \qquad (8)$$

This differs from the $k$-means clustering problem in Equation (1) in that it requires cluster centers $c_j$ to be points in the set $X$. While this problem bears similarities to the $k$-means problem, it is a significantly harder problem, and cannot be solved favorably with Lloyd's alternating algorithm; successful $k$-medoids algorithms include partitioning around medoids (Kaufman, 1990) and CLARANS (Ng & Han, 2002). However, in the $k$-means transformer, utilizing Gumbel-softmax (Jang et al., 2017) attention in the decoder cross-attention allows points from $X$ to be chosen as cluster centers, producing a valid $k$-medoids clustering solution.

Overall, the transformer circuit implementation of the $k$-means algorithm provides a flexible yet interpretable framework to instantiate a diverse variety of clustering algorithms.

## 3. Discussion

The encoder-decoder transformer was developed to solve sequence-to-sequence tasks (Vaswani et al., 2017). One such task is text summarization, where the model takes a large input text and produces a faithful summary. We analogized the $k$-means problem as clustering (summarizing) a large set of points into a small set of cluster centers. This led to *a small, practically-viable encoder-decoder transformer* that exactly realizes Lloyd's algorithm for $k$-means clustering (Theorem 2.1, Theorem 2.2) with a model whose size is smaller than that of the input clustering problem (Corollary 2.1). Importantly, the development of this transformer circuit implementation of $k$-means provides a general approach for developing alternative and novel clustering algorithms through interpretable architectural modifications in the transformer.

Prior theoretical work has shown that infinitely wide neural networks can approximate any function, establishing universal approximation results in an asymptotic sense (Hornik et al., 1989; Hartman et al., 1990). Similarly, certain transformer variants, such as universal transformers or models equipped with recurrent computational steps such as chain-of-thought, have been proven Turing complete, demonstrating that they can simulate arbitrary computation given sufficient depth or time (Strobl et al., 2024; Pérez et al., 2021; Li & Wang, 2025; Jiang et al., 2026; Merrill & Sabharwal, 2024; Dehghani et al., 2019). These results do imply the existence of an implementation of Lloyd's algorithm, but at high cost in time and space; in contrast, our work provides an efficient and compact transformer architecture with an explicit parameterization that *exactly* implements Lloyd's algorithm. Importantly, we show empirically that this hand-specified model can also be learned from data to perform effective clustering.

Our transformer construction that exactly performs Lloyd's algorithm complements emerging work that investigates the in-context learning abilities in transformers across a range of function-learning tasks (Von Oswald et al., 2023; Ahn et al., 2023; Huang et al., 2024; Zhang et al., 2024). In-context learning refers to the ability of transformers to infer a task from example input-output pairs provided in context, and to apply the inferred function to new inputs without any parameter updates (Garg et al., 2022; Akyürek et al., 2023). More specifically, in supervised settings where labeled examples $(x_i, y_i)_{i \in [n]}$ are provided in the context along with a single unlabeled query $x_q$, the model produces a prediction $\hat{y}_q$ by implicitly executing an algorithm in-context during the forward pass. However, most prior research has focused on supervised objectives (e.g., mean squared error), and often rely on simplifying assumptions within the attention mechanism, such as the use of linear attention – a weighted-average approximation of standard softmax attention. These assumptions depart significantly from the original transformer architecture, which relies on selective arg-max-like attention behavior. In contrast, we leverage the full transformer architecture and parameterize it from first principles to exactly implement Lloyd's algorithm in-context, while also exploring the learnability of this construction. And though some other related works have previously shown where input tokens can "self-cluster" in-context (Geshkovski et al., 2023; 2025), we instead specify an architecture that *exactly* reproduces the widely-used Lloyd's algorithm for $k$-means clustering.

Finally, we showed the generality of our proposed architecture by instantiating alternative clustering algorithms via architectural modifications, such as soft $k$-means and trimmed $k$-means, highlighting how transformer circuit mechanisms can be employed for clustering. An interesting implication of our analytic results is that, for any clustering algorithm that we realize as a transformer, there are infinitely many parameter configurations (the $Q, K$ projection matrices), implying the existence of infinitely many instantiations that realize the same algorithm; see Remark E.1 in Appendix E.

**Limitations.** Our theoretical results rely on the use of the limiting soft-max attention with $\gamma \to \infty$, which is practically realizable in the forward-pass, but is hard for backpropagation. However, we show that the $k$-means transformer with $\gamma = 1$ (thus more backprop-friendly) is able to learn an effective clustering algorithm (Figure 4). Furthermore, while we theoretically establish expressivity (ability to realize known algorithms), we leave any corresponding theoretical generalization guarantees for future study. Finally, while our proposed architecture can be trained on clustering tasks with a fixed number of points $n$ and generalizes to problems with larger number of points $n' > n$ (as in Figure 4d), it does not support training on $k$-clustering tasks with a fixed number of clusters $k$ and generalizing to problems with a larger number $k' > k$.

## 4. Conclusion

We demonstrate how the widely studied Lloyd's algorithm for $k$-means clustering can be exactly realized with transformer circuits, and subsequently trained for improved clustering. We also show how interpretable architectural changes lead to various (and sometimes novel) clustering algorithms. Together, these results highlight how algorithmic transparency within transformer circuits can yield both theoretical insight and practical advances. We hope this work serves as a foundation for further exploration of neural implementations of classical methods.

## Impact Statement

This paper presents work whose goal is to advance the field of understanding machine learning. There are many potential societal consequences of our work, none of which we feel must be specifically highlighted here.

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

## A. Numerical Validation for Prescribed Transformer Weights

We next comprehensively assess how different the performances are (i.e., the difference in the log of $k$-means objective between Lloyd's and $k$-means transformer). Relevant code can be found at https://github.com/rithram/kmeans-trf. We test performances across a range of $\gamma$ and data distributional properties, including the dimension $d$ for samples $x \in \mathbb{R}^d$, the total number of samples $n$, the number of clusters $k$, and a constant scaling factor. As expected (and consistent with our theoretical results), we find that increasing $\gamma$ uniformly improves performance of the $k$-means transformer (Figure 7). We also find that as the dimension $d$ of samples increases, the $k$-means transformer behaves more closely to Lloyd's algorithm independent of $\gamma$ (Figure 7A,F,G). This is because the pairwise squared Euclidean distance between points grows with the number of dimensions (assuming the variance is the same/independent within each dimension). Thus, since the soft-max is applied on the attention matrix, which is computed as the pairwise negative squared Euclidean distance between points/tokens, increasing the embedding dimension amplifies the limiting effect in the soft-max without changing $\gamma$.

We also find that as the number of points and clusters increases, divergence between Lloyd's and $k$-means transformer increases (Figure 7B,C). However, this can be ameliorated by increasing $\gamma$, and/or multiplying all samples $x \in X$ by a fixed positive scalar (Figure 7D,E). Multiplying by a constant (positive) factor produces behavior similar to increasing the embedding dimensions of $x \in X$, since scaling all points by a factor $> 1$ increases the pairwise squared Euclidean distance between points, improving the sharpness of the soft-max. Thus, multiplicative scaling of points provides a straightforward (and invertible) approach to improving the performance of the $k$-means transformer; the $k$-means problem is scale invariant.

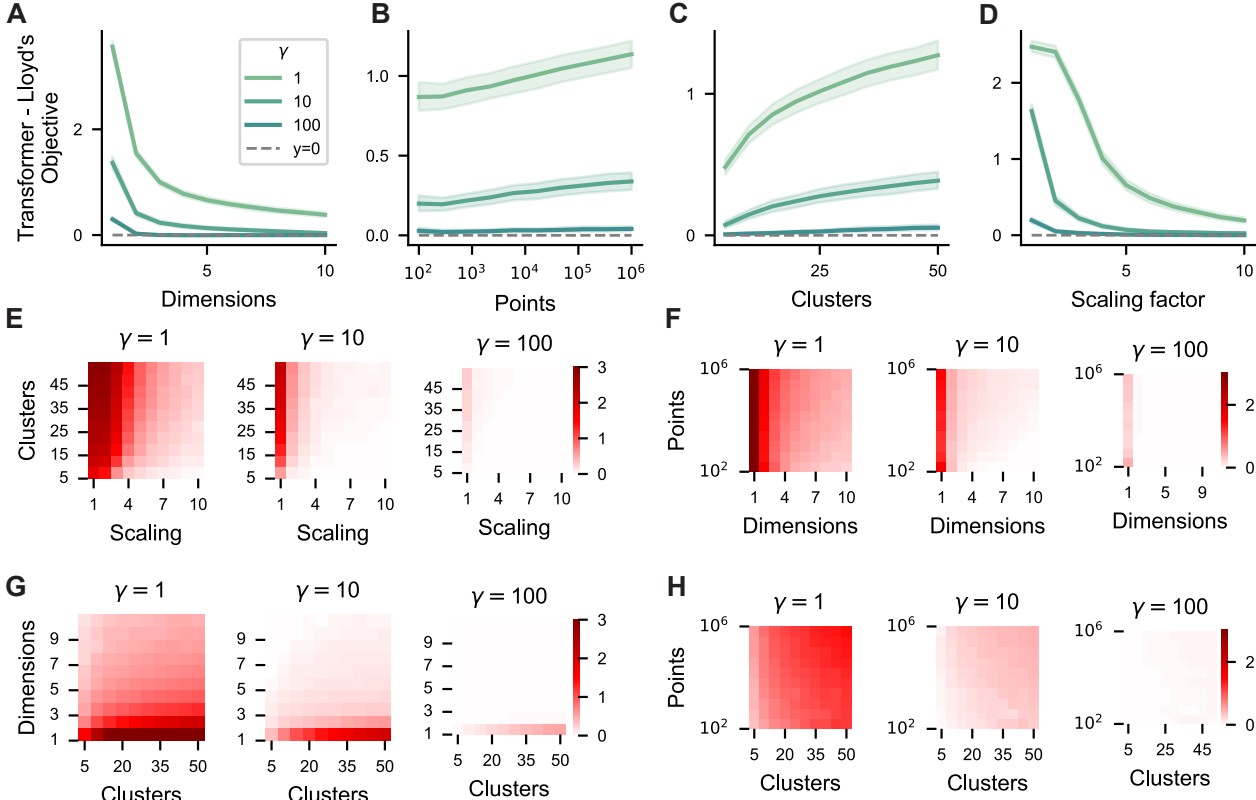

*Figure 7.* Comparing the performance of the $k$-means transformer to Lloyd's algorithm across a range of parameter choices. Consistent with our theory, all transformer models exhibited improved correspondence to Lloyd's algorithm as $\gamma$ increases. **A)** Increasing the dimensionality of the embedding space of points improves correspondence between the transformer and Lloyd's algorithm. Correspondence between models is computed as the difference between the log of transformer and Lloyd's objective function after 10 iterations / layers. Increasing the number of **B)** points-to-be-clustered **C)** number of clusters degrades correspondence. **D)** Scaling the points by a multiplicative constant improves correspondence as the magnitude of the constant increases. This implies that just scaling the features of the data can improve performance without having to change parameters of the architecture. **E-H)** We show the relative changes of jointly altering parameter-pairs in a 2D matrix. Each matrix element reflects the difference of the log $k$-means objective between the transformer and Lloyd's implementation. Error bars (in **A-D**) denote the difference of objective functions across all other parameter ranges, while keeping the parameter of interest fixed.

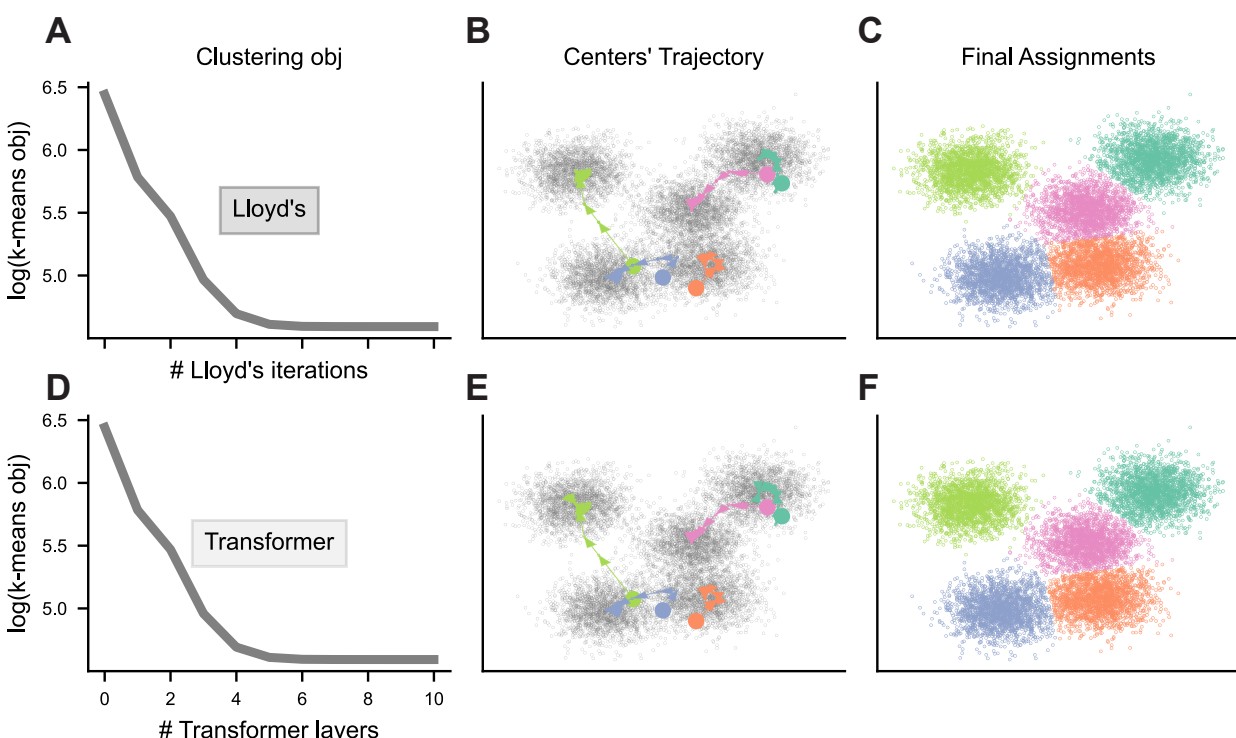

*Figure 8.* **Larger version of Figure 3.** We compare the $k$-means transformer to Lloyd's algorithm in $d = 2$ dimensions with $n = 10000$ points and $k = 5$ clusters using $\gamma = 10^4$. We present: **(A-C)** The objective function, evolution of cluster centers, and final assignments after 10 iterations of Lloyd's algorithm. **(D-F)** Same as **(A-C)**, but after forward-pass through 10 layers of the $k$-means transformer. (In panels **B** and **E**, initial cluster centers are the circles ● while the evolving cluster centers are shown as ▲.)

# B. Training the Transformer

In this section, we discuss the details of training the transformer to learn to cluster. We discuss the training task distributions, the training objective, and finally present results on training convergence and generalization for various choices of the data dimensionality $d$, the number of clusters $k$ and various other training hyperparameters. Relevant code can be found at https://github.com/rithram/kmeans-trf.

## B.1. Distribution of clustering tasks

To train our proposed transformer model, we considered a distribution of clustering tasks, where each task consists of samples generated from a mixture of Gaussians. For the $\tau$-th task, the samples $X_\tau \in \mathbb{R}^{d \times n}$ are generated by:

 (i) randomly sampling the $d$-dimensional centers
 (ii) randomly sampling the $d$ scales of the Gaussians for each of the centers,
(iii) randomly sampling the mixture model weights,
(iv) sampling $n$ samples from this mixture distribution, and
 (v) translate then scaling all data to be in the $[0,1]^d$.

The final scaling step is performed as Euclidean clustering is invariant to scaling and translation. Note that, as the data dimensionality $d$ increases, the diameter of the set of samples increases.

Given the set of samples, we randomly sample $k$ points from $X_\tau$ to define the set of initial centers $C_\tau \in \mathbb{R}^{d \times k}$.

## B.2. Smoothed training objective

Let us denote the weights of the single transformer layers as $W = \{(Q_\mu, K_\mu, V_\mu), \mu = 1, \ldots, 4\}$. Given the $\tau$-th task denoted at $(X_\tau, C_\tau)$, we first embed them as $(X_\tau^{(0)}, \tilde{C}_\tau^{(0)})$, where

$$X_\tau^{(0)} \leftarrow \begin{bmatrix} X_\tau \\ 0_{k \times n} \end{bmatrix} \in \mathbb{R}^{(d+k) \times n}, \quad \tilde{C}_\tau^{(0)} \leftarrow \begin{bmatrix} C_\tau \\ I_k \end{bmatrix} \in \mathbb{R}^{(d+k) \times k}. \tag{9}$$

Then we apply the transformer updates once as defined in Equation (4) to obtain $(X_\tau^{(1)}(W), \tilde{C}_\tau^{(1)}(W))$, where we make the dependence of the transformer output on its weights $W$ explicit.

We can define the $k$-means objective given samples $X$ and centers $C$ as:

$$L(X,C) = \sum_{i=1}^{n} \min_{j \in [k]} \|x_i - c_j\|^2. \tag{10}$$

However, note that this objective is not differentiable, and thus cannot be directly used from training the transformer. Thus, we utilize a smoothed upperbound of $L(X,C)$ using the softmax operation with a temperature $\lambda > 0$, defined as:

$$\tilde{L}_\lambda(X,C) = \sum_{i=1}^{n} \frac{\sum_{j=1}^{k} \exp(-\|x_i - c_j\|^2 / \lambda) \|x_i - c_j\|^2}{\sum_{j=1}^{k} \exp(-\|x_i - c_j\|^2 / \lambda)}, \tag{11}$$

where $L(X,C) = \lim_{\lambda \to 0_+} \tilde{L}_\lambda(X,C)$. Thus, the training objective for learning the transformer weights $W$ is given by:

$$\min_W \sum_\tau \tilde{L}_\lambda(X_\tau, \hat{C}_\tau^{(1)}(W)), \tag{12}$$

where $\hat{C}_\tau(W) \in \mathbb{R}^{d \times k}$ is defined as $\hat{C}_\tau(W) \triangleq \tilde{C}_\tau^{(1)}(W)[d :, :]$ as the first $d$ rows of single transformer layer output $\tilde{C}_\tau^{(1)}(W)$. We consider $\lambda$ as a hyperparameter, and we will demonstrate the effect of this hyperparameter on the training and generalization.

## B.3. Training convergence and effects of hyperparameters

For a given data dimensionality $d$ and number of clusters $k$, we train the model for 10000 steps with an initial learning rate of 0.01 and a decay rate of 0.5 on a loss plateau (where the validation loss does not decrease for consecutively for 250 steps).

For each training step, we sample 32 clustering tasks, and utilize that to compute the smoothed objective in Equation (11). For validation, we utilize 320 separate generated clustering tasks. For all our training and validation clustering tasks, we consider $n = 512$ samples to be clustered. For a given training with $d$ dimensions and $k$ clusters, the embedding dimensions of our transformer $d_{\mathsf{emb}} = d + k$.

As an ablation, we consider a version of our proposed transformer where we drop the additional $k$ dimensions in the token embedding and trained a transformer with $d_{\mathsf{emb}} = d$, using the same training tasks and smoothed clustering objective. All following results contain the result for both our proposed transformer architecture and this ablation with a smaller number of embedding dimensions.

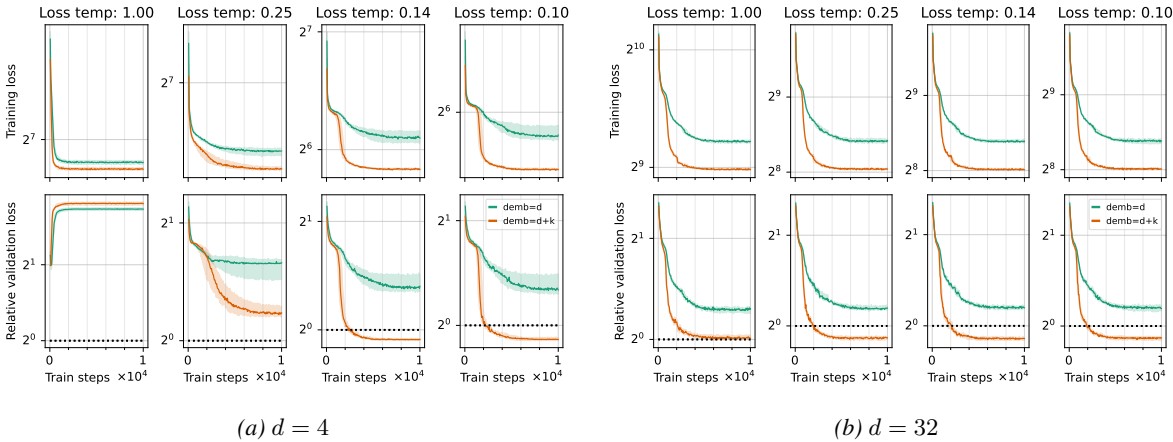

*(a) $d = 4$*  *(b) $d = 32$*

*Figure 9.* Training convergence for varying values of the smoothed loss temperature $\lambda > 0$. Here we select $k = 6$. The top row in each figure tracks the training loss (aggregating the batch smoothed loss $\sum_\tau \tilde{L}_\tau(X_\tau, \hat{C}_\tau(W))$ over the previous 50 training steps) aggregated over 10 random trials with the ribbons indicating the inter-quartile range (*lower is better*). The bottom row in each figure tracks the relative validation loss computed using the validation clustering tasks as $\sum_\tau L(X_\tau, \hat{C}_\tau(W))/L(X, C^{(1)})$, where $C^{(1)}$ is the cluster centers obtained via one step of Lloyd's algorithm as shown in Figure 1A (*lower is better*). The dashed line at $y = 1$ in the bottom row of each figure denotes the performance of a single step of Lloyd's algorithm.

Figure 9 studies the effect of the temperature hyperparameter $\lambda$ for the smoothed training objective in Equation (11) as we vary it from $\lambda = 1$ to $\lambda = 0.1$. With low dimensionality data with $d = 4$ in Figure 9a, we see that training is not able to match the performance of Lloyd's algorithm for $\lambda \in \{1, 0.25\}$, highlighting the mismatch between the desired $k$-means objective and its smoothed counterpart. However, for $\lambda \in \{0.14, 0.1\}$, we see that the training is able to obtain single-layer transformer weights that allow it to outperform single-step Lloyd's algorithm by the significant margin; note that the vertical axis is on a logarithmic scale. For higher dimensions with $d = 32$ in Figure 9b, we see that the training is able to find single-layer transformer weights that match or outperform single-step Lloyd's algorithm for all values of $\lambda$. This is because the pairwise Euclidean distances increase on average as $d$ increases since we assume that the samples to be clustered were translated and scaled to the hypercube $[0, 1]^d$. For the following results, we will fix the smoothed objective temperature $\lambda = 0.1$.

The results in Figure 9 also show that, while the training converges, the relative validation loss of the transformer with $d_{\mathsf{emb}} = d$ indicates that the trained transformer model is unable to match the performance of Lloyd's algorithm, indicating that the additional dimensions in our proposed token embedding is somewhat essential for the model to be able to match Lloyd's algorithm.

Figure 10a demonstrates the effect of changing the number of desired clusters $k$ in the $k$-means problem with $k \in \{6, 10, 16, 25\}$ (for a fixed $d$ and $\lambda$). Note that the embedding dimension $d_{\mathsf{emb}} = d + k$ of our proposed transformer architecture is tied to the number of desired clusters $k$. The results show that, for all values of $k$, with a smoothed loss temperature $\lambda = 0.1$, the learned single-layer transformer significantly outperforms the single-step Lloyd's algorithm. This performance improvement is quite stable, though for the largest value of $k = 25$, there are a couple of trials (as seen with the inter-quartile range) where the training was unable to learn useful weights. This has to do with the fact that, in general, for a fixed temperature $\lambda$, the gap between the exact non-differentiable $k$-means objective $L(X, C)$ in Equation (10) and the smoothed $\tilde{L}_\lambda(X, C)$ in Equation (11) grows with $k$ as the softmax spreads more weight away from the argmax to the remaining $k - 1$ values.

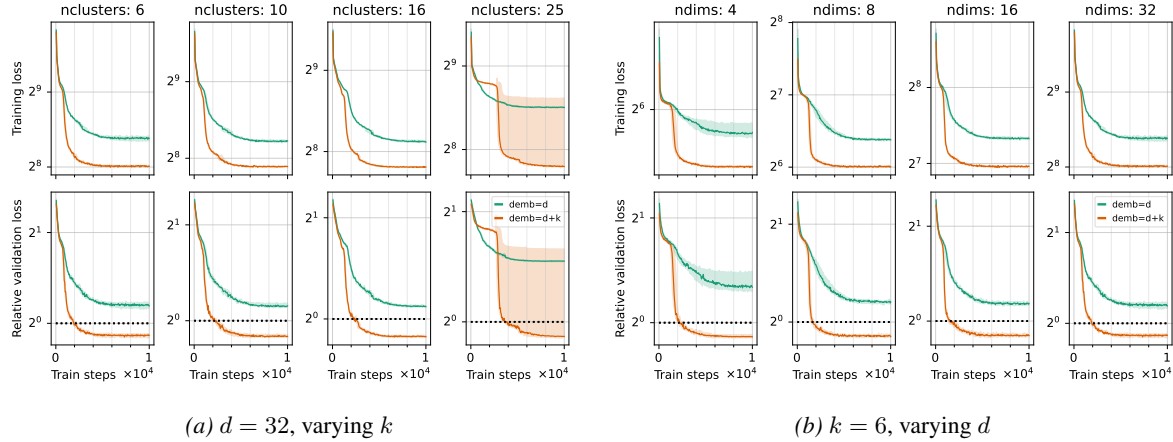

*(a) $d = 32$, varying $k$*        *(b) $k = 6$, varying $d$*

*Figure 10.* Training convergence for varying problem setups. Here we select $\lambda = 0.1$. The top row in each figure tracks the training loss (see Figure 9 for details) aggregated over 10 random trials with the ribbons indicating the inter-quartile range (*lower is better*). The bottom row in each figure tracks the relative validation loss computed using the validation clustering tasks (see Figure 9 for details) (*lower is better*). The dashed line at $y = 1$ in the bottom row of each figure denotes the performance of a single step of Lloyd's algorithm.

In Figure 10b, we study the effect of varying the data dimensionality $d$ from 4 to 32 (for a fixed $k$ and $\lambda$), and see that, for all values of $d$, the learned single-layer transformer continues to outperform the single-step Lloyd's algorithm by a significant margin. In both Figure 10a and Figure 10b, we see that the ablation with the transformer with $d_{\mathsf{emb}} = d$ continues to be unable to match the Lloyd's algorithm, indicating that the additional $k$ dimensions provide a crucial boost in expressivity.

**Multi-step clustering.** While we train the single-layer transformer effectively for single-step clustering, we can easily make use of this learned transformer for multi-step clustering as follows for input $(X, C^{(0)})$:

- For clustering step $t = 1, \ldots, T$:
  - Embed $(X, C^{(t)})$ as $(X^{(0)}, \tilde{C}^{(0)})$ with $X^{(0)} \leftarrow \begin{bmatrix} X \\ 0_{k \times n} \end{bmatrix}$ and $\tilde{C}^{(0)} \leftarrow \begin{bmatrix} C^{(t)} \\ I_k \end{bmatrix}$.
  - For input $(X^{(0)}, \tilde{C}^{(0)})$ to the single-layer transformer, get output $(X^{(1)}, \tilde{C}^{(1)})$.
  - Use the first $d$ rows of $\tilde{C}^{(1)}$ as the updated centers $C^{(t)}$.
- Return $C^{(T)}$.

We use this procedure for the clustering results in Figure 4.

### B.4. Further empirical evaluation of trained transformer

Here we consider the $k$-means transformer trained on clustering tasks generated from mixtures of Gaussians (as described earlier), and evaluate it on clustering tasks generated from other mixture distributions (Cauchy, Laplace, Gumbel, LogNormal). The results in Table 1 and Figure 11 show that all distribution mixtures, the trained $k$-means transformer is able to improve the $k$-means objective over the initial centers, and outperforms Lloyd's algorithm for all but the tasks generated from mixtures of Cauchy distributions.

*Table 1.* We evaluate the trained $k$-means transformer with $d = 32, k = 10$ (trained on clustering tasks generated from mixtures of Gaussians) on clustering tasks generated from other mixture distribution families — namely Cauchy, Laplace, Gumbel and LogNormal. We compare its performance to Lloyd's algorithm and report the average log $k$-means objective after 20 iterations aggregated over 320 test tasks. See Figure 11 for the complete trajectory.

| Family | Initial objective | Lloyd's | $k$-means transformer |
|---|---|---|---|
| Normal | 6.3430 | 5.2636 | 5.0052 |
| Cauchy | 3.4078 | 3.2528 | 3.3443 |
| Gumbel | 6.2408 | 5.2557 | 5.0723 |
| Laplace | 6.0447 | 5.1014 | 4.9505 |
| LogNormal | 6.2965 | 5.2406 | 5.0026 |

*Figure 11.* Same at Table 1 but here we present the complete trajectory of the $k$-means objective over the 20 clustering iterations.

Next we apply the trained $k$-means transformer to 15 OpenML datasets (Bischl et al., 2025). This model is trained *solely on synthetic clustering tasks* as discussed earlier with $d = 32, k = 10$. Hence, strong generalization is only *expected* on new clustering tasks obtained from the same generative process. We have no such control on the OpenML datasets. We execute Lloyd's and the trained model for 10 iterations, and report the mean (std) log kmeans objective (40 trials). In all cases, the trained model improves the kmeans objective from the initial centers, demonstrating that the model has learned some generalized clustering mechanisms. However, it underperforms Lloyd's by 1-7% as the real data does not match the training task distribution.

*Table 2.* We execute Lloyd's algorithm and the trained $k$-means transformer (with $d = 32, k = 10$) for 10 iterations on 15 OpenML (Bischl et al., 2025) datasets with dimensionality greater than or equal to 32, and report the mean$_{\pm \text{std}}$ log $k$-means objective aggregated over 40 trials. Each trial consists of (i) different subsets of size 1024 from the original set if the number of samples in the dataset is greater than 1024, otherwise all samples are used, (ii) different dimension subsets of size 32 if the original number of dimensions is greater than 32, otherwise all the data dimensions are used, and (iii) different sets of initial centers.

| Dataset | Initial obj | Lloyd's | $k$-means trf | relative diff |
|---|---|---|---|---|
| gina-agnostic | $8.35_{\pm 0.14}$ | $7.88_{\pm 0.13}$ | $7.95_{\pm 0.13}$ | 0.93% |
| musk | $6.98_{\pm 0.15}$ | $6.38_{\pm 0.11}$ | $6.45_{\pm 0.11}$ | 1.09% |
| mfeat-factors | $6.96_{\pm 0.10}$ | $6.33_{\pm 0.06}$ | $6.41_{\pm 0.06}$ | 1.22% |
| GermanCredit | $8.41_{\pm 0.11}$ | $7.90_{\pm 0.10}$ | $8.00_{\pm 0.10}$ | 1.24% |
| mfeat-fourier | $7.01_{\pm 0.07}$ | $6.47_{\pm 0.03}$ | $6.55_{\pm 0.03}$ | 1.25% |
| ionosphere | $6.36_{\pm 0.10}$ | $6.00_{\pm 0.04}$ | $6.08_{\pm 0.03}$ | 1.30% |
| feat-karhunen | $6.83_{\pm 0.06}$ | $6.28_{\pm 0.03}$ | $6.38_{\pm 0.03}$ | 1.53% |
| ada-agnostic | $7.68_{\pm 0.14}$ | $7.15_{\pm 0.11}$ | $7.26_{\pm 0.11}$ | 1.58% |
| optdigits | $7.73_{\pm 0.12}$ | $7.13_{\pm 0.11}$ | $7.27_{\pm 0.12}$ | 1.85% |
| sonar | $5.51_{\pm 0.06}$ | $4.97_{\pm 0.06}$ | $5.09_{\pm 0.08}$ | 2.48% |
| sylva-agnostic | $6.67_{\pm 0.30}$ | $6.21_{\pm 0.29}$ | $6.39_{\pm 0.29}$ | 2.89% |
| oil-spill | $6.29_{\pm 0.32}$ | $5.57_{\pm 0.20}$ | $5.79_{\pm 0.16}$ | 3.95% |
| ailerons | $5.27_{\pm 0.20}$ | $4.69_{\pm 0.14}$ | $4.96_{\pm 0.14}$ | 5.81% |
| pc4 | $5.82_{\pm 0.18}$ | $5.29_{\pm 0.15}$ | $5.61_{\pm 0.17}$ | 5.99% |
| pc3 | $5.61_{\pm 0.20}$ | $5.08_{\pm 0.12}$ | $5.44_{\pm 0.19}$ | 7.07% |

# C. General Architecture

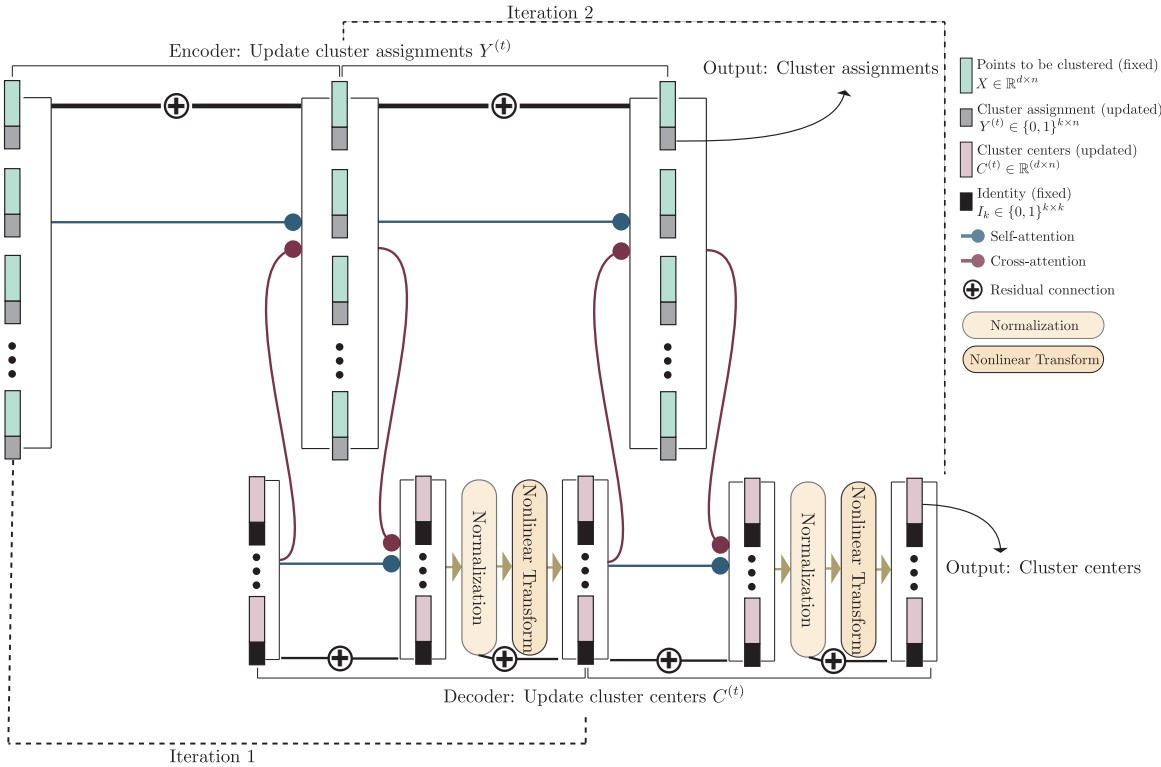

*Figure 12.* **The transformer architecture as an algorithmic skeleton for $k$-means clustering.** We generalize our $k$-means transformer architecture to give increased algorithmic expressibility by incorporating normalization (e.g., LayerNorm (Ba et al., 2016) and RM-SNorm (Zhang & Sennrich, 2019)) and nonlinear transformations (e.g., multilayer perceptrons) — standard ingredients of the transformer architecture. As detailed in Table 3, different choices for each of these transformer components can lead to different (and novel) clustering algorithms.

*Table 3.* **Different configurations of the transformer architecture result in different clustering algorithms.** We consider different configurations of (i) the encoder and decoder self-attention (Enc-SA and Dec-SA respectively), (ii) the cross-attention with encoder tokens as queries and decoder tokens as keys/values (Enc-CA), (iii) the cross-attention with decoder tokens as queries and encoder tokens as keys/values (Dec-CA), (iv) the normalization (Norm) and (v) the token-wise nonlinear transformation using a feed-forward network with a single hidden layer (NLT). Figure 12 visualizes this architectural template. SMA denotes the standard soft-max attention while LSMA denotes the limiting soft-max attention where the soft-max temperature approaches its lower limit. LinA denotes linear attention where the attention scores do not go through a nonlinear function (such as $\exp(\cdot)$) before the normalization. GSMA denotes the Gumbel-soft-max attention where we sample a key/value token based on the attention scores (Jang et al., 2017). NMA denotes the "norm-max" attention resulting from the norm $\alpha$-negentropy regularized $\arg\max$ (Blondel et al., 2020; dos Santos et al., 2024); note that soft-max is Shannon negentropy regularized $\arg\max$. SSMA denotes sparse versions of the soft-max such as sparse-max (Martins & Astudillo, 2016) or top-$k$ attention (Gupta et al., 2021). The $\langle,\rangle$ and $\ell_2$ denote the dot-product based (standard) and negative squared Euclidean distance based attention scores respectively. The problems and algorithms marked with $\star$ signify that we can also consider variations such as spherical clustering and use of $\langle,\rangle$ attention scores instead of $\ell_2$ attention scores. The '$\checkmark$' denotes the use of a component, while '-' marks unused component.

| Enc-SA | Enc-CA | Dec-SA | Dec-CA | Norm | NLT | Algorithm | Appendix |
|--------|--------|--------|--------|------|-----|-----------|----------|
| $\ell_2$-LSMA | $\ell_2$-LSMA | $\langle,\rangle$-LSMA | $\langle,\rangle$-LSMA | - | - | Lloyd's algorithm / Euclidean $k$-means | E |
| $\langle,\rangle$-LSMA | $\langle,\rangle$-LSMA | $\langle,\rangle$-LSMA | $\langle,\rangle$-LSMA | - | $\checkmark$ | Lloyd's algorithm / Euclidean $k$-means | F |
| $\langle,\rangle$-LSMA | $\langle,\rangle$-LSMA | $\langle,\rangle$-LSMA | $\langle,\rangle$-LSMA | $\checkmark$ | - | Lloyd's algorithm / Spherical $k$-means | G |
| $\ell_2$-LSMA | $\ell_2$-SMA | $\langle,\rangle$-LSMA | $\langle,\rangle$-LinA | - | - | $\star$Soft $k$-means | E |
| $\ell_2$-LSMA | $\ell_2$-LSMA | $\langle,\rangle$-LSMA | $\ell_2$-NMA | - | - | $\star$Trimmed $k$-means | H |
| $\ell_2$-LSMA | $\ell_2$-LSMA | $\langle,\rangle$-LSMA | $\ell_2$-SSMA | - | - | $\star$Robust $k$-means | I.1 |
| $\ell_2$-LSMA | $\ell_2$-LSMA | $\langle,\rangle$-LSMA | $\ell_2$-GSMA | - | - | $\star$Euclidean $k$-medoids | I.2 |

## D. Selective Attention for Clustering

As discussed above, a key component of attention, and so of transformer processing, is the *soft-max* operation, denoted $\sigma_\gamma$, which for inverse-temperature $\gamma > 0$ and $a \in \mathbb{R}^m$ has entries

$$\sigma_\gamma(a)^i \triangleq \frac{\exp(\gamma a^i)}{\sum_{j\in[m]} \exp(\gamma a^j)}, \tag{13}$$

where $a^i$ denotes the $i$-th entry in the vector $a$. We can write Equation (13) as $\sigma_\gamma(a) = \exp(\gamma a)(\mathbf{1}_m^\top \exp(\gamma a))^{-1}$, where vector $\exp(\gamma a)$ is entrywise, having $\exp(\gamma a)^i = \exp(\gamma a^i)$, so that $\mathbf{1}_m^\top \exp(\gamma a)$ is the denominator in Equation (13). If $A \in \mathbb{R}^{m\times n}$, then applying the soft-max operation to each column of $A$ can thus be expressed as $\sigma_\gamma(A) = \exp(\gamma A)\mathsf{diag}(\mathbf{1}_m^\top \exp(\gamma A))^{-1}$, where $\exp(\gamma A)$ is entrywise, and the $\mathsf{diag}$ operator fills the nonzero entries of an $n \times n$ diagonal matrix with the entries of an $n$-dimensional vector. Given a vector $z \in \mathbb{R}^m$, $z^\top \sigma_\gamma(a)$ is the real number that is a linear combination of the entries of $z$, using the weights of $\sigma_\gamma(a)$. Similarly, for $Z \in \mathbb{R}^{d\times m}$ and matrix $A \in \mathbb{R}^{m\times n}$, each column of $Z\sigma_\gamma(A) \in \mathbb{R}^{d\times n}$ is a linear combination of columns of $Z$, using the weights of a column of $A$.

The attention operation is said to be applied from $X \in \mathbb{R}^{d\times n}$ to $Z \in \mathbb{R}^{d\times m}$, using (learnable) attention parameters $Q, K, V \in \mathbb{R}^{d\times d}$, by computing [4]

$$\mathcal{A}^\gamma_{\langle,\rangle}(X, Z; Q, K, V) \triangleq VZ\sigma_\gamma(A) \in \mathbb{R}^{d\times n}, \tag{14}$$

where $A = (KZ)^\top(QX) \in \mathbb{R}^{m\times n}$ is the pre-soft-max attention score matrix. For vector $x$, let $\sigma_{\mathtt{lin}}(x)^i \triangleq x^i / \sum_{i'} x^{i'}$. Then *linear attention* is $\mathcal{A}^{\mathtt{lin}}_{\langle,\rangle}(X, Z; Q, K, V) \triangleq VZ\sigma_{\mathtt{lin}}(A)$, where $A$ is as above, and $\sigma_{\mathtt{lin}}$ is applied columnwise. Note that here the $(j, i)$-th entry of the attention score matrix has $A^{ji} = \langle Kz_j, Qx_i \rangle$, (the $\langle,\rangle$-attention in Table 3). However, one can also define the attention score as $A^{ji} = -\|Kz_j - Qx_i\|^2$ (the $\ell_2$-attention in Table 3) (Tsai et al., 2019), and this version will be denoted as $\mathcal{A}^\gamma_2(\ldots)$. This is more formally expressed as follows.

**Definition D.1.** For matrices $X \in \mathbb{R}^{d\times n}$ with columns $x_i$, and $C \in \mathbb{R}^{d\times k}$ with columns $c_j$, define the matrix $\mathsf{DP}(C, X) \in \mathbb{R}^{k\times n}$ with $\mathsf{DP}(C, X)^{ji} = -\|c_j - x_i\|^2$.

With these definitions,

$$\mathcal{A}^\gamma_2(X, Z; Q, K, V) \triangleq VZ\sigma_\gamma(\mathsf{DP}(KZ, QX)) \in \mathbb{R}^{d\times n}. \tag{15}$$

When $\sigma_{\mathtt{lin}}$ is used instead of $\sigma_\gamma$, the result is $\mathcal{A}^{\mathtt{lin}}_2(X, Z; Q, K, V)$.

Note that for integer $m \geq 0$, with $0_{p\times q}$ denoting the $(p \times q)$ zero matrix,

$$\mathsf{DP}\left(\begin{bmatrix} C \\ 0_{m\times k} \end{bmatrix}, \begin{bmatrix} X \\ 0_{m\times n} \end{bmatrix}\right) = \mathsf{DP}(C, X) \tag{16}$$

When $\gamma = \infty$ (discussed below), we omit it, and have $\mathcal{A}_{\langle,\rangle}(\ldots)$ and $\mathcal{A}_2(\ldots)$. Note that, there is no $\gamma$ in linear attention as it has no effect on linear attention (the $\gamma$ values get canceled out in the numerator and denominator of $\sigma_{\mathtt{lin}}$.

As $\gamma \to \infty$, the vector $\sigma_\gamma(a)$ becomes concentrated on the indices of the largest coordinates of $a$. We can define $\sigma_\infty(a) = \lim_{\gamma\to\infty} \sigma_\gamma(a)$, the limiting soft-max. If $M$ is the set of indices at which $a$ is maximum, then in general $\sigma_\infty(a)^i = \frac{1}{|M|}$ if $i \in M$, and is zero otherwise. For example, the vector $[1, 0, 1, -2]$ has $M = \{1, 3\}$, so $|M| = 2$ and $\sigma_\infty([1, 0, 1, -2]) = [1/2, 0, 1/2, 0]$. Since the attention is split evenly across the maxima, $\sigma_\infty$ has been called AHAT or *averaging hard attention* in the literature (Strobl et al., 2024). If $|M| = 1$, that is, $a$ has a unique maximum coordinate, then $\sigma_\infty(a)$ is a one-hot encoding, with a single nonzero entry $\frac{1}{|M|} = 1$ at the index in $M$, and is zero otherwise. As an example, $\sigma_\infty([1.4, 1.5, -10]) = [0, 1, 0]$.

To see the use of the unique ($|M| = 1$) case in the context of clustering, let the $n$ points to be clustered $\{x_1, \ldots, x_n\}$ serve as the query tokens, arranged into $X \in \mathbb{R}^{d\times n}$, the current $k$ centers $\{c_1, \ldots, c_k\}$ as the key tokens, arranged as $C \in \mathbb{R}^{d\times k}$, and we use the $\ell_2$-attention score $A = \mathsf{DP}(C, X)$. That is, the attention score matrix $A = \mathsf{DP}(C, X) \in \mathbb{R}^{k\times n}$ has $A^{ji} = -\|c_j - x_i\|^2$. Then $\sigma_\infty(A)$, which applies soft-max to each column of $A$, will give us the per-point cluster

---

[4]In general, the $Q, K \in \mathbb{R}^{d'\times d}$ for some $d' \neq d$ for increased expressivity, but we will only require $d' = d$, and hence forego this additional notation.

assignment, with the $i$-th column of $\sigma_\infty(A)$ denoting the cluster assigned to point $i$. (It follows also that $C\sigma_\infty(A) \in \mathbb{R}^{d \times n}$ has column $i$ equal to the center closest to $x_i$.)

Consider the following example where we have $n = 9$ points (queries) and $k = 3$ cluster centers (keys), resulting in a $3 \times 9$ attention score matrix (with negative squared Euclidean distance based scores):

$$A = \begin{bmatrix} -0.1 & -0.2 & -0.7 & -0.2 & -0.8 & -0.3 & -0.8 & -0.2 & -0.1 \\ -0.5 & -0.1 & -0.1 & -0.4 & -0.7 & -0.2 & -0.9 & -0.3 & -0.2 \\ -0.7 & -0.3 & -0.2 & -0.6 & -0.1 & -0.3 & -0.1 & -0.7 & -0.7 \end{bmatrix}. \tag{17}$$

Considering the cluster assignment step in Figure 1A (line 4), points $\{1, 4, 8, 9\}$ should be assigned to cluster 1, points $\{2, 3, 6\}$ to cluster 2, and points $\{5, 7\}$ to cluster 3. Applying the columnwise limiting soft-max to this attention matrix results in the following:

$$\sigma_\infty(A) = \begin{bmatrix} 1 & 0 & 0 & 1 & 0 & 0 & 0 & 1 & 1 \\ 0 & 1 & 1 & 0 & 0 & 1 & 0 & 0 & 0 \\ 0 & 0 & 0 & 0 & 1 & 0 & 1 & 0 & 0 \end{bmatrix}, \tag{18}$$

with each column corresponding to a cluster index in one-hot encoded form: columns $\{1, 4, 8, 9\}$ are $[1, 0, 0]$ (or cluster 1), columns $\{2, 3, 6\}$ are $[0, 1, 0]$ (cluster 2), and columns $\{5, 7\}$ are $[0, 0, 1]$ (cluster 3).

The non-unique case ($|M| > 1$) of the limiting soft-max is also useful for clustering. Consider the $k$ centers as the queries, and the $n$ points as the keys, and an attention score matrix $B \in \mathbb{R}^{n \times k}$ where $B_{ij} = 1$ if point $i$ is assigned to cluster $j$, or $0$ otherwise. Note that, this attention score matrix happens to be exactly the transpose of the cluster assignment matrix obtained in Equation (18) — that is, $B = \sigma_\infty(A)^\top$. Applying columnwise limiting soft-max $\sigma_\infty(B)$ to this attention score matrix $B$ results in each column being inversely scaled by the number of nonzeros in that column (which is equal to the number of points assigned to the corresponding cluster). Continuing with the example in Equation (18), we will have:

$$B = \sigma_\infty(A)^\top = \begin{bmatrix} 1 & 0 & 0 \\ 0 & 1 & 0 \\ 0 & 1 & 0 \\ 1 & 0 & 0 \\ 0 & 0 & 1 \\ 0 & 1 & 0 \\ 0 & 0 & 1 \\ 1 & 0 & 0 \\ 1 & 0 & 0 \end{bmatrix} \quad \text{with} \quad \sigma_\infty(B) = \begin{bmatrix} 1/4 & 0 & 0 \\ 0 & 1/3 & 0 \\ 0 & 1/3 & 0 \\ 1/4 & 0 & 0 \\ 0 & 0 & 1/2 \\ 0 & 1/3 & 0 \\ 0 & 0 & 1/2 \\ 1/4 & 0 & 0 \\ 1/4 & 0 & 0 \end{bmatrix}. \tag{19}$$

The product of the matrix $X$ and the limiting soft-max attention matrix $\sigma_\infty(B)$ results in exactly the new cluster centers computed in the Lloyd's algorithm for Euclidean $k$-means (Figure 1A, line 5). This can be demonstrated in the above example as follows:

$$X \cdot \sigma_\infty(B) = [x_1,\ x_2,\ x_3,\ x_4,\ x_5,\ x_6,\ x_7,\ x_8,\ x_9] \begin{bmatrix} 1/4 & 0 & 0 \\ 0 & 1/3 & 0 \\ 0 & 1/3 & 0 \\ 1/4 & 0 & 0 \\ 0 & 0 & 1/2 \\ 0 & 1/3 & 0 \\ 0 & 0 & 1/2 \\ 1/4 & 0 & 0 \\ 1/4 & 0 & 0 \end{bmatrix} \tag{20}$$

$$= \left[ \frac{(x_1 + x_4 + x_8 + x_9)}{4},\ \frac{(x_2 + x_3 + x_6)}{3},\ \frac{(x_5 + x_7)}{2} \right],$$

which are exactly the cluster centers computed in Lloyd's. Thus, Equation (18) and Equation (20) demonstrate that the attention mechanism is uniquely suited for clustering, with the attention mechanism directly computing the necessary quantities in the Lloyd's algorithm.

Making a general claim as in the examples above, we have the following lemma, after a definition

**Definition D.2.** For $C \in \mathbb{R}^{d \times k}$, $X \in \mathbb{R}^{d \times n}$, define the matrix $Y^{\gamma}(C, X) \triangleq \sigma_{\gamma}(\mathtt{DP}(C, X)) \in \mathbb{R}^{k \times n}$. When $\gamma = \infty$, we can omit it.

**Lemma D.1.** *For $X \in \mathbb{R}^{d \times n}$ and $C \in \mathbb{R}^{d \times k}$, $A = \mathtt{DP}(C, X)$ has $\sigma_{\infty}(A) = Y(C, X)$. Also for $B = \sigma_{\infty}(A)^{\top}$, $\sigma_{\infty}(B)_{ij} = 1/m_j$ when $x_i$ has $c_j$ closest and there are $m_j$ such $x_i$, and $\sigma_{\infty}(B)_{ij}$ is zero otherwise. Finally, if $C$ denotes the centers at one iteration of Lloyd's algorithm, then*

$$X\sigma_{\infty}(B) = X\sigma_{\infty}(Y(C, A)^{\top}) = X\sigma_{\infty}(\sigma_{\infty}(\mathtt{DP}(C, X))^{\top})$$

*denotes the centers at the next iteration.*

*Proof.* Omitted. □

We note that for $x \in \{0, 1\}^d$, $\sigma_{\infty}(x) = \sigma_{\mathtt{lin}}(x)$, and if also $\|x\| = 1$ ($x$ is a natural basis vector), then $\sigma_{\infty}(x) = x$. Thus for example $\sigma_{\infty}(Y(C, A)^{\top}) = \sigma_{\mathtt{lin}}(Y(C, A)^{\top})$, and $\sigma_{\infty}(I_d) = \sigma_{\mathtt{lin}}(I_d) = I_d$.

---

**Algorithm 1** Lloyd's algorithm to cluster $n$ points into $k$ clusters

---

1   Input: Instances $\{x_i \in \mathbb{R}^d, i \in [n]\}$
2   Input: Number of iterations $T$
3   Input: Initial centers $\{c_j^{(0)} \in \mathbb{R}^d, j \in [k]\}$
4   **for** $t = 1, \ldots, T$ **do**
5     $\forall i \in [n], \; a_i^{(t)} \leftarrow \arg\min_{j \in [k]} \left\| x_i - c_j^{(t-1)} \right\|^2$
6     $\forall j \in [k], \; c_j^{(t)} \leftarrow \arg\min_{c \in \mathbb{R}^d} \sum_{i \in [n]: a_i^{(t)} = j} \left\| x_i - c \right\|^2$
7   **return** $\{c_j^{(T)}, j \in [k]\}$

---

**Algorithm 2** Soft $k$-means clustering

---

1   Input: Instances $\{x_i \in \mathbb{R}^d, i \in [n]\}$
2   Input: Number of iterations $T$
3   Input: Initial centers $\{c_j^{(0)} \in \mathbb{R}^d, j \in [k]\}$
4   Input: Inverse temperature $\gamma > 0$
5   **for** $t = 1, \ldots, T$ **do**
    // Partially assign points to clusters
6     $\forall i \in [n], j \in [k], \; w_{ij}^{(t)} \leftarrow \dfrac{\exp\left(-\gamma \left\| x_i - c_j^{(t-1)} \right\|^2\right)}{\sum_{j'=1}^{k} \exp\left(-\gamma \left\| x_i - c_{j'}^{(t-1)} \right\|^2\right)}$
    // Update centers with partial assignments
7     $\forall j \in [k], \; c_j^{(t)} \leftarrow \arg\min_{c \in \mathbb{R}^d} \sum_{i=1}^{n} w_{ij}^{(t)} \left\| x_i - c \right\|^2$
8   **return** $\{c_j^{(T)}, j \in [k]\}$

---

## E. In-context Lloyd's for Euclidean Clustering

Here we will establish transformer configurations that lead to the in-context execution of Lloyd's algorithm (Figure 1A, also Algorithm 1). We will present the theoretical result in a general form which will also allow us to show a similar result for the soft $k$-means clustering algorithm (Dunn, 1974; Bezdek, 2013) shown in Algorithm 2. Note that soft $k$-means clustering allows points to be partially assigned to multiple clusters (line 5, Algorithm 2. Given these partial assignments, the centers are updated making use of all the instances (as opposed to only instances assigned to the cluster as in discrete $k$-means). The partial assignment weights are used to compute the new cluster centers (line 6, Algorithm 2).

### E.1. Transformers with Euclidean attention

While we demonstrate above how the (limiting) soft-max attention between points and cluster centers can be used to implement Lloyd's algorithm, we have not shown how to repeatedly perform the cluster assignment and centers update steps in-context in a transformer architecture. For this, we use specific embeddings of the points and centers, and make use of the self-attention and residual connections in a transformer.

**Embedding the input.** We embed each point $x_i \in \mathbb{R}^d$ into a $(d + k)$-dimensional vector $[x_i^\top, y_i^\top]^\top$, where the last $k$ dimensions are placeholders for the (evolving) cluster assignment of this point. Initially, we can set $y_i = 0_k$ for all $i = 1, \ldots, n$. Arranging these $n$ embeddings as $(d + k)$-dimensional columns gives us the initial encoder embeddings $X^{(0)} \in \mathbb{R}^{(d+k) \times n}$, which we can also write as $\begin{bmatrix} X \\ Y^{(0)} \end{bmatrix}$, where $X \in \mathbb{R}^{d \times n}$ is the input, and $Y^{(0)} \in \mathbb{R}^{k \times n}$ is the matrix of cluster assignments.

We also embed the initial centers $c_j \in \mathbb{R}^d$ into a $(d + k)$-dimensional vector $[c_j^\top, e_j^\top]^\top$, where $e_j$ is the $j$-th characteristic vector (a one-hot vector with the $j$-th entry equal to 1), and the last $k$ dimensions denote the index of the cluster, while the first $d$ dimensions are the placeholders for the (evolving) cluster centers. Stacking the per-center embeddings gives us the initial decoder embeddings $\tilde{C}^{(0)} \in \mathbb{R}^{(d+k) \times n}$, which we can also write as $\tilde{C}^{(0)} \triangleq \begin{bmatrix} C^{(0)} \\ I_k \end{bmatrix}$.

In short, we use matrices

$$X^{(t)} \triangleq \begin{bmatrix} X \\ Y^{(t)} \end{bmatrix}$$

$$Y^{(t)} = Y(C^{(t-1)}, X) \text{ has } Y_{ji}^{(t)} = 1 \text{ if } x_i \text{ assigned to } c_j, \text{ and } 0 \text{ otherwise}$$

$$\tilde{C}^{(t)} \triangleq \begin{bmatrix} C^{(t)} \\ I_k \end{bmatrix} \tag{21}$$

**Updates in-context.** Consider the $(t+1)$-th transformer layer, with $X^{(t)}$ and $C^{(t)}$ denoting the input to the $(t+1)$-th encoder and decoder layer respectively. Then we define the attention based updates as follows:

$$X^{(t+1)} \leftarrow X^{(t)} + \mathcal{A}_2(X^{(t)}, \tilde{C}^{(t)}, Q_1, K_1, V_1) + \mathcal{A}_2(X^{(t)}, X^{(t)}, Q_2, K_2, V_2)$$

$$\tilde{C}^{(t+1)} \leftarrow \underbrace{\tilde{C}^{(t)}}_{\text{residual}} + \underbrace{\mathcal{A}_{\langle,\rangle}(\tilde{C}^{(t)}, X^{(t+1)}, Q_3, K_3, V_3)}_{\text{cross-attention}} + \underbrace{\mathcal{A}_{\langle,\rangle}(\tilde{C}^{(t)}, C^{(t)}, Q_4, K_4, V_4)}_{\text{self-attention}}, \tag{22}$$

where, recalling Figure 2, in both the encoder and decoder updates, the first term on the right-hand-side corresponds to the residual connections, the second term corresponds to the cross-attention, and the last term corresponds to the self-attention. Here, the $\{(Q_\mu, K_\mu, V_\mu), \mu = 1, \ldots, 4\}$ correspond to the parameters of the transformer. As we discussed above, the cross-attention computes the updated per-point cluster assignments and cluster centers. We can set the transformer parameters appropriately so that:

1. The first $d$ dimensions of the encoder tokens $X^{(t)}$ do not change, while the last $k$ dimensions are appropriately updated to match the cluster assignments as per the Lloyd's iteration (Algorithm 1, line 4).
2. The last $k$ dimensions of the decoder tokens $C^{(t)}$ do not change, while the first $d$ dimensions are appropriately updated to match new cluster centers as per the Lloyd's iteration (Algorithm 1, line 5).
3. The encoder cross-attention computes the update to the last $k$ dimensions of the encoder tokens as shown in Equation (18), while the decoder cross-attention computes the update to the first $d$ dimensions of the decoder tokens as shown in Equation (20).
4. The encoder (decoder) self-attention exactly computes the additive inverse of the current last $k$ dimensions (first $d$ dimensions) of the encoder (decoder) tokens, and when combined with the residual connection in Equation (22), it effectively resets the current cluster assignments (cluster centers) to zero, allowing the additive cross-attention update to modify the encoder (decoder) tokens as per Lloyd's algorithm.

More precisely, we show the following:

**Theorem E.1.** *Given as input a set of points $S = \{x_1, \ldots, x_n\}$ and initial centers $C = \{c_1^{(0)}, \ldots, c_k^{(0)}\}$, or arranged in columns, $X \in \mathbb{R}^{d \times n}$ and $C \in \mathbb{R}^{d \times k}$, there exists parameters $\{(Q_\mu, K_\mu, V_\mu), \mu = 1, \ldots, 4\}$ for the transformer*

$$X^{(t+1)} \leftarrow X^{(t)} + \mathcal{A}_2^\gamma(X^{(t)}, \tilde{C}^{(t)}, Q_1, K_1, V_1) + \mathcal{A}_2(X^{(t)}, X^{(t)}, Q_2, K_2, V_2)$$

$$\tilde{C}^{(t+1)} \leftarrow \tilde{C}^{(t)} + \mathcal{A}_{\langle,\rangle}^\xi(\tilde{C}^{(t)}, X^{(t+1)}, Q_3, K_3, V_3) + \mathcal{A}_{\langle,\rangle}(\tilde{C}^{(t)}, \tilde{C}^{(t)}, Q_4, K_4, V_4) \tag{23}$$

*such that if $\gamma = \xi = \infty$, then the output of the $T$-th transformer layer exactly matches the output of Lloyd's algorithm (Algorithm 1) with the same input after $T$ iterations. That is, $Y^{(t+1)} = Y(C^{(t)}, X)$, and $C^{(t+1)} = X\sigma_\infty((Y^{(t+1)})^\top)$, for $t \in [T-1]$, as in Lloyd's algorithm. If $\gamma < \infty$ and $\xi = \mathtt{lin}$ (that is, linear attention), then $Y^{(t+1)} = Y^\gamma(C^{(t)}, X)$ and $C^{(t+1)} = X\sigma_{\mathtt{lin}}(Y^{(t+1)})^\top$, for $t \in [T-1]$, as in soft $k$-means (Algorithm 2).*

We will specify our matrices for attention in a compact form, using the following notation.

**Definition E.1.** For integers $0 < i \leq j \leq d$, let $I_d^{i:j}$ denote the linear operator on vectors $x \in \mathbb{R}^d$ with $d \geq j$ such that $(I_d^{i:j}x)_k = x_k$ for $i \leq k \leq j$, and zero otherwise. (We can implement $I_d^{i:j}$ as a diagonal matrix with entries $(I_d^{i:j})_{k,k} = 1$ for $k = i, i+1, \ldots, j$, and all other entries zero.) When $j = d$, we may write $I_d^{i:}$, and when $i = 1$, we may write $I_d^{:j}$. When $d$ can be inferred from context (e.g., from matrices in a product with $I_d^{i:j}$), then we may write e.g. $I_d^{i:}$.

Note that for $x \in \mathbb{R}^d$, $I_d^{i:j}x$ zeroes out the entries $x$ with indices in $1, 2 \ldots, i-1$ and $j+1, j+2, \ldots d$, and leaves the rest alone.

*Proof of Theorem E.1.* Let $K_1 = Q_1 = K_2 = Q_2 = I_{d+k}^{:d}$, and $V_1 = -V_2 = I_{d+k}^{d+1:}$. Note that $I_{d+k}^{:d}$ and $I_{d+k}^{d+1:}$ are $(d+k) \times (d+k)$ matrices as defined in Definition E.1, operating on vector in $\mathbb{R}^{d+k}$.

To update cluster assignments for Lloyd's algorithm, that is, to compute $X^{(t+1)}$ from $X^{(t)}$ and $\tilde{C}^{(t)}$, we compute

$$
\begin{aligned}
&\mathcal{A}_2^\gamma(X^{(t)}, \tilde{C}^{(t)}, I_{d+k}^{:d}, I_{d+k}^{:d}, I_{d+k}^{d+1:}) + \mathcal{A}_2(X^{(t)}, X^{(t)}, I_{d+k}^{:d}, I_{d+k}^{:d}, -I_{d+k}^{d+1:}) \\
&= I_{d+k}^{d+1:}\tilde{C}^{(t)}\sigma_\gamma(\mathrm{DP}(I_{d+k}^{:d}\tilde{C}^{(t)}, I_{d+k}^{:d}X^{(t)})) - I_{d+k}^{d+1:}X^{(t)}\sigma_\infty(\mathrm{DP}(I_{d+k}^{:d}X^{(t)}, I_{d+k}^{:d}X^{(t)})) \\
&= \begin{bmatrix} 0_{d\times k} \\ I_k \end{bmatrix} \sigma_\gamma\left(\mathrm{DP}\left(\begin{bmatrix} C^{(t)} \\ 0_{k\times k} \end{bmatrix}, \begin{bmatrix} X \\ 0_{k\times n} \end{bmatrix}\right)\right) - \begin{bmatrix} 0_{d\times n} \\ Y^{(t)} \end{bmatrix}\sigma_\infty(\mathrm{DP}\left(\begin{bmatrix} X \\ 0_{k\times n} \end{bmatrix}, \begin{bmatrix} X \\ 0_{k\times n} \end{bmatrix}\right)) \\
&= \begin{bmatrix} 0_{d\times k} \\ I_k \end{bmatrix} \sigma_\gamma(\mathrm{DP}(C^{(t)}, X)) - \begin{bmatrix} 0_{d\times n} \\ Y^{(t)} \end{bmatrix}\sigma_\infty(\mathrm{DP}(X, X)) \qquad \text{using (16)} \\
&= \begin{bmatrix} 0_{d\times k} \\ I_k \end{bmatrix} Y^\gamma(C^{(t)}, X) - \begin{bmatrix} 0_{d\times n} \\ Y^{(t)} \end{bmatrix} I_n. \qquad \text{using Lemma D.1, } \sigma_\infty(\mathrm{DP}(X,X)) = I_n \\
&= \begin{bmatrix} 0_{d\times n} \\ Y^\gamma(C^{(t)}, X) - Y^{(t)} \end{bmatrix}
\end{aligned}
$$
(24)

We add this quantity to $X^{(t)} = \begin{bmatrix} X \\ Y^{(t)} \end{bmatrix}$ and assign it to $X^{(t+1)}$, so $X^{(t+1)} \leftarrow \begin{bmatrix} X \\ Y^\gamma(C^{(t)}, X) \end{bmatrix}$. In particular, $Y^{(t+1)} = Y(C^{(t)}, X)$, as claimed, when $\gamma = \infty$, as in Lloyd's algorithm, Algorithm 1, and when $\gamma < \infty$, $Y^{(t+1)} = Y^\gamma(C^{(t)}, X)$, as in soft $k$-means, Algorithm 2.

Let $Q_3 = K_3 = Q_4 = K_4 = I_{d+k}^{d+1:}$, and $V_3 = -V_4 = I_{d+k}^{:d}$. To update cluster centers for Lloyd's algorithm, we compute

$$
\begin{aligned}
&\mathcal{A}_{\langle,\rangle}^\xi(\tilde{C}^{(t)}, X^{(t+1)}; I_{d+k}^{d+1:}, I_{d+k}^{d+1:}, I_{d+k}^{:d}) + \mathcal{A}_{\langle,\rangle}(\tilde{C}^{(t)}, \tilde{C}^{(t)}; I_{d+k}^{d+1:}, I_{d+k}^{d+1:}, -I_{d+k}^{:d}) \\
&= I_{d+k}^{:d}X^{(t+1)}\sigma_\xi((X^{(t+1)})^\top I_{d+k}^{d+1:}I_{d+k}^{d+1:}\tilde{C}^{(t)}) - I_{d+k}^{:d}\tilde{C}^{(t)}\sigma_\infty((\tilde{C}^{(t)})^\top I_{d+k}^{d+1:}I_{d+k}^{d+1:}\tilde{C}^{(t)}) \\
&= \begin{bmatrix} X \\ 0_{k\times n} \end{bmatrix}\sigma_\xi\left(\begin{bmatrix} 0_{d\times n} \\ Y^{(t+1)} \end{bmatrix}^\top \begin{bmatrix} 0_{d\times k} \\ I_k \end{bmatrix}\right) - \begin{bmatrix} C^{(t)} \\ 0_{k\times k} \end{bmatrix}\sigma_\infty\left(\begin{bmatrix} 0_{d\times k} \\ I_k \end{bmatrix}^\top \begin{bmatrix} 0_{d\times k} \\ I_k \end{bmatrix}\right) \\
&= \begin{bmatrix} X \\ 0_{k\times n} \end{bmatrix}\sigma_\xi((Y^{(t+1)})^\top) - \begin{bmatrix} C^{(t)} \\ 0_{k\times k} \end{bmatrix}\sigma_\infty(I_k) \\
&= \begin{bmatrix} X\sigma_\xi((Y^{(t+1)})^\top) \\ 0_{k\times k} \end{bmatrix} - \begin{bmatrix} C^{(t)} \\ 0_{k\times k} \end{bmatrix}.
\end{aligned}
$$
(25)

We assign this quantity plus $C^{(t)}$ to $\tilde{C}^{(t+1)}$, that is, $\tilde{C}^{(t+1)} \leftarrow \begin{bmatrix} C^{(t)} \\ I_k \end{bmatrix} + \begin{bmatrix} X\sigma_\xi((Y^{(t+1)})^\top) \\ 0_{k\times k} \end{bmatrix} - \begin{bmatrix} C^{(t)} \\ 0_{k\times k} \end{bmatrix} = \begin{bmatrix} X\sigma_\xi((Y^{(t+1)})^\top) \\ I_k \end{bmatrix}$.
When $\xi$ is $\infty$, we have $C^{(t+1)} = X\sigma_\infty((Y^{(t+1)})^\top)$, as claimed for Algorithm 1. When $\xi$ is $\mathtt{lin}$, we have $C^{(t+1)} = X\sigma_{\mathtt{lin}}((Y^{(t+1)})^\top)$, as required for Algorithm 2.

Thus, we have shown that there are transformers, using Euclidean attention, that implement an iteration of Lloyd's algorithm, for one setting of parameters, and soft $k$-means, for another. $\qquad\square$

*Remark* E.1. In Theorem E.1, the query/key parameters are given by $K_1 = Q_1 = K_2 = Q_2 = I_{d+k}^{:d}$. However, note that Euclidean distances between points $x, x'$ are rotation invariant — that is, $\|x - x'\|^2 = \|Rx - Rx'\|^2$ for any rotation matrix $R$. Thus, $K_1 = Q_1 = K_2 = Q_2 = RI_{d+k}^{:d}$ would be valid parameters for any rotation matrix $R \in \mathbb{R}^{(d+k)\times(d+k)}$. As there are infinite such rotation matrices, there are thus infinite such query/key parameters that give us the desired behaviour. Similarly, for dot-product attention with query/key projection matrices $Q_\mu, K_\mu$, any unitary matrix $U \in \mathbb{R}^{d_{\mathrm{emb}} \times d_{\mathrm{emb}}}$ with $UU^\top = I_{d_{\mathrm{emb}}}$ gives us an alternate valid parameterization of $UQ_\mu, UK_\mu$, as we have $\langle x, x'\rangle = \langle Ux, Ux'\rangle$, thus $\langle Q_\mu x, K_\mu x'\rangle = \langle UQ_\mu x, UK_\mu x'\rangle$.

## F. Dot-product Attention and Quadratic Activations for Euclidean Clustering

Here we show how to apply dot product (usual) attention, rather than Euclidean attention. The algorithm uses much the same scheme as for Euclidean attention, but with points "lifted" to higher dimension using squared norms, so that squared Euclidean distance is expressible as a dot product. We lift the input points in preprocessing, and augment the system with a quadratic activation function, to compute the updated lifting of updated cluster center points.

**Definition F.1.** For integers $0 < i < d$, let $\pi_d^i \in \mathbb{R}^{d \times d}$ denote twice the negation of the matrix that swaps entries $i-1$ and $i$, that is, $(\pi_d^i)^{jj} = 2$, for $j \neq i-1, i$, and $(\pi_d^i)^{(i-1)i} = (\pi_d^i)^{i(i-1)} = 2$, and all other entries zero. When $d$ can be inferred from context (e.g. appearance in a matrix product), then it may be omitted.

**Definition F.2.** For $x \in \mathbb{R}^d$, let $\ell(x) \in \mathbb{R}^{d+2}$ be its lifted representation with $\ell(x)^i = x^i$ for $i \in [d]$, and with $\ell(x)^{d+1} = \|x\|^2$, and $\ell(x)^{d+2} = -1/2$. For $X \in \mathbb{R}^{d \times n}$, let $\ell(X) \in \mathbb{R}^{(d+2) \times n}$ denote columnwise application of $\ell()$.

**Lemma F.1.** *Let* $x, y \in \mathbb{R}^d$. *Then* $-\|x-y\|^2 = \ell(x) \cdot \pi_{d+2}^{d+2} \ell(y)$. *Let* $X \in \mathbb{R}^{d \times n}$ *and* $C \in \mathbb{R}^{d \times k}$. *Then* $\ell(C)^\top \pi_{d+2}^{d+2} \ell(X) = \mathrm{DP}(C, X)$ *and*

$$\sigma_\gamma(\ell(C)^\top \pi_{d+2}^{d+2} \ell(X)) = Y^\gamma(C, X),$$

*including for* $\gamma = \infty$.

*Proof.* Since $\pi_{d+2}^{d+2} \ell(y)$ swaps the last two entries of $\ell(y) \in \mathbb{R}^{d+2}$ and multiplies all entries with 2, we have $\ell(x)^\top \pi_{d+2}^{d+2} \ell(y) = 2[x \cdot y + (-1/2)\|y\|^2 + \|x\|^2 (-1/2)] = -\|x-y\|^2$, yielding the first claim. We have

$$(\ell(C)^\top \pi_{d+2}^{d+2} \ell(X))^{ji} = \ell(c_j)^\top \pi_{d+2}^{d+2} \ell(x_i) = -\|c_j - x_i\|^2 = \mathrm{DP}(C, X)^{ji},$$

so that $\sigma_\gamma(\ell(C)^\top \pi_{d+2}^{d+2} \ell(X)) = Y^\gamma(C, X)$ follows, as claimed. $\square$

**Theorem F.1.** *Given as input a set of points* $S = \{x_1, \ldots, x_n\}$ *and initial centers* $C = \{c_1^{(0)}, \ldots, c_k^{(0)}\}$, *or arranged in columns,* $X \in \mathbb{R}^{d \times n}$ *and* $C^{(0)} \in \mathbb{R}^{d \times k}$, *there exists* $C^{(t)} \in \mathbb{R}^{d \times k}$, $Y^{(t)} \in \mathbb{R}^{k \times n}$, $X^{(t)} = \begin{bmatrix} \ell(X) \\ Y^{(t)} \end{bmatrix} \in \mathbb{R}^{(d+k+2) \times n}$, $\tilde{C}^{(t)} = \begin{bmatrix} \ell(C^{(t)}) \\ I_k \end{bmatrix} \in \mathbb{R}^{(d+k+2) \times n}$, *neural network* $\mathcal{N}$ *using quadratic activations, and parameters* $\{(Q_\mu, K_\mu, V_\mu), \mu = 1, \ldots, 4\}$ *for the transformer*

$$\begin{aligned}
X^{(t+1)} &\leftarrow X^{(t)} + \mathcal{A}_{\langle,\rangle}(X^{(t)}, \tilde{C}^{(t)}, Q_1, K_1, V_1) + \mathcal{A}_{\langle,\rangle}(X^{(t)}, X^{(t)}, Q_2, K_2, V_2) \\
\tilde{C}^{(t+1/2)} &\leftarrow \tilde{C}^{(t)} + \mathcal{A}_{\langle,\rangle}(\tilde{C}^{(t)}, X^{(t+1)}, Q_3, K_3, V_3) + \mathcal{A}_{\langle,\rangle}(\tilde{C}^{(t)}, \tilde{C}^{(t)}, Q_4, K_4, V_4) \\
\tilde{C}^{(t+1)} &\leftarrow C^{(t+1/2)} + \mathcal{N}(C^{(t+1/2)})
\end{aligned} \tag{26}$$

*such that the output of the $T$-th transformer layer exactly matches the output of Lloyd's algorithm (Algorithm 1) with the same input after $T$ iterations. That is,* $Y^{(t+1)} = Y(C^{(t)}, X)$, *and* $C^{(t+1)} = X\sigma_\infty((Y^{(t+1)})^\top)$, *for* $t \in [T-1]$, *as in Lloyd's algorithm (Algorithm 1).*

By using $\gamma < \infty$ and $\sigma_{\mathtt{lin}}$ as in Theorem E.1, we can also obtain a version that implements soft $k$-means using inner product attention.

*Proof.* Let $d' = d + 2$. Let $K_1 = K_2 = I_{d'+k}^{:d'}$, $Q_1 = Q_2 = \pi_{d'+k}^{d'}$, and $V_1 = -V_2 = I_{d'+k}^{d'+1:}$. We compute

$$
\mathcal{A}_{\langle,\rangle}(X^{(t)}, \tilde{C}^{(t)}, I_{d'+k}^{:d'}, \pi_{d'+k}^{d'}, I_{d'+k}^{d'+1:}) + \mathcal{A}_{\langle,\rangle}(X^{(t)}, X^{(t)}, I_{d'+k}^{:d'}, \pi_{d'+k}^{d'}, -I_{d'+k}^{d'+1:})
$$

$$
= I_{d'+k}^{d'+1:}\tilde{C}^{(t)}\sigma_\infty((\tilde{C}^{(t)})^\top I_{d'+k}^{:d'}\pi_{d'+k}^{d'}X^{(t)})) - I_{d'+k}^{d'+1:}X^{(t)}\sigma_\infty((X^{(t)})^\top I_{d'+k}^{:d'}\pi_{d'+k}^{d'}X^{(t)}))
$$

$$
= I_{d'+k}^{d'+1:}\begin{bmatrix}\ell(C^{(t)})\\ I_k\end{bmatrix}\sigma_\infty\left(\begin{bmatrix}\ell(C^{(t)})\\ I_k\end{bmatrix}^\top I_{d'+k}^{:d'}\pi_{d'+k}^{d'}\begin{bmatrix}\ell(X)\\ Y^{(t)}\end{bmatrix}\right) - I_{d'+k}^{d'+1:}\begin{bmatrix}\ell(X)\\ Y^{(t)}\end{bmatrix}\sigma_\infty\left(\begin{bmatrix}\ell(X)\\ Y^{(t)}\end{bmatrix}^\top I_{d'+k}^{:d'}\pi_{d'+k}^{d'}\begin{bmatrix}\ell(X)\\ Y^{(t)}\end{bmatrix}\right)
$$

$$
= \begin{bmatrix}0_{d'\times k}\\ I_k\end{bmatrix}\sigma_\infty\left(\begin{bmatrix}\ell(C^{(t)})\\ 0_{k\times k}\end{bmatrix}^\top\begin{bmatrix}\pi_{d'}^{d'}\ell(X)\\ Y^{(t)}\end{bmatrix}\right) - \begin{bmatrix}0_{d'\times n}\\ Y^{(t)}\end{bmatrix}\sigma_\infty\left(\begin{bmatrix}\ell(X)\\ 0_{k\times n}\end{bmatrix}^\top\begin{bmatrix}\pi_{d'}^{d'}\ell(X)\\ Y^{(t)}\end{bmatrix}\right)
$$

$$
= \begin{bmatrix}0_{d'\times k}\\ I_k\end{bmatrix}\sigma_\infty(\ell(C^{(t)})^\top\pi_{d'}^{d'}\ell(X))) - \begin{bmatrix}0_{d'\times n}\\ Y^{(t)}\end{bmatrix}\sigma_\infty(\ell(X)^\top\pi_{d'}^{d'}\ell(X))
$$

$$
= \begin{bmatrix}0_{d'\times k}\\ I_k\end{bmatrix}Y(C^{(t)}, X) - \begin{bmatrix}0_{d'\times n}\\ Y^{(t)}\end{bmatrix}I_n \quad \text{using Lemma F.1, and } Y(X, X) = I_n
$$

$$
= \begin{bmatrix}0_{d'\times n}\\ Y(C^{(t)}, X) - Y^{(t)}\end{bmatrix}.
$$

(27)

Adding this to $X^{(t)} = \begin{bmatrix}\ell(X)\\ Y^{(t)}\end{bmatrix}$, we assign $\begin{bmatrix}\ell(X)\\ Y(C^{(t)}, X)\end{bmatrix}$ to $X^{(t+1)}$, so $Y^{(t+1)} = Y(C^{(t)}, X)$, as claimed.

Let $Q_3 = K_3 = Q_4 = K_4 = I_{d'+k}^{d':}$, $V_3 = I_{d'+k}^{:d}$, and $V_4 = -I_{d'+k}^{:d}$. To update cluster centers for Lloyd's algorithm, we compute

$$
\mathcal{A}_{\langle,\rangle}(\tilde{C}^{(t)}, X^{(t+1)}; I_{d'+k}^{d':}, I_{d'+k}^{d':}, I_{d'+k}^{:d}) + \mathcal{A}_{\langle,\rangle}(\tilde{C}^{(t)}, \tilde{C}^{(t)}; I_{d'+k}^{d':}, I_{d'+k}^{d':}, -I_{d'+k}^{:d})
$$

$$
= I_{d'+k}^{:d}X^{(t+1)}\sigma_\infty((X^{(t+1)})^\top I_{d'+k}^{d':}I_{d'+k}^{d':}\tilde{C}^{(t)}) - I_{d'+k}^{:d}\tilde{C}^{(t)}\sigma_\infty((\tilde{C}^{(t)})^\top I_{d'+k}^{d':}I_{d'+k}^{d':}\tilde{C}^{(t)})
$$

$$
= \begin{bmatrix}X\\ 0_{(k+2)\times n}\end{bmatrix}\sigma_\infty\left(\begin{bmatrix}0_{d'\times n}\\ Y^{(t+1)}\end{bmatrix}^\top\begin{bmatrix}0_{d'\times k}\\ I_k\end{bmatrix}\right) - \begin{bmatrix}\ell(C^{(t)})\\ 0_{k\times k}\end{bmatrix}\sigma_\infty\left(\begin{bmatrix}0_{d'\times k}\\ I_k\end{bmatrix}^\top\begin{bmatrix}0_{d'\times k}\\ I_k\end{bmatrix}\right)
$$

(28)

$$
= \begin{bmatrix}X\\ 0_{(k+2)\times n}\end{bmatrix}\sigma_\infty((Y^{(t+1)})^\top) - \begin{bmatrix}\ell(C^{(t)})\\ 0_{k\times k}\end{bmatrix} \quad \text{using Lemma D.1.}
$$

This quantity is added to $\tilde{C}^{(t)} = \begin{bmatrix}\ell(C^{(t)})\\ I_k\end{bmatrix}$, obtaining $\tilde{C}^{(t+1/2)} = \begin{bmatrix}X\sigma_\infty((Y^{(t+1)})^\top)\\ 0_{2\times k}\\ I_k\end{bmatrix}$. The matrix $X\sigma_\infty((Y^{(t+1)})^\top)$ will become $C^{(t+1)}$, so the final claim of the theorem will follow.

However, it remains to fill in the squared norm entries in row $d + 1$ and $-\mathbf{1}_k^\top/2$ in row $d + 2$, to maintain $\ell(C^{(t+1)})$. We use a feed-forward network and a quadratic activation function, $\tau(\cdot)$, where for matrix $A$, $\tau(A)^{ij} = (A^{ij})^2$. Our network, with input layer weights $\begin{bmatrix}I_d & 0_{d\times(k+2)}\end{bmatrix}$ and hidden layer weights and bias $\begin{bmatrix}0_{d\times d}\\ \mathbf{1}_d^\top\\ 0_{(k+1)\times d}\end{bmatrix}$ and $\begin{bmatrix}0_{(d+1)\times k}\\ -\mathbf{1}_k^\top/2\\ 0_{k\times k}\end{bmatrix}$ respectively, computes the following:

$$
\mathcal{N}(C^{(t+1/2)}) \triangleq \begin{bmatrix}0_{d\times d}\\ \mathbf{1}_d^\top\\ 0_{(k+1)\times d}\end{bmatrix}\tau\left(\begin{bmatrix}I_d & 0_{d\times(k+2)}\end{bmatrix}\begin{bmatrix}C^{(t+1)}\\ 0_{2\times k}\\ I_k\end{bmatrix}\right) + \begin{bmatrix}0_{(d+1)\times k}\\ -\mathbf{1}_k^\top/2\\ 0_{k\times k}\end{bmatrix}
$$

$$
= \begin{bmatrix}0_{d\times d}\\ \mathbf{1}_d^\top\\ 0_{(k+1)\times d}\end{bmatrix}\tau(C^{(t+1)}) + \begin{bmatrix}0_{(d+1)\times k}\\ -\mathbf{1}_k^\top/2\\ 0_{k\times k}\end{bmatrix} = \begin{bmatrix}0_{d\times k}\\ (\hat{C}^{(t+1)})^\top\\ -\mathbf{1}_k^\top/2\\ 0_{k\times k}\end{bmatrix},
$$

(29)

where $\hat{C}^{(t+1)} \in \mathbb{R}^k$ is the vector of the squared norms of the updated centers, with $(\hat{C}^{(t+1)})^j = \left\|c_j^{(t+1)}\right\|^2$. Thus, with the feed-forward networks and the residual connection, we obtain

$$
\tilde{C}^{(t+1/2)} + \mathcal{N}(C^{(t+1/2)}) = \begin{bmatrix}C^{(t+1)}\\ 0_{2\times k}\\ I_k\end{bmatrix} + \begin{bmatrix}0_{d\times k}\\ (\hat{C}^{(t+1)})^\top\\ -\mathbf{1}_k^\top/2\\ 0_{k\times k}\end{bmatrix} = \begin{bmatrix}C^{(t+1)}\\ (\hat{C}^{(t+1)})^\top\\ -\mathbf{1}_k^\top/2\\ I_k\end{bmatrix} = \begin{bmatrix}\ell(C^{(t+1)})\\ I_k\end{bmatrix} = \tilde{C}^{(t+1)},
$$

using the lifted version of $C^{(t+1)}$, as claimed. $\square$

---

**Algorithm 3** Lloyd's algorithm variation for spherical clustering

---

1  Input: Instances $\{x_i \in \mathcal{S}^{d-1}, i \in [n]\}$
2  Input: Initial centers $\{c_j^{(0)} \in \mathcal{S}^{d-1}, j \in [k]\}$
3  **for** $t = 1, \ldots, T$ **do**
        // Assign points to clusters
4      $\forall i \in [n]$, $a_i^{(t)} \leftarrow \arg\max_{j \in [k]} \left\langle x_i, c_j^{(t-1)} \right\rangle$
        // Update cluster centers on the sphere
5      $\forall j \in [k]$, $c_j^{(t)} \leftarrow \arg\max_{c \in \mathcal{S}^{d-1}} \sum_{i:a_i^{(t)}=j} \langle x_i, c \rangle$
6  **return** $\{c_j^{(T)}, j \in [k]\}$

---

## G. Spherical Clustering

The spherical $k$-means clustering problem can be solved via Lloyd's iterations with the modifications that all the points lie on a sphere, and the cluster centers also need to be on the sphere. The corresponding algorithm is presented in Algorithm 3. The cluster assignments are performed by maximizing instance-center dot-products instead of minimizing their respective Euclidean distances (line 4, Algorithm 3) as we assume that all instances $x_i$ and all centers $c_j$ are on the unit sphere (that is, $\|x_i\| = 1, \|c_j\| = 1$). The cluster center updates also find new centers on the unit sphere that maximize the sum of the dot-products to all instances in the cluster (line 5, Algorithm 3). Before presenting a theorem regarding spherical $k$-means, we need some definitions and lemmas.

**Definition G.1** (LN (Ba et al., 2016)). Let $\text{LN} : \mathbb{R}^d \to \mathbb{R}^d$ be defined as

$$\text{LN}(z; \epsilon, \gamma, \beta) = \frac{z - 1_d \mathbb{E}[z]}{\sqrt{\text{Var}(z) + \epsilon}} \cdot \gamma + \beta.$$

Here $\mathbb{E}[z] = \frac{1}{d} 1_d^\top z$ and $\text{Var}(z) = \frac{1}{d} \sum_{i \in [d]} (z_i - \mathbb{E}[z])^2$. For a matrix $C \in \mathbb{R}^{d \times k}$, $\text{LN}(C)$ denotes $\text{LN}()$ applied columnwise.

**Lemma G.1.** *For $z$ with $\mathbb{E}[z] = 0$, we have $\text{LN}(z; 0, \frac{1}{\sqrt{d}}, 0) = \frac{z}{\|z\|}$.*

We can put the key computations of spherical Lloyd's algorithm into our notation.

**Definition G.2.** Given $X \in \mathbb{R}^{d \times n}, C \in \mathbb{R}^{d \times k}$, define $Y_S(C, X) \in \mathbb{R}^{k \times n}$ as having $Y_S(C, X)^{j'i} = 1$ if $c_{j'}$ maximizes the dot product with $x_i$ over all $j \in [k]$, and zero otherwise.

**Lemma G.2.** *Given $X \in \mathbb{R}^{d \times n}$ with $1_d^\top X = 0_{1 \times n}$, $C \in \mathbb{R}^{d \times k}$, $Y_S(C, X) = \sigma_\infty(C^\top X)$. For given $j \in [k]$, let*

$$c_j' = \arg\max_{c \in \mathbb{R}^d, \|c\|=1} \sum_{i:Y_S(C,X)^{ji}=1} c^\top x_i.$$

*Then $c_j' = \text{LN}(w; 0, \frac{1}{\sqrt{d}}, 0)$, where $w = \sum_{i:Y_S(C,X)^{ji}=1} x_i$. If $C'$ has columns comprising the $c_j'$, then $C' = \text{LN}(W)$, where $W = X \sigma_\infty(Y_S(C, X)^\top)$.*

*Proof.* Omitted. Note that $1_d^\top X = 0_{1 \times n}$ is needed to have $C' = \text{LN}(W)$. $\qquad\square$

**Theorem G.1.** *Given $X \in \mathbb{R}^{d \times n}$ where the columns of $X$ are unit vectors with zero-sum coordinates, that is, $\|x_i\| = 1$ and $1_d^\top x_i = 0$ for $i \in [n]$. Given also $C^{(0)} \in \mathbb{R}^{d \times k}$, there exists $C^{(t)} \in \mathbb{R}^{d \times k}$, $Y^{(t)} \in \mathbb{R}^{k \times n}$, and $X^{(t)} = \begin{bmatrix} X \\ Y^{(t)} \end{bmatrix} \in \mathbb{R}^{(d+k) \times n}$, and $\tilde{C}^{(t)} = \begin{bmatrix} C^{(t)} \\ I_k \end{bmatrix}, \tilde{C}^{(t+1/2)} = \begin{bmatrix} C^{(t+1/2)} \\ I_k \end{bmatrix} \in \mathbb{R}^{(d+k) \times k}$, together with parameters $\{(Q_\mu, K_\mu, V_\mu), \mu = 1, \ldots, 4\}$ for the transformer*

$$X^{(t+1)} \leftarrow X^{(t)} + \mathcal{A}_{\langle,\rangle}(X^{(t)}, \tilde{C}^{(t)}, Q_1, K_1, V_1) + \mathcal{A}_{\langle,\rangle}(X^{(t)}, X^{(t)}, Q_2, K_2, V_2)$$

$$\tilde{C}^{(t+1/2)} \leftarrow \tilde{C}^{(t)} + \mathcal{A}_{\langle,\rangle}(\tilde{C}^{(t)}, X^{(t+1)}, Q_3, K_3, V_3) + \mathcal{A}_{\langle,\rangle}(\tilde{C}^{(t)}, \tilde{C}^{(t)}, Q_4, K_4, V_4)$$

$$\tilde{C}^{(t+1)} \leftarrow \begin{bmatrix} \text{LN}(C^{(t+1/2)}) \\ I_k \end{bmatrix}. \tag{30}$$

*such that the output of the $T$-th transformer layer exactly matches the output of spherical Lloyd's algorithm (Algorithm 3) with the same input after $T$ iterations. That is, $Y^{(t+1)} = \sigma_\infty((C^{(t)})^\top X)$, and $C^{(t+1)} = \mathrm{LN}(X\sigma_\infty((Y^{(t+1)})^\top))$, for $t \in [T-1]$, as in spherical Lloyd's algorithm, Algorithm 3.*

*Proof.* Let $K_1 = K_2 = Q_1 = Q_2 = I_{d+k}^{:d}$ and $V_1 = -V_2 = I_{d+k}^{d+1:}$. We compute

$$
\begin{aligned}
&\mathcal{A}_{\langle,\rangle}(X^{(t)}, \tilde{C}^{(t)}, I_{d+k}^{:d}, I_{d+k}^{:d}, I_{d+k}^{d+1:}) + \mathcal{A}_{\langle,\rangle}(X^{(t)}, X^{(t)}, I_{d+k}^{:d}, I_{d+k}^{:d}, -I_{d+k}^{d+1:}) \\
&= I_{d+k}^{d+1:}\tilde{C}^{(t)}\sigma_\infty((\tilde{C}^{(t)})^\top I_{d+k}^{:d} I_{d+k}^{:d} X^{(t)})) - I_{d+k}^{d+1:}X^{(t)}\sigma_\infty((X^{(t)})^\top I_{d+k}^{:d} I_{d+k}^{:d} X^{(t)})) \\
&= I_{d+k}^{d+1:}\begin{bmatrix} C^{(t)} \\ I_k \end{bmatrix}\sigma_\infty\left(\begin{bmatrix} C^{(t)} \\ I_k \end{bmatrix}^\top I_{d+k}^{:d}\begin{bmatrix} X \\ Y^{(t)} \end{bmatrix}\right) - I_{d+k}^{d+1:}\begin{bmatrix} X \\ Y^{(t)} \end{bmatrix}\sigma_\infty\left(\begin{bmatrix} X \\ Y^{(t)} \end{bmatrix}^\top I_{d+k}^{:d}\begin{bmatrix} X \\ Y^{(t)} \end{bmatrix}\right) \\
&= \begin{bmatrix} 0_{d\times k} \\ I_k \end{bmatrix}\sigma_\infty\left(\begin{bmatrix} C^{(t)} \\ 0_{k\times k} \end{bmatrix}^\top\begin{bmatrix} X \\ Y^{(t)} \end{bmatrix}\right) - \begin{bmatrix} 0_{d\times n} \\ Y^{(t)} \end{bmatrix}\sigma_\infty\left(\begin{bmatrix} X \\ 0_{k\times n} \end{bmatrix}^\top\begin{bmatrix} X \\ Y^{(t)} \end{bmatrix}\right) \\
&= \begin{bmatrix} 0_{d\times k} \\ I_k \end{bmatrix}\sigma_\infty((C^{(t)})^\top X)) - \begin{bmatrix} 0_{d\times n} \\ Y^{(t)} \end{bmatrix}\sigma_\infty(X^\top X) \\
&= \begin{bmatrix} 0_{d\times k} \\ I_k \end{bmatrix}Y_S(C^{(t)}, X) - \begin{bmatrix} 0_{d\times n} \\ Y^{(t)} \end{bmatrix}Y_S(X, X) \quad \text{using Lemma G.2} \\
&= \begin{bmatrix} 0_{d\times k} \\ I_k \end{bmatrix}Y_S(C^{(t)}, X) - \begin{bmatrix} 0_{d\times n} \\ Y^{(t)} \end{bmatrix}I_n. \quad \text{using } Y_S(X, X) = I_n \\
&= \begin{bmatrix} 0_{d\times n} \\ Y_S(C^{(t)}, X) - Y^{(t)} \end{bmatrix}.
\end{aligned}
\tag{31}
$$

Adding $X^{(t)} = \begin{bmatrix} X \\ Y^{(t)} \end{bmatrix}$ to this, we obtain $X^{(t+1)} \leftarrow \begin{bmatrix} X \\ Y_S(C^{(t)}, X) \end{bmatrix}$, so $Y^{(t+1)} = Y_S(C^{(t)}, X)$, as claimed.

Let $Q_3 = K_3 = Q_4 = K_4 = I_{d+k}^{d+1:}$, $V_3 = V_4 = I_{d+k}^{:d}$. To update cluster centers for spherical Lloyd's algorithm, we compute

$$
\begin{aligned}
&\mathcal{A}_{\langle,\rangle}(\tilde{C}^{(t)}, X^{(t+1)}; I_{d+k}^{d+1:}, I_{d+k}^{d+1:}, I_{d+k}^{:d}) + \mathcal{A}_{\langle,\rangle}(\tilde{C}^{(t)}, \tilde{C}^{(t)}; I_{d+k}^{d+1:}, I_{d+k}^{d+1:}, -I_{d+k}^{:d}) \\
&= I_{d+k}^{:d}X^{(t+1)}\sigma_\infty((X^{(t+1)})^\top I_{d+k}^{d+1:} I_{d+k}^{d+1:}\tilde{C}^{(t)}) - I_{d+k}^{:d}\tilde{C}^{(t)}\sigma_\infty((\tilde{C}^{(t)})^\top I_{d+k}^{d+1:} I_{d+k}^{d+1:}\tilde{C}^{(t)}) \\
&= \begin{bmatrix} X \\ 0_{k\times n} \end{bmatrix}\sigma_\infty\left(\begin{bmatrix} 0_{d\times n} \\ Y^{(t+1)} \end{bmatrix}^\top\begin{bmatrix} 0_{d\times k} \\ I_k \end{bmatrix}\right) - \begin{bmatrix} C^{(t)} \\ 0_{k\times k} \end{bmatrix}\sigma_\infty\left(\begin{bmatrix} 0_{d\times k} \\ I_k \end{bmatrix}^\top\begin{bmatrix} 0_{d\times k} \\ I_k \end{bmatrix}\right) \\
&= \begin{bmatrix} X \\ 0_{k\times n} \end{bmatrix}\sigma_\infty((Y^{(t+1)})^\top) - \begin{bmatrix} C^{(t)} \\ 0_{k\times k} \end{bmatrix}\sigma_\infty(I_k) \\
&= \begin{bmatrix} X\sigma_\infty((Y^{(t+1)})^\top - C^{(t)}) \\ 0_{k\times n} \end{bmatrix}
\end{aligned}
\tag{32}
$$

We add this quantity to $\tilde{C}^{(t)}$ to obtain $\tilde{C}^{(t+1/2)} \leftarrow \begin{bmatrix} X\sigma_\infty((Y^{(t+1)})^\top) \\ I_k \end{bmatrix}$, so that $C^{(t+1/2)} = X\sigma_\infty((Y^{(t+1)})^\top)$. We then assign $\begin{bmatrix} \mathrm{LN}(C^{(t+1/2)}) \\ I_k \end{bmatrix}$ to $\tilde{C}^{(t+1)}$, so that, with Lemma G.2, we have $C^{(t+1)} = \mathrm{LN}(X\sigma_\infty((Y^{(t+1)})^\top))$, as claimed. $\square$

*Remark* G.1. Note that there is a variation of $\mathrm{LN}$ (the layer normalization (Ba et al., 2016)) known as the root-mean-square or RMS normalization (Zhang & Sennrich, 2019) defined as $\left(\frac{z}{\|z\|} \cdot \gamma + \beta\right)$ which one can use in place of the layer normalization defined in Definition G.1 to remove the zero-sum coordinates assumption in Theorem G.1.

---

**Algorithm 4** Trimmed $k$-means

---

1  Input: Instances $\{x_i \in \mathbb{R}^d, i \in [n]\}$
2  Input: Initial centers $\{c_j^{(0)} \in \mathbb{R}^d, j \in [k]\}$
3  Input: Robustness parameters $\tau \in [0, 100)$
4  **for** $t = 1, \ldots, T$ **do**
    // Assign points to clusters
5      $\forall i \in [n], a_i^{(t)} \leftarrow \arg\min_{j \in [k]} \left\| x_i - c_j^{(t-1)} \right\|^2$
    // Obtain per-cluster outlier threshold
6      $\forall j \in [k], \rho_j \leftarrow \tau\text{-th \%-tile} \left( \left\{ \left\| x_i - c_j^{(t-1)} \right\|^2 \forall i \in [n] : a_i^{(t)} = j \right\} \right)$
    // Update cluster centers only with inliers
7      $\forall j \in [k], c_j^{(t)} \leftarrow \arg\min_{c \in \mathbb{R}^d} \sum_{i:a_i^{(t)}=j} \mathbb{I}\left( \left\| x_i - c_j^{(t-1)} \right\|^2 \leq \rho_j \right) \|x_i - c\|^2$

8  **return** $\{c_j^{(T)}, j \in [k]\}$

---

## H. Trimmed Euclidean Clustering

The trimmed $k$-means algorithm for robust clustering of Cuesta-Albertos et al. (1997) is given in Algorithm 4. Along with the instances to be clustered, and the initial cluster centers, this algorithm also takes as an input a percentile threshold $\tau \in [0, 100)$. The cluster assignment proceeds as in the standard Lloyd's algorithm by assigning each instance to its closest cluster center (line 5, Algorithm 4). After the cluster assignments, for each cluster $j \in [k]$, we find a Euclidean distance threshold $\rho_j$ corresponding to the $\tau$-th percentile of the distances of the assigned instances to the cluster center (line 6, Algorithm 4). Given this per-cluster threshold, the cluster centers are updated only using the assigned instances that are within the $\tau$-th percentile distance threshold, thus ignoring the outliers in the set of assigned instances.

While the limiting soft-max attention or LSMA $\sigma_\infty$ applied to the transposed cluster assignment matrix gives us the necessary weights to compute the updated cluster centers in Lloyd's algorithm (Algorithm 1), it does not give us the desired update in the trimmed $k$-means as we need to filter instances before computing the center updates. For this, we turn our attention to the $\alpha$-*norm-max* $\text{NM}_\gamma^\alpha$ defined as (Blondel et al., 2020; dos Santos et al., 2024)

$$\text{NM}_\gamma^\alpha(x) = \arg\max_{z:z^i \in [0,1], \sum_i z^i = 1} \langle \gamma x, z \rangle - \|z\|_\alpha, \tag{33}$$

where $\gamma$ serves the role of the inverse temperature, and the output is a vector on the probability simplex. As $\alpha \to \infty$, the objective in (33) becomes $(\langle \gamma x, z \rangle - \max_j z^j)$. Thus, the solution to this optimization problem (and thus, the output of the norm-max activation) involves the selection of a threshold $\rho_\gamma$ and setting $z^i \leftarrow 0$ if $\gamma x^i < \rho_\gamma$. For the remaining set of indices $P = \{i : \gamma x^i > \rho_\gamma\}$, $z^i \leftarrow 1/|P|$. This norm-max activation has the desired property of filtering certain entries based on a threshold (setting their values to zero), and then assigning equal values to all the non-zero entries.

Utilizing this activation in the updates in Theorem E.1

$$X^{(t+1)} \leftarrow X^{(t)} + \mathcal{A}_2(X^{(t)}, \tilde{C}^{(t)}, Q_1, K_1, V_1) + \mathcal{A}_2(X^{(t)}, X^{(t)}, Q_2, K_2, V_2)$$
$$\tilde{C}^{(t+1)} \leftarrow \tilde{C}^{(t)} + \mathcal{A}_2^\xi(\tilde{C}^{(t)}, X^{(t+1)}, Q_3, K_3, V_3) + \mathcal{A}_{\langle,\rangle}(\tilde{C}^{(t)}, \tilde{C}^{(t)}, Q_4, K_4, V_4), \tag{34}$$

where $\xi = \text{NM}_\gamma^\infty$ denotes the norm-max based attention, where the soft-max in the attention is replaced by the norm-max. The parameters $\{(Q_\mu, K_\mu, V_\mu), \mu = 1, 2, 4\}$ for the encoder side cross and self-attention, and the decoder side self-attention remain as in Theorem E.1 (that is, $K_1 = Q_1 = K_2 = Q_2 = I_{d+k}^{:d}$, $V_1 = -V_2 = I_{d+k}^{d+1:}$ and $Q_4 = K_4 = I_{d+k}^{d+1:}$, $V_4 = -I_{d+k}^{:d}$), and the only modification is the $(Q_3, K_3, V_3)$ in the decoder cross-attention for the cluster center update:

- First, we use $\ell_2$-attention in $\mathcal{A}_2^\xi(\tilde{C}^{(t)}, X^{(t+1)}, Q_3, K_3, V_3)$ instead of $\langle,\rangle$-attention in the cluster center update in Theorem E.1.

- Second, we utilize the norm-max attention $\texttt{NM}_\gamma^\infty$ on the Euclidean distance based attention scores.
- The value projection matrix $V_3 = I_{d+k}^{:d}$ as in Theorem E.1. However, the query and key projection matrices $Q_3, K_3$ are set as $Q_3 = K_3 = I_{d+k}^{:d} + \delta I_{d+k}^{d+1:}$, where $\delta$ is a large positive constant.

Given this parameter setting, the pre-activation attention scores in decoder cross-attention are given by $\texttt{DP}(K_3 X^{(t+1)}, Q_3 \tilde{C}^{(t)}) \in \mathbb{R}^{n \times k}$ where the $(i,j)$-th entry is given as

$$\texttt{DP}(K_3 X^{(t+1)}, Q_3 \tilde{C}^{(t)})^{ij} = \begin{cases} -\left\| x_i - c_j^{(t)} \right\|^2 & \text{if } y_i^{(t+1)} = e_j \text{ (that is, if } x_i \text{ assigned to } j^{\text{th}} \text{ cluster)} \\ -\left\| x_i - c_j^{(t)} \right\|^2 - 2\delta & \text{if } y_i^{(t+1)} \neq e_j. \end{cases} \tag{35}$$

Thus, for the $j$-th cluster (the $j$-th query), all the instances (keys) assigned to it (that is, $y_i^{(t+1)} = e_j$) get a score corresponding to their negative squared Euclidean distance $\left\| c_j^{(t)} - x_i \right\|^2$, while instances (keys) not assigned to this cluster ($y_i^{(t+1)} \neq e_j$) get a really low score $\approx -2\delta$ as $\delta$ is a large positive constant.

At this point, the norm-max $\texttt{NM}_\gamma^\infty$ applied column-wise to $\texttt{DP}(K_3 X^{(t+1)}, Q_3 \tilde{C}^{(t)})$ effectively selects a column-wise threshold $\rho_j^\gamma$ that depends on the column and the inverse-temperature $\gamma$, and sets all values less than $\rho_j^\gamma$ to zero, and the per-column non-zero values are given the same positive value such that the column sums to one. That is

$$\texttt{NM}_\gamma^\infty \left( \texttt{DP}(K_3 X^{(t+1)}, Q_3 \tilde{C}^{(t)}) \right)^{ij} = \begin{cases} 0 & \text{if } \texttt{DP}(K_3 X^{(t+1)}, Q_3 \tilde{C}^{(t)})^{ij} < \rho_j^\gamma \\ \frac{1}{|\{i : \texttt{DP}(K_3 X^{(t+1)}, Q_3 \tilde{C}^{(t)})^{ij} \geq \rho_j^\gamma\}|} & \text{otherwise.} \end{cases} \tag{36}$$

For an appropriately selected $\gamma > 0$, the zeros in column $j$ of $\texttt{NM}_\gamma^\infty(\texttt{DP}(K_3 X^{(t+1)}, Q_3 \tilde{C}^{(t)}))$ corresponds to the points not belonging to cluster $j$, and the outliers (if any) in cluster $j$, while the non-zero values are all equal and correspond to the inliers. Thus, $\texttt{NM}_\gamma^\infty(\texttt{DP}(K_3 X^{(t+1)}, Q_3 \tilde{C}^{(t)}))V_3 X$ will give us the centers of each cluster computed only using the inliers in the corresponding cluster, thus giving us the desired behaviour as in Algorithm 4. This cross-attention update, combined with the self-attention update (as in Theorem E.1) gives the desired cluster update in the transformer architecture.

# I. New Clustering Algorithms

Here we detail new algorithms we find by making modifications to our general transformer architecture (Figure 12).

## I.1. Another robust clustering algorithm

We can apply the attention mechanism in a different way obtain a kind of robust $k$-means, where only datapoints in the cluster of a given center affect the new center (as in Lloyd's algorithm), but among such datapoints, those closer to the center have a larger weight in determining the new cluster center than those farther away. That is, we can use a kind of outlier reduction in Lloyd's algorithm.

Given as input $X \in \mathbb{R}^{d \times n}$ and $C^{(0)} \in \mathbb{R}^{d \times k}$, this algorithm maintains

$$
\begin{aligned}
C^{(t)} &\in \mathbb{R}^{d \times k} \\
X^{(t)} &= \begin{bmatrix} \ell(X) \\ Y^{(t)} \\ \hat{Y}^{(t)} \end{bmatrix} \in \mathbb{R}^{(d+2k+2) \times n} \text{ with } Y^{(t)} = Y(C^{(t-1)}, X), \ \hat{Y}^{(t)} = \mathtt{DP}(C^{(t-1)}, X) + \delta Y^{(t)} \text{ for large } \delta > 0 \\
\tilde{C}^{(t)} &= \begin{bmatrix} \ell(C^{(t)}) \\ I_k \\ I_k \end{bmatrix} \in \mathbb{R}^{(d+2k+2) \times n}, \text{ with } C^{(t)} = X \sigma_\xi((\hat{Y}^{(t)})^\top),
\end{aligned}
\tag{37}
$$

where $\sigma_\xi$ could be $\sigma_\gamma$, sparse-max (Martins & Astudillo, 2016), $\sigma_{\mathtt{lin}}$ or some other map taking $\hat{Y}$ to $\Delta_n \triangleq \{y \in \mathbb{R}^n \mid y \geq 0, \mathbf{1}_n^\top y = 1\}$. The attention mechanism will use inner products, so a neural network $\mathcal{N}$ using quadratic activations to update $\ell(C^{(t)})$ will be needed as well (see §F).

Let $\sigma_I()$ denote the identity, so $\sigma_I(x) = x$ for $x \in \mathbb{R}^n$. Let $\mathcal{A}_{\langle,\rangle}^I$ denote attention using the identity operator.

Let $\mathcal{T}_i$ denote a triple of square matrices $Q_i, K_i, V_i$ of appropriate size. One round will be implemented as:

$$
\begin{aligned}
X^{(t+1/2)} &\leftarrow X^{(t)} + \mathcal{A}_{\langle,\rangle}(X^{(t)}, \tilde{C}^{(t)}, \mathcal{T}_1) + \mathcal{A}_{\langle,\rangle}(X^{(t)}, X^{(t)}, \mathcal{T}_2) \\
X^{(t+1)} &\leftarrow X^{(t+1/2)} + \mathcal{A}_{\langle,\rangle}^I(X^{(t+1/2)}, \tilde{C}^{(t)}, \mathcal{T}_3) \\
\tilde{C}^{(t+1/2)} &\leftarrow \tilde{C}^{(t)} + \mathcal{A}_{\langle,\rangle}^\xi(\tilde{C}^{(t)}, X^{(t+1)}, \mathcal{T}_4) + \mathcal{A}_{\langle,\rangle}(\tilde{C}^{(t)}, \tilde{C}^{(t)}, \mathcal{T}_5) \\
\tilde{C}^{(t+1)} &\leftarrow C^{(t+1/2)} + \mathcal{N}(C^{(t+1/2)})
\end{aligned}
$$

such that the output of the $T$-th transformer layer exactly matches the behavior described above. In particular, with $d' = d+2$ (note that $\ell(X) \in \mathbb{R}^{d' \times n}$ and $\ell(C^{(t)}) \in \mathbb{R}^{d' \times k}$), we have

$$
X^{(t+1/2)} - X^{(t)}
$$

$$
= \mathcal{A}_{\langle,\rangle}(X^{(t)}, \tilde{C}^{(t)}, I_{d'+2k}^{d'}, \pi_{d'+2k}^{d'}, I_{d'+2k}^{d'+1:d'+k}) + \mathcal{A}_{\langle,\rangle}(X^{(t)}, X^{(t)}, I_{d'+2k}^{d'}, \pi_{d'+2k}^{d'}, -I_{d'+2k}^{d'+1:})
$$

$$
= I_{d'+2k}^{d'+1:d'+k} \begin{bmatrix} \ell(C^{(t)}) \\ I_k \\ I_k \end{bmatrix} \sigma_\infty \left( \begin{bmatrix} \ell(C^{(t)}) \\ I_k \\ I_k \end{bmatrix}^\top I_{d'+2k}^{d'} \pi_{d'+2k}^{d'} \begin{bmatrix} \ell(X) \\ Y^{(t)} \\ \hat{Y}^{(t)} \end{bmatrix} \right) - I_{d'+2k}^{d'+1:} \begin{bmatrix} \ell(X) \\ Y^{(t)} \\ \hat{Y}^{(t)} \end{bmatrix} \sigma_\infty \left( \begin{bmatrix} \ell(X) \\ Y^{(t)} \\ \hat{Y}^{(t)} \end{bmatrix}^\top I_{d'+2k}^{d'} \pi_{d'+2k}^{d'} \begin{bmatrix} \ell(X) \\ Y^{(t)} \\ \hat{Y}^{(t)} \end{bmatrix} \right)
$$

$$
= \begin{bmatrix} 0_{d' \times k} \\ I_k \\ 0_{k \times k} \end{bmatrix} \sigma_\infty \left( \begin{bmatrix} \ell(C^{(t)}) \\ 0_{2k \times k} \end{bmatrix}^\top \begin{bmatrix} \pi_{d'+2k}^{d'} \ell(X) \\ Y^{(t)} \\ \hat{Y}^{(t)} \end{bmatrix} \right) - \begin{bmatrix} 0_{d' \times n} \\ Y^{(t)} \\ \hat{Y}^{(t)} \end{bmatrix} \sigma_\infty \left( \begin{bmatrix} \ell(X) \\ 0_{2k \times n} \end{bmatrix}^\top \begin{bmatrix} \pi_{d'+2k}^{d'} \ell(X) \\ Y^{(t)} \\ \hat{Y}^{(t)} \end{bmatrix} \right)
$$

$$
= \begin{bmatrix} 0_{d' \times k} \\ I_k \\ 0_{k \times k} \end{bmatrix} \sigma_\infty(\ell(C^{(t)})^\top \pi_{d'+2k}^{d'} \ell(X))) - \begin{bmatrix} 0_{d' \times n} \\ Y^{(t)} \\ \hat{Y}^{(t)} \end{bmatrix} \sigma_\infty(\ell(X)^\top \pi_{d'+2k}^{d'} \ell(X))
$$

$$
= \begin{bmatrix} 0_{d' \times k} \\ I_k \\ 0_{k \times k} \end{bmatrix} Y(C^{(t)}, X) - \begin{bmatrix} 0_{d' \times n} \\ Y^{(t)} \\ \hat{Y}^{(t)} \end{bmatrix} I_n \quad \text{using Lemma F.1, and } Y(X, X) = I_n
$$

$$
= \begin{bmatrix} 0_{d' \times n} \\ Y(C^{(t)}, X) - Y^{(t)} \\ -\hat{Y}^{(t)} \end{bmatrix},
$$

so that $X^{(t+1/2)} = X^{(t)} + \begin{bmatrix} 0_{d' \times n} \\ Y(C^{(t)}, X) - Y^{(t)} \\ -\hat{Y}^{(t)} \end{bmatrix} = \begin{bmatrix} \ell(X) \\ Y(C^{(t)}, X) \\ 0_{k \times n} \end{bmatrix}$.

---

**Algorithm 5** Robust $k$-means

---

1   Input: Instances $S = \{x_i \in \mathbb{R}^d, i \in [n]\}$

2   Input: Initial centers $\{c_j^{(0)} \in S, j \in [k]\}$

3   **for** $t = 1, \ldots, T$ **do**

     // Assign points to clusters

4      $\forall i \in [n], a_i^{(t)} \leftarrow \arg\min_{j \in [k]} \left\| x_i - c_j^{(t-1)} \right\|^2$

5      **for** $j = 1, \ldots, k$ **do**

         // Weigh points assigned to cluster

6          $\forall i \in [n] : a_i^{(t)} = j, w_{ij} \leftarrow - \left\| x_i - c_j^{(t)} \right\|^2$

         // Normalize weights

7          $\{\hat{w}_{ij}, i \in [n], a_i^{(t)} = j\} \leftarrow \mathrm{NAct}(\{w_{ij}, i \in [n], a_i^{(t)} = j\})$

         // Compute weighted center

8          $c_j^{(t+1)} \leftarrow \sum_{i:a_i^{(t)}=j} \hat{w}_{ij} x_i$

9   **return** $\{c_j^{(T)}, j \in [k]\}$

---

We now compute $X^{(t+1)}$ as:

$$
X^{(t+1)} - X^{(t+1/2)} = \mathcal{A}_{\langle,\rangle}^I \left( X^{(t+1/2)}, \tilde{C}^{(t)}, I_{d'+2k}^{:d'+k}, \begin{bmatrix} \pi_{d'}^{d'} & 0_{d' \times 2k} \\ 0_{2k \times d'} & \delta I_{2k} \end{bmatrix}, I_{d'+2k}^{d'+k+1:} \right)
$$

$$
= I_{d'+2k}^{d'+k+1:} \begin{bmatrix} \ell(C^{(t)}) \\ I_k \\ I_k \end{bmatrix} \sigma_I \left( \begin{bmatrix} \ell(C^{(t)}) \\ I_k \\ I_k \end{bmatrix}^\top I_{d'+2k}^{:d'+k} \begin{bmatrix} \pi_{d'}^{d'} & 0_{d' \times 2k} \\ 0_{2k \times d'} & \delta I_{2k} \end{bmatrix} \begin{bmatrix} \ell(X) \\ Y(C^{(t)}, X) \\ 0_{k \times n} \end{bmatrix} \right)
$$

$$
= \begin{bmatrix} 0_{k \times k} \\ 0_{k \times k} \\ I_k \end{bmatrix} \sigma_I \left( \begin{bmatrix} \ell(C^{(t)}) \\ \delta I_k \\ 0_{k \times k} \end{bmatrix}^\top \begin{bmatrix} \pi_{d'}^{d'} \ell(X) \\ Y(C^{(t)}, X) \\ 0_{k \times n} \end{bmatrix} \right)
$$

$$
= \begin{bmatrix} 0_{k \times k} \\ 0_{k \times k} \\ I_k \end{bmatrix} \sigma_I \left( \ell(C^{(t)})^\top \pi_{d'}^{d'} \ell(X) + \delta Y(C^{(t)}, X) \right)
$$

$$
= \begin{bmatrix} 0_{k \times k} \\ 0_{k \times k} \\ \mathrm{DP}(C^{(t)}, X) + \delta Y(C^{(t)}, X) \end{bmatrix},
$$

so that $X^{(t+1)} = X^{(t+1/2)} + \begin{bmatrix} 0_{k \times k} \\ 0_{k \times k} \\ \mathrm{DP}(C^{(t)}, X) + \delta Y(C^{(t)}, X) \end{bmatrix}$, yielding $X^{(t+1)}$ as described in Equation (37).

To compute $\tilde{C}^{(t+1)}$, we first compute $\tilde{C}^{(t+1/2)}$ as in Equation (38),

$$
\tilde{C}^{(t+1/2)} - \tilde{C}^{(t)}
$$

$$
= \mathcal{A}_{\langle,\rangle}^\xi(\tilde{C}^{(t)}, X^{(t+1)}, I_{d'+2k}^{d'+k+1:}, I_{d'+2k}^{d'+k+1:}, I_{d'+2k}^{:d}) + \mathcal{A}_{\langle,\rangle}(\tilde{C}^{(t)}, \tilde{C}^{(t)}, I_{d'+2k}^{d'+k+1:}, I_{d'+2k}^{d'+k+1:}, -I_{d'+2k}^{:d+1})
$$

$$
= I_{d'+2k}^{:d} X^{(t+1)} \sigma_\xi \left( (X^{(t+1)})^\top I_{d'+2k}^{d'+k+1:} I_{d'+2k}^{d'+k+1:} \tilde{C}^{(t)} \right) - I_{d'+2k}^{:d+1} \tilde{C}^{(t+1)} \sigma \left( (\tilde{C}^{(t)})^\top I_{d'+2k}^{d'+k+1:} I_{d'+2k}^{d'+k+1:} \tilde{C}^{(t)} \right)
$$

$$
= \begin{bmatrix} X \\ 0_{(2k+1) \times n} \end{bmatrix} \sigma_\xi((\hat{Y}^{(t+1)})^\top) - \begin{bmatrix} C^{(t)} \\ \tilde{C}_{d+1,*} \\ 0_{(2k+1) \times k} \end{bmatrix} \sigma(I_k)
$$

$$
= \begin{bmatrix} X \sigma_\xi((\hat{Y}^{(t+1)})^\top) - C^{(t)} \\ -\tilde{C}_{d+1,*} \\ 0_{(2k+1) \times k} \end{bmatrix}.
$$

So we have

$$
\tilde{C}^{(t+1/2)} = \tilde{C}^{(t)} + \begin{bmatrix} X \sigma_\xi((\hat{Y}^{(t+1)})^\top) - C^{(t)} \\ -\tilde{C}_{d+1,*} \\ 0_{(2k+1) \times k} \end{bmatrix} = \begin{bmatrix} X \sigma_\xi((\hat{Y}^{(t+1)})^\top) \\ 0_{(2k+2) \times k} \end{bmatrix},
$$

as in (37). It remains to compute the squared norms of the new cluster centers, which can be done just as for Appendix F.

This transformer architecture can be also be expressed as an algorithm as shown in Algorithm 5. After the standard cluster assignment step (line 5), each point $i$ assigned to a cluster $j$ is weighed based on its distance to the current center $c_j^{(t)}$ (line

7), and these weights are normalized in line 8 using some form of a normalizing activation (denoted by "NAct") such as soft-max or sparse-max (Martins & Astudillo, 2016). Then the new cluster center is $c_j^{(t+1)}$ is obtained by computing a weighted mean of the points assigned to the cluster.

This algorithm is different from the standard Lloyd's algorithm in that the cluster center is not the exact mean of the assigned points, but rather a weighted mean, with points farther away from the previous center being weighed down. This reduces the effect of outliers assigned to the cluster, thereby making the algorithm more robust than standard Lloyd's (Algorithm 1). This is different from the trimmed $k$-means algorithm (Algorithm 4) since trimmed $k$-means uses a threshold to remove the outliers completely before computing the unweighted mean of the remaining assigned points. This makes the algorithm susceptible to the choice of the threshold, and can be significantly influenced by outliers if the threshold is not appropriately selected (and thus, the outliers are not appropriately removed from consideration). In contrast, this new robust algorithm (Algorithm 5) weighs points assigned to the cluster, and thus will weigh an outlier low as it would be far away from its closest cluster center.

While the soft $k$-means (Algorithm 2) also makes use of weighted means, there is a significant difference between Algorithm 5 and soft $k$-means — while soft $k$-means computes a weighted mean of all the points $\{x_i, i \in [n]\}$, Algorithm 5 computes a weighted mean of points assigned to the cluster $\{x_i, i \in [n], a_i^{(t)} = j\}$, thereby retaining the discrete nature of the standard $k$-means clustering problem.

In our transformer architecture, the desired behavior is obtained by weighing all points for a given cluster $j \in [n]$ based on the negative squared Euclidean distance and a large additive offset $\delta$ for points assigned to this cluster. For a large enough $\delta$, the points not assigned to this cluster will have zero normalized weights, while the normalized weights for the points assigned to the cluster will solely depend on their negative squared Euclidean distance (with no depedence on $\delta$). This is because normalized activations such as soft-max and sparse-max are translation invariant — that is, $\text{NAct}(\{p_i, i \in [m]\}) = \text{NAct}(\{p_i + \delta, i \in [m]\})$ for some $m$ unnormalized weights $\{p_i, i \in [m]\}$ and an additive translation/offset $\delta$.

**Empirical evaluation.** We present some preliminary results here. However, we qualify that evaluation of robust clustering typically considers datasets with outliers. In real datasets, the outliers (just like the clusters) are unknown. Thus, various evaluations rely on synthetic datasets. We present some preliminary results in Table 4, identifying situations where they achieve better $k$-means objective than Lloyd's. Across all settings, Algorithm 4 outperforms Lloyd's (*italics* in Table 4) on 4/10 datasets, while the new Algorithm 5 does so for 6/10 datasets, demonstrating competitive behavior.

*Table 4.* We compare Algorithm 4 (trimmed $k$-means) vs Algorithm 5 (robust $k$-means) on 10 OpenML datasets (Bischl et al., 2025) with $k = 10$ for varying degrees of robustness ($\tau$ in Algorithm 4 and the nonlinear activation inverse temperature $\gamma$ in Algorithm 5). We report the average log $k$-means objective aggregated across 10 trials (*lower is better*). Cases where either of the algorithms outperform Lloyd's algorithm in terms of the $k$-means objective are *emphasized*.

| Dataset | Initial obj. | Lloyd's | Alg 4 $\tau = 0.95$ | — $\tau = 0.99$ | Alg 5 $\gamma = 0.01$ | — $\gamma = 0.001$ |
|---|---|---|---|---|---|---|
| GermanCredit | 9.1393 | 8.6342 | 8.6345 | *8.6339* | 8.6364 | *8.6333* |
| twonorm | 8.1141 | 7.6126 | 7.6130 | 7.6127 | 7.6129 | 7.6127 |
| pc3 | 6.2494 | 5.6776 | 5.7997 | 5.6823 | 5.6802 | 5.6777 |
| ada-agnostic | 9.6838 | 9.1439 | 9.1448 | 9.1440 | 9.1882 | *9.1428* |
| ringnorm | 7.7387 | 7.3020 | 7.3023 | *7.3019* | 7.3021 | *7.3020* |
| pc4 | 6.3693 | 5.8479 | 5.8558 | 5.8493 | *5.8395* | *5.8476* |
| pendigits | 8.7020 | 8.0354 | *8.0313* | *8.0265* | 8.1473 | *8.0279* |
| sylvine | 8.4709 | 7.9196 | 7.9268 | 7.9209 | 7.9316 | 7.9249 |
| optdigits | 10.269 | 9.6075 | 9.6092 | 9.6079 | 9.7229 | 9.6075 |
| mfeat-karhunen | 8.2904 | 7.6923 | 7.6937 | *7.6922* | *7.6911* | 7.6924 |

### I.2. Randomized medoids clustering

A closely related clustering problem is that of Euclidean $k$-medoids, defined as:

$$\min_{c_1,\dots,c_k \in X} \sum_{i=1}^{n} \min_{j=1,\dots,k} \|x_i - c_j\|^2, \tag{38}$$

where it differs from the Euclidean $k$-means clustering problem in Equation (1) in that it requires the cluster centers $c_j, j = 1, \dots, k$ to be actual points in the set $X$. A critical difference between the Lloyd's style alternating optimization

---

**Algorithm 6** Randomized $k$-medoids

---

1  Input: Instances $S = \{x_i \in \mathbb{R}^d, i \in [n]\}$
2  Input: Initial centers $\{c_j^{(0)} \in S, j \in [k]\}$
3  Input: Sampling gap $\delta > 0$
4  **for** $t = 1, \ldots, T$ **do**
        // Assign points to clusters
5      $\forall i \in [n], \ a_i^{(t)} \leftarrow \arg\min_{j \in [k]} \left\| x_i - c_j^{(t-1)} \right\|^2$
        // Sample cluster medoid
6      $\forall j \in [k], \ c_j^{(t)} \sim \mathsf{Cat}(\{w_{ij}, i \in [n]\})$
7      where $w_{ij} = \dfrac{\exp\left(-\left\| x_i - c_j^{(t-1)} \right\|^2 - 2\delta \mathbb{I}(a_i^{(t)} \neq j)\right)}{\sum_{i'=1}^{n} \exp\left(-\left\| x_{i'} - c_j^{(t-1)} \right\|^2 - 2\delta \mathbb{I}(a_{i'}^{(t)} \neq j)\right)}$

8  **return** $\{c_j^{(T)}, j \in [k]\}$

---

and other existing $k$-medoids algorithms (such as PAM (Kaufman, 1990)) is that, while the alternating algorithm selects candidates for a cluster medoid update from within the cluster, PAM allows the selection of candidates for cluster medoid update from outside the cluster. We will instantiate this behaviour in our transformer architecture as follows:

Continuing with the transformer updates as considered in Theorem E.1

$$X^{(t+1)} \leftarrow X^{(t)} + \mathcal{A}_2(X^{(t)}, \tilde{C}^{(t)}, Q_1, K_1, V_1) + \mathcal{A}_2(X^{(t)}, X^{(t)}, Q_2, K_2, V_2)$$
$$\tilde{C}^{(t+1)} \leftarrow \tilde{C}^{(t)} + \mathcal{A}_2^{\xi}(\tilde{C}^{(t)}, X^{(t+1)}, Q_3, K_3, V_3) + \mathcal{A}_{\langle, \rangle}(\tilde{C}^{(t)}, \tilde{C}^{(t)}, Q_4, K_4, V_4), \tag{39}$$

where we will discuss the cross-attention activation $\xi$ in the sequel, and the parameters $\{(Q_\mu, K_\mu, V_\mu), \mu = 1, 2, 4\}$ for the encoder side cross and self-attention, and the decoder side self-attention remain as in Theorem E.1 (that is, $K_1 = Q_1 = K_2 = Q_2 = I_{d+k}^{:d}$, $V_1 = -V_2 = I_{d+k}^{d+1:}$ and $Q_4 = K_4 = I_{d+k}^{d+1:}$, $V_4 = -I_{d+k}^{:d}$), and the $(Q_3, K_3, V_3)$ parameters in the decoder cross-attention for the cluster center update are set as in Appendix H (that is, $V_3 = I_{d+k}^{:d}$, $Q_3 = K_3 = I_{d+k}^{:d} + \delta I_{d+k}^{d+1:}$, where $\delta$ is a positive constant).

Given this parameter setting, the pre-activation attention scores in decoder cross-attention are given by $\mathsf{DP}(K_3 X^{(t+1)}, Q_3 \tilde{C}^{(t)}) \in \mathbb{R}^{n \times k}$ where the $(i, j)$-th entry is given as

$$\mathsf{DP}(K_3 X^{(t+1)}, Q_3 \tilde{C}^{(t)})^{ij} = \begin{cases} -\left\| x_i - c_j^{(t)} \right\|^2 & \text{if } y_i^{(t+1)} = e_j \text{ (that is, if } x_i \text{ assigned to } j^{\text{th}} \text{ cluster)} \\ -\left\| x_i - c_j^{(t)} \right\|^2 - 2\delta & \text{if } y_i^{(t+1)} \neq e_j. \end{cases} \tag{40}$$

Thus, for the $j$-th cluster (the $j$-th query), all the instances (keys) assigned to it (that is, $y_i^{(t+1)} = e_j$) get a score corresponding to their negative squared Euclidean distance $\left\| c_j^{(t)} - x_i \right\|^2$, while instances (keys) not assigned to this cluster ($y_i^{(t+1)} \neq e_j$) get their scores (the negative squared Euclidean distances) shifted by $2\delta$. This is very similar to the pre-activation attention score matrix used in Appendix H for the trimmed $k$-means algorithm; the main difference is that we do not require $\delta$ to be a large positive value here. Instead, the positive constant $\delta \geq 0$ serves as a user-specified parameter that allows us to create a separation between points assigned to the cluster, and points assigned to other clusters.

If we treat these pre-activation scores $\mathsf{DP}(K_3 X^{(t+1)}, Q_3 \tilde{C}^{(t)})^{ij}$ as the logarithm of the unnormalized probability of sampling point $i$ for cluster $j$, we can utilize the Gumbel-soft-max activation (Jang et al., 2017) to sample a point $i \in [n]$ from this categorical distribution for each of the cluster $j \in [k]$. This is performed as follows for any cluster $j \in [k]$ given a Gumbel-soft-max temperature parameter $\lambda > 0$:

• Obtain $n$ samples $g_{ij} \sim \text{Gumbel}(0, 1)$ for $i \in [n]$.
• Compute the $i$-th entry of the $n$-dimensional vector $p_j \in \mathbb{R}^n$ as

$$p_j^i = \frac{\exp((\mathsf{DP}(K_3 X^{(t+1)}, Q_3 \tilde{C}^{(t)})^{ij} + g_{ij})/\lambda)}{\sum_{i' \in [n]} \exp((\mathsf{DP}(K_3 X^{(t+1)}, Q_3 \tilde{C}^{(t)})^{i'j} + g_{i'j})/\lambda)} \tag{41}$$

for each $i \in [n]$, giving us a $n$ dimensional vector $p_j$ from the $(n-1)$-dimensional simplex.

As the temperature $\lambda \to 0$, $p_j$ becomes a one-hot vector, and $X p_j$ serves as a sample from the categorical distribution defined in Algorithm 6, line 6-7 for the $j$-th cluster.

Thus, with $\xi = \text{GSM}^\lambda$ as the Gumbel-soft-max activation (with the temperature $\lambda \to 0$), applied column-wise to the $\text{DP}(K_3 X^{(t+1)}, Q_3 \tilde{C}^{(t)})$ matrix, we get the Gumbel-soft-max attention or GSMA. Each column in the resulting matrix $\text{GSM}^0(\text{DP}(K_3 X^{(t+1)}, Q_3 \tilde{C}^{(t)}))$ corresponds to a sample (in the form of a one-hot $n$-dimensional vector) from the per-cluster categorical distribution described in Algorithm 6, lines 6-7, with the positive parameter $\delta$ controlling how much we are allowed to sample from outside of the cluster. Thus, the cross-attention update for the cluster medoids is given as follows:

$$
\mathcal{A}_2^\xi(\tilde{C}^{(t)}, X^{(t+1)}, Q_3, K_3, V_3) = V_3 X^{(t+1)} \text{GSM}^0(\text{DP}(K_3 X^{(t+1)}, Q_3 \tilde{C}^{(t)}))
$$

$$
= \begin{bmatrix} X \\ 0_{k \times n} \end{bmatrix} \text{GSM}^0(\text{DP}(K_3 X^{(t+1)}, Q_3 \tilde{C}^{(t)})), = \begin{bmatrix} X \\ 0_{k \times n} \end{bmatrix} [p_1; p_2; \ldots; p_k] = \begin{bmatrix} C^{(t+1)} \\ 0_{k \times k} \end{bmatrix},
$$

thus each updated cluster mediod candidate is an actual point, sampled as per the specified categorical distribution. The self-attention update along with the residual connection handles the resetting of the cluster medoid candidates as discussed previously for other clustering algorithms. This transformer architecture is thus equivalent to a novel randomized $k$-medoids algorithm presented in Algorithm 6.

Note that, if it is possible to modify the medoid sampling distribution in our Gumbel-softmax in the forward pass (i.e., as in Tiwari et al. (2020)), we obtain a competitive $k$-medoids algorithm instantiated within our transformer architecture. However, we leave this investigation for future work.

**Empirical evaluation.** Here we evaluate Algorithm 6, varying values of $\delta \in \{0, 0.01, 1, 100, 10^4, 10^8, 10^{15}\}$, and present the results in Table 5. The best objectives (*emphasized* in Table 5) are found by values of $\delta > 0$, thereby lowering the selection probability from outside the current cluster, but not too large $\delta$ which enforces within cluster sampling. The results highlight that it is critical to balance (i) the ability to sample from outside the current cluster, with (ii) exploiting strong centroid candidates within the current cluster. This tradeoff can be tuned with $\delta$ in Algorithm 6, which is obtained via architectural modifications to the base $k$-means transformer.

*Table 5.* We evaluate Algorithm 6 on 10 OpenML (Bischl et al., 2025) datasets with $k = 10$ for varying values of the sampling gap hyperparameter $\delta > 0$. We report the average log $k$-medoids objective aggregated across 10 trials (*lower is better*). The best objectives for each dataset are *emphasized below*.

| Dataset | Initial obj | $\delta = 0$ | $\delta = 0.01$ | $\delta = 1$ | $\delta = 10^2$ | $\delta = 10^4$ | $\delta = 10^8$ | $\delta = 10^{15}$ |
|---|---|---|---|---|---|---|---|---|
| GermanCredit | 9.1393 | 9.1021 | *9.0945* | 9.0993 | 9.0950 | 9.1029 | 9.1027 | 9.1147 |
| twonorm | 8.1141 | 7.9807 | 7.9807 | *7.9741* | 7.9910 | 7.9902 | 8.0070 | 8.0215 |
| pc3 | 6.2322 | 6.0533 | *6.0201* | 6.0489 | 6.0278 | 6.0264 | 6.0438 | 6.0217 |
| ada-agnostic | 9.7136 | 9.6223 | 9.6057 | *9.6023* | 9.6128 | 9.6313 | 9.6317 | 9.6341 |
| ringnorm | 7.7808 | *7.6342* | 7.6393 | 7.6428 | 7.6603 | 7.6643 | 7.6691 | 7.6691 |
| pc4 | 6.4307 | 6.2480 | 6.2309 | 6.2239 | *6.2102* | 6.2113 | 6.2312 | 6.2412 |
| pendigits | 8.7588 | 8.5852 | 8.5812 | 8.5747 | 8.4872 | 8.4844 | *8.4672* | 8.4799 |
| sylvine | 8.4547 | 8.3895 | 8.3857 | 8.3861 | *8.3542* | 8.3788 | 8.3804 | 8.3580 |
| optdigits | 10.268 | 10.172 | 10.173 | 10.170 | 10.172 | *10.137* | *10.137* | 10.140 |
| mfeat-karhunen | 8.2856 | 8.2412 | 8.2423 | 8.2392 | 8.2374 | 8.2312 | *8.2295* | 8.2424 |

