# OpenReview forum: "Transformer Circuits Can Realize Clustering Algorithms"
_ICML.cc/2026/Conference — ICML 2026 spotlight_

### Official Review · Reviewer_QGDE · 2026-02-25

**Soundness:** 3
**Presentation:** 3
**Significance:** 3
**Originality:** 3
**Overall Recommendation:** 4
**Confidence:** 4

**Summary:**

This paper investigates whether the Transformer architecture can implement exact algorithmic computations beyond its role as a statistical sequence model. The authors design a "k-means Transformer" and prove that its standard components (attention, residuals, feed-forward modules) can exactly execute Lloyd’s algorithm for k-means clustering.

**Compliance With Llm Reviewing Policy:**

Affirmed.

**Final Justification:**

This paper establishes a clear bridge between classical discrete optimization and modern neural architectures. It provides a "white-box" view of how attention mechanisms map to Lloyd's algorithm in k-means. During the author's rebuttal, they effectively addressed my concerns. I believe this is a well-researched and experimentally rich article.

**Key Questions For Authors:**

Please see weakness.

**Limitations:**

yes

**Strengths And Weaknesses:**

Strengths:
- This paper establishes a clear bridge between classical discrete optimization and modern neural architectures.
- It provides a "white-box" view of how attention mechanisms map to Lloyd's algorithm in k-means.
- It demonstrates flexibility by deriving various clustering algorithms through component modifications.

Weaknesses:
- Numerical Stability and Validity of Finite γ (Figure 3):
The theoretical proof relies on limit soft-max attention. In Figure 3, the authors use γ=10,000 to simulate finite temperature behavior. However, computing soft-max with such a large number poses a high risk of numerical overflow during exponentiation in soft-max, and causes the result to collapse into the hard limit case. Figure 3 may fails to demonstrate robustness at finite γ.
- Equation 2 introduces placeholders I_k and Y, which are not present in the standard Lloyd’s algorithm. Please clarify the exact mathematical mapping between this modified forward pass and the classic Lloyd iteration. Furthermore, is the observed performance gain primarily due to these extra computational terms rather than the architecture's ability to simulate Lloyd’s algorithm?
- Equation 3 introduces learnable projection matrices Q and K for the attention mechanism. Classic Lloyd’s algorithm computes Euclidean distances directly without projecting data into a new subspace. Does this introduce an extra projection step, fundamentally changing the algorithm to "project-then-cluster"? And please provide a rigorous theoretical comparison of time and space complexity between the k-means Transformer and classic Lloyd’s algorithm.
- Appendix B details the training process but raises concerns regarding optimization and practicality. Deep learning models heavily depend on Stochastic Gradient Descent (SGD) and are highly sensitive to hyperparameters such as learning rate, weight decay, and batch size. In contrast, a key advantage of traditional algorithms like Lloyd’s is their simplicity and parameter-free nature (no learning rate tuning required). The paper lacks a discussion on how these critical hyperparameters impact the k-means Transformer across different data distributions.

---

> ### Author Rebuttal · Authors · 2026-03-31
>
> > ### QGDE-1 *Regarding limiting softmax attention, finite $\gamma$ and numerical stability*
>
> We now include an additional analysis in rebuttal to Reviewers that provides a bounding approximation in transformers with finite $\gamma$; see discussion in **DT5H-6**.
> Additionally, we point the reviewer to Figure 5 in Appendix A where we study the finite values of $\gamma$ necessary to match Lloyd's algorithm empirically.
> The use for $\gamma = 10^4$ was only for the $d=2$ dimensional clustering problem in Figure 3 presented for visualization purposes. As shown in Figure 5, $\gamma \in [10, 100]$ suffices for clustering problem of general sizes. Furthermore, the transformer trained on clustering tasks uses $\gamma = 1$.
>
> Finally, we note that softmax is usually implemented to avoid numerical overflow by leveraging its translation invariance --- the largest entry in the softmax input is often subtracted from the remaining entries, ensuring that the largest inputs to the $\exp(\cdot)$ is zero (and remaining are $<0$).
>
> ---
> > ### OGDE-2 *Mapping token embeddings and transformer parameters to Lloyd's algorithm*
>
> The additional parameters (e.g., $I_k$ and $Y$) arise from the transformer’s token‑embedding formulation. Their exact correspondence is given in Section 2.2 and Theorem 2.1. These parameters are also implicit in standard Lloyd’s algorithm: $I_k$ stores cluster indices and $Y$ stores per‑point cluster assignments, both in one‑hot form.
>
> With the construction in Theorem 2.1, the kmeans transformer behaves exactly like Lloyd's algorithm. The performance gain in section 2.3 is seen only when the transformer is trained on a distribution of clustering tasks, and evaluated on clustering tasks from the same task distribution (**see FZsM-1 for new experiments on different distributions, as well as LKjr-1 for new experiments on real-world data**). When the transformer outperforms Lloyd's algorithm, it does not simulate Lloyd's algorithm exactly but rather executes a different algorithm that is learned during the training phase (see discussion in **FZsM-2**).
>
> ---
> > ### QGDE-3 *Distinguishing Euclidean clustering with Lloyd's from "project-then-cluster" in the kmeans transformer*
>
> It is important to distinguish between the constructive parameterization of a transformer that *exactly* implements Lloyd’s algorithm and the learnable transformer trained to minimize the $k$-means objective. In the constructive case, the model does not project the data into a new subspace: $Q$ and $K$ are identity maps, and cluster assignments and center updates are governed by pairwise Euclidean distances in the original space.
>
> In contrast, in the learnable model $Q$ and $K$ are trained, allowing the model to cluster in learned subspaces. The trained model is evaluated on 320 distinct clustering problems drawn from different data distributions (**see Fig 4 and new experiments in FZsM-1**), so it does not learn a single shared subspace, but instead can extract task-specific subspaces that enable faster reduction of the $k$-means objective.
>
> ---
> > ### QGDE-4 *Computational cost of kmeans transformer compared to Lloyd's kmeans algorithm*
>
> Due to space constraints, please refer to response **LJkr-4** for discussion on computational costs.
>
> ---
> > ### QGDE-5 *Sensitivity to hyperparameter in training transformers to cluster*
>
> While deep learning models can be sensitive to hyperparameters, the k‑means transformer studied here is a single‑layer (non‑deep) model, which simplifies training. We use Adam with default PyTorch settings (no weight decay) and tune only the initial learning rate. We use 0.01; when training to convergence, downstream performance is largely insensitive to this choice except for unrealistically large values (e.g., 1). Appendix B illustrates that meaningful clustering algorithms can be recovered via standard gradient‑based optimization and can achieve better clustering when training and test distributions align.
>
> Our goal is not to propose a simpler or more robust alternative to Lloyd’s algorithm, but to provide a constructive parameterization showing that a transformer can exactly implement Lloyd’s k‑means updates. This aligns with prior work on in‑context learning, which studies whether transformers can represent well‑defined algorithms such as linear regression. As in that literature, such constructions are not expected to be practically competitive with classical algorithms due to architectural and computational overhead. The learnability results are therefore intended to be suggestive: gradient‑based training can recover algorithmic structure, potentially enabling learned variants better matched to specific data distributions.
>
> Our work should therefore be viewed as a constructive, explanatory contribution rather than a practical alternative to classical k‑means, clarifying how modern sequence models can represent and learn algorithmic structure and motivating learned approaches to clustering.

---

> > ### Author Rebuttal · Reviewer_QGDE · 2026-04-01
> >
> > The questions I raised have been fully resolved. I decide to raise my overall recommendation to 4: Weak accept.

---

### Official Review · Reviewer_FZsM · 2026-03-13

**Soundness:** 3
**Presentation:** 3
**Significance:** 3
**Originality:** 3
**Overall Recommendation:** 5
**Confidence:** 4

**Summary:**

In this paper, the authors show that the transformer architecture can be parameterized to implement the Lloyd's algorithm for k-means clustering, as well as be trained to perform k-means clustering. In addition, the authors show that existing and new variants of clustering algorithms can be obtained by adding interpretable changes the transformer architecture. Empirically, for clustering a mixture of Gaussians, the authors show that the standard transformer architecture can be trained to outperform the Lloyd's algorithm.

**Compliance With Llm Reviewing Policy:**

Affirmed.

**Final Justification:**

I have carefully read the authors' response to each reviewer. The additional experiments on real-world data look promising. It addresses my main initial concern on the empirical setting being too synthetic. I strongly recommend the authors include all these additions to the revised paper.

**Key Questions For Authors:**

- In Corollary 2.1, does the size of k-means transformer mean the number of nonzero parameters? You should be explicit about what "size" measures here.
- Why does the learned transformer model perform better than the Lloyd's algorithm?
- How well do the two new clustering algorithms in Appendix I perform empirically?
- This is not discussed in Limitations, but can the proposed architecture be trained for a specific data dimension d and applied to a different data dimension d'? (I think not.)

**Limitations:**

yes

**Strengths And Weaknesses:**

Strengths:
- The paper is really well-written. The notations are consistent, the claims are clear and well-supported. In term of writing style, it is clearly a pleasant paper to read. Nothing is over claimed, no ambiguities.
- Characterizing the ability of transformers to represent and learn clustering algorithms is a fundamental and interesting problem. It fits nicely into the line of work to study the ability of transformers to execute and learn algorithms.

Neutral:
- The technical analysis is fairly simple and straightforward.

Weaknesses:
- The numerical experiment is limited to clustering a mixture of Gaussians. One the base model is trained, the authors should perform additional experiments and test on potentially out-of-distribution data: (1) a mixture of Gaussians generated from different means; (2) distributions other than Gaussian mixtures. It is okay if the learned model does not generalize well beyond the training data, but it will show more clearly the limit of this approach.
- It is not explained well why the learned transformer model even perform better than the Lloyd’s algorithm. The authors should at least provide some discussion on it. Does the transformer actually learn the Lloyd’s algorithm, or something else?
- The new clustering algorithms (robust k-means and randomized Euclidean k-medoids) are not evaluated empirically.

---

> ### Author Rebuttal · Authors · 2026-03-31
>
> > ### FZsM-1 *Evaluate trained transformer on varied task distributions*
>
> We first describe the clustering tasks used for training and evaluation (also in Appendix B.1). We use a hierarchical generative process. Starting from a distribution of tasks $p$, a specific $t$-th task is generated by: (i) sampling the $k$ means, covariances, and weights $\lbrace (\mu_j^{(t)}, \sigma_j^{(t)}, w_j^{(t)} ) \sim p, j \in [k] \rbrace$, (ii) defining a $d$-dimensional Gaussian mixture model or GMM as $\mathcal G^{(t)} \triangleq \sum_j w_j^{(t)} \mathcal N(\mu_j^{(t)}, \sigma_j^{(t)})$, (iii) generating $n$ samples from this GMM $\lbrace x_i \sim \mathcal G^{(t)}, i \in [n] \rbrace$ as the set of points to be clustered, and (iv) randomly sampling $k$ of these generated samples as the initial centers.
>
> Thus, each of our clustering tasks are already generated from a different GMM, and the test clustering tasks used to evaluate the generalization of the learned transformer in Figure 4(a)-(c) are distinct from the ones used for training. We also consider the scenario where the transformer is trained on clustering tasks with $n = 512$ points (in $d = 32$ dimensions), and evaluated on clustering tasks with $n \not= 512$ in Figure 4(d), again with each of the clustering tasks generated from different GMMs.
>
> Here we consider the kmeans transformer trained on clustering tasks generated from GMMs (as described above), and evaluate it on clustering tasks generated from other mixture distributions (Cauchy, Laplace, Gumbel, LogNormal). We compare its performance to Lloyd's and report the average log kmeans objective after 20 iterations (over 32 test tasks). We see that for all distribution mixtures, the trained transformer is able to improve the kmeans objective over the initial centers, and outperforms Lloyd's algorithm for all but the tasks generated from mixtures of Cauchy distributions.
>
> |Family   |Init |Lloyd|Trained trf|
> |---------|-----|-----|-----|
> |Gaussian |6.337|5.250|5.017|
> |Cauchy   |3.411|3.254|3.360|
> |Gumbel   |6.243|5.246|5.064|
> |Laplace  |6.043|5.090|4.956|
> |Lognormal|6.299|5.249|4.996|
>
> ---
> > ### FZsM-2 *Why trained transformer outperforms Lloyd's*
>
> At a conceptual level, we train the transformer over a distribution of clustering tasks, and learning from that training data enables the model to identify better updates *for this specific distribution*, but without any guarantees for clustering tasks that do not come from this distribution.
>
> However, understanding why the learned transformer performs better, and the conditions under which the transformer will outperform Lloyd's would require significant technical analysis regarding the distribution of clustering tasks, among various other factors. Our goal is to encourage research showing that transformers can realize precise, known clustering algorithms, paralleling results by Von Oswald et al. (2023) and Ayurek et al. (2023) for linear regression via gradient descent. As with that line of work, which has since spawned substantial literature (Ahn, et al. 2023; Huang et al. 2023; Zhang et al. 2023; Lu et al. 2025; Letey et al., 2025 and many others), we expect understanding when and why transformers succeed to require similarly extensive follow-up.
>
> ##### Lu, Y., et al. "Asymptotic theory of in-context learning by linear attention." PNAS 2025.
> ##### Letey, Mary I., et al. "Pretrain-Test Task Alignment Governs Generalization in In-Context Learning." arXiv 2025.
>
> ---
> > ### FZsM-3 *Evaluate Alg 5 (robust kmeans) and Alg 6 (randomized kmedoids)*
>
> Please refer to the additional evaluations presented in **DT5H-4** and **DT5H-5**.
>
> ---
> > ### FZsM-4 *Regarding number of parameters in Corollary 2.1*
>
> We will clarify explicitly that we consider the number of non-zero elements in the model parameters as all parameters specified in Theorem 2.1 correspond to *diagonal* $((d+k)\times (d+k))$ matrices, which require only $(d+k)$ parameters each to store.
>
> ---
> > ### FZsM-5 *Train model for specific dimensionality $d$ and apply to data with $d' \not= d$ dimensions*
>
> One should not expect a model trained on $d$ dimensions to be applicable to $d'$ dimensional data, and this is the case not only with transformers and in-context learning, but across machine learning.
>
> However, if we have a kmeans transformer with a large $d$, then it can be *applied to* clustering tasks with a different $d' \leq d$ by zero-padding the $d'$-dimensional data to make it $d$-dimensional since Euclidean distances are unaffected by the zero-padded dimensions. If $d' > d$, this would not be applicable directly unless we are able to do some form of distance preserving dimensionality reduction.
>
> The construction in Theorem 2.1 will exactly execute Lloyd's algorithm on this zero-padded data. Whether the trained transformer will successfully cluster for $d' \leq d$ will depend on how the distribution of the $d'$-dimensional clustering tasks aligns with the training distribution.

---

> > ### Author Rebuttal · Reviewer_FZsM · 2026-04-02
> >
> > I'd like to thank the authors for their response. I also read their responses to other reviewers, and the additional results from numerical experiments on benchmarks are really promising. Overall, I think this is a good work. I will increase my score from 4 to 5 to reflect on that.

---

### Official Review · Reviewer_LJkr · 2026-03-13

**Soundness:** 3
**Presentation:** 2
**Significance:** 4
**Originality:** 3
**Overall Recommendation:** 5
**Confidence:** 3

**Summary:**

The paper explores how transformers can be constructed to learn exact algorithmic computations. It focuses on the k-means clustering algorithm and shows, both theoretically and empirically, that this is possible. Furthermore, the paper explores variants of k-means, such as soft, spherical, and trimmed k-means, which arise from modifications to the transformer architecture. The main insight and contribution of the paper is the formalization of k-means as a transformer and the resulting perspective on the possibility of realizing an algorithm within a transformer.

**Compliance With Llm Reviewing Policy:**

Affirmed.

**Final Justification:**

The rebuttal addressed my main concerns and I decided to raise my score from weak accept to accept.

**Key Questions For Authors:**

1. Can the authors provide empirical results on at least a few real-world clustering benchmarks, for example smaller UCI datasets, and compare the transformer-based approach against standard k-means and relevant variants?

2. In which practical settings do the authors envision that a transformer implementation of k-means would be preferable to the original algorithm?

3. How does the computational cost of the proposed transformer-based approach compare to standard k-means in terms of runtime, memory, and scalability?

4. Can the authors include analyses of the learned weights, attention patterns, loss landscapes, or other internal visualizations of the trained transformer to better understand what has been learned?

5. Can the authors clarify whether the observed improvements over standard k-means on synthetic data are robust across different settings, and what properties of the data favor the proposed method?

**Limitations:**

yes

**Strengths And Weaknesses:**

### Soundness
**Strengths:**
- The paper derives the k-means algorithm as a transformer from first principles.
- The synthetic experiments are systematic and show that the proposed algorithm can outperform the original k-means.

**Weaknesses / Concerns:**
- No experiments on real data are provided. Not even smaller UCI clustering benchmarks are considered.
- I am missing some analysis of the learned weights, attention patterns, and other visualizations that would help explore the trained transformer.


### Presentation
**Strengths:**
- The authors provide an elaborate visualization of how k-means can be realized as transformer circuits in Figures 2 and 9.
- The paper starts from first principles and explains each step in detail.
- The discussion positions the approach well as a first concrete implementation of an exact unsupervised algorithm within the transformer architecture, confirming previous existence results.

**Weaknesses / Suggestions:**
- The paper is a very dense read, and I would suggest including some paragraphs with more intuitive explanations and figures.
- While Figures 2 and 9 are appreciated, they are also very dense and difficult to understand. I would recommend that the authors either redesign the figures or divide them into smaller subparts.

### Significance
**Strengths:**
- The authors provide one of the first exact implementations of an algorithm (k-means) within the transformer architecture.
- They show how variations in the architecture can lead to existing k-means variants as well as new ones.
- k-means is one of the fundamental algorithms and is related to many other areas of unsupervised learning, such as matrix factorization, and it serves as the basis for many other clustering algorithms. This work might open the door to implementing other related algorithms following the outlined approach.

**Weaknesses / Limitations:**
While the theoretical investigation is very interesting and I understand that this paper is more about what transformer can do, I see several limitations on the applied side:
- From a practical and more applied point of view, I wonder when someone would want to use a transformer version of k-means in practice.
- The lack of real-world experiments makes it difficult to judge the practical relevance.
- The original k-means algorithm often requires hundreds of steps to converge, while the transformer version seems to converge faster, I wonder about the computational overhead.


### Originality
**Strengths:**
-  The implementation of k-means as a transformer circuit is very interesting and to my knowledge they are the first to show the implementation of an unsupervised algorithm.

**Weaknesses:**
- They build upon existing theoretical results and investigations on the clustering behaviour of attention
- Akyürek et al. (2023) showed similar results for linear regression

---

> ### Author Rebuttal · Authors · 2026-03-31
>
> > ### LJkr-1 *Evaluate trained transformer on benchmarks*
>
> We first clarify our claim that the trained model is expected to outperform Lloyd's algorithm only on clustering tasks from the same task distribution as those used for training (please see **FZsM-1**) as presented in Figure 4. In existing literature, synthetic data are used to understand the ability of transformers to faithfully express that algorithm. This also allows control over the task distributions, enabling systematic generalization evaluation.
>
> Here we apply the trained k-means transformer to 15 OpenML datasets. This model is trained *solely on synthetic clustering tasks* (see Appendix B.1 for task distribution) with $d = 32, k = 10$. Hence, strong generalization is *expected* on new clustering tasks obtained from the same generative process. We have no such control on the OpenML datasets. We execute Lloyd's and the trained model for 10 iterations, and report the mean (std) log kmeans objective (40 trials). In all cases, the trained model improves the kmeans objective from the initial centers, demonstrating that the model has learned some generalized clustering mechanisms. However, it underperforms Lloyd's by 1-7\% as the real data does not match the training task distribution.
>
> |Dataset|Initial|Lloyd|Trained trf|rel-diff(\%)|
> |-|-|-|-|-|
> |gina-agnostic|8.35(0.14)|7.88(0.13)|7.95(0.13)|0.93|
> |musk|6.98(0.15)|6.38(0.11)|6.45(0.11)|1.09|
> |mfeat-factors|6.96(0.10)|6.33(0.06)|6.41(0.06)|1.22|
> |GermanCredit|8.41(0.11)|7.90(0.10)|8.00(0.10)|1.24|
> |mfeat-fourier|7.01(0.07)|6.47(0.03)|6.55(0.03)|1.25|
> |ionosphere|6.36(0.10)|6.00(0.04)|6.08(0.03)|1.30|
> |mfeat-karhunen|6.83(0.06)|6.28(0.03)|6.38(0.03)|1.53|
> |ada-agnostic|7.68(0.14)|7.15(0.11)|7.26(0.11)|1.58|
> |optdigits|7.73(0.12)|7.13(0.11)|7.27(0.12)|1.85|
> |sonar|5.51(0.06)|4.97(0.06)|5.09(0.08)|2.48|
> |sylva-agnostic|6.67(0.30)|6.21(0.29)|6.39(0.29)|2.89|
> |oil-spill|6.29(0.32)|5.57(0.20)|5.79(0.16)|3.95|
> |ailerons|5.27(0.20)|4.69(0.14)|4.96(0.14)|5.81|
> |pc4|5.82(0.18)|5.29(0.15)|5.61(0.17)|5.99|
> |pc3|5.61(0.20)|5.08(0.12)|5.44(0.19)|7.07|
>
> ---
> > ### LJkr-2 *Interpret weights of the trained model*
>
>  While analyses of learned weights, attention patterns, or loss landscapes can certainly be informative, appropriate reliable mechanistic interpretation of trained transformers is often a substantial research endeavor in its own right (hence, the field of MechInterp). Furthermore, we are unable to share visualizations in the admitted ICML rebuttal format.
>
> ---
> > ### LJkr-3 *Practical utility of transformer implementation and positioning relative to Akyurek et al (2023)*
>
> As the reviewer notes, our goal is to theoretically characterize what algorithmic mechanisms are possible within the transformer architecture, rather than to claim that a trained transformer universally outperforms classical algorithms. This aligns with prior in-context learning literature, which studies whether mechanisms such as linear self-attention can implement algorithms like linear regression in-context (e.g., Akyurek et al. 2023). This line of work primarily aims to understand transformer capabilities, rather than proposing immediately deployable practical methods.
>
> Within this framing, our contribution serves two purposes: (i) we show that simple architectural and circuit-level modifications (e.g., LayerNorm or sparse attention) enable the design of multiple (existing and novel) algorithms; and (ii) as demonstrated in Section 2.3, when clustering tasks are available for training, and downstream tasks follow the same distribution, a transformer can be trained to achieve strong clustering performance on that task family (please refer to **FZsM-1**). In such settings, our results suggest that a trained kmeans transformer provides improved performance.
>
> ---
> > ### LJkr-4 *Computational cost of kmeans transformer compared to Lloyd's*
>
> Each Lloyd's iteration takes $O(dnk)$ time and $O(nk)$ memory. A single transformer layer generally takes $O( d^2(n + k) + d (nk + n^2 + k^2) )$ time and $O(n^2 + k^2 + nk)$ memory. The $O(d^2(n+k))$ term in the runtime is due to the token projections as queries/keys/values; the $O(dnk)$ and $O(d(n^2+k^2))$ terms are from the cross and self-attention.
>
> The $O(d^2)$ term in the runtime becomes $O(d)$ with diagonal $Q, K, V$ projection matrices (as in Theorem 2.1). With sparse self-attention (as in Theorem 2.1 where a token only needs to attend to itself in the self-attention), the $O(d(n^2 + k^2))$ runtime term becomes $O(d(n+k))$. This results in a $O(dnk)$ dominating term in the runtime as in standard Lloyd's; the memory cost also becomes $O(nk)$.
>
> This is standard with transformers implementing algorithms, since the focus is to understand what algorithms transformer can implement in-context rather than to develop the most efficient form of that algorithm.
>
> ---
> > ### LJkr-5 *Robustness of observed improvements of trained transformer*
>
> Please refer to responses **FZsM-1** and **FZsM-2**.

---

> > ### Author Rebuttal · Reviewer_LJkr · 2026-04-04
> >
> > I thank the authors for addressing my questions and concerns. After reading the rebuttals to mine and other reviews, I decided to raise my score.
> >
> > > Regarding the "LJkr-2 Interpret weights of the trained model".
> >
> > I agree that a full mechanistic interpretability study would not be feasible, but some analysis would still be interesting. Also just to note, it is allowed to share images via anonymous links during the ICML reviewing. I do not think that this is a critical missing part in the paper, so the paper can be accepted without it.

---

> > > ### Author Response · Authors · 2026-04-06
> > >
> > > > ### LJkr: [...] some analysis would still be interesting.
> > >
> > > We thank reviewer LJkr for their acknowledgement and support. We apologize for our oversight regarding acceptable ICML author response format. We present two visualizations ([learned weights](https://imgur.com/eKLV5Ph) and [attention maps](https://imgur.com/apQl1pn)) here for the model trained with $d=32$ and $k=10$ clusters:
> > > - [Learned weights](https://imgur.com/eKLV5Ph): We present the learned weights for $\lbrace (Q_\mu, K_\mu, V_\mu, Q_\mu^\top K_\mu ), \mu \in [4]\rbrace$ for the 4 different attention modules in equation (4), and visualize them next to the weights (equation (5)) used in Theorem 2.1 to realize Lloyd's algorithm. We can make the following observations:
> > >   - The learned $Q_\mu^\top K_\mu$ matrices match the dominant diagonal form of the theoretical constructions in 3/4 cases even though the learned  $Q_\mu, K_\mu$ matrices do not look similar to the constructed ones in equation (5). This somewhat aligns with our Remark E.1 in Appendix E (page 22).
> > >   - The learned $Q_3^\top K_3$ for the cross-attention in the cluster center update appears significantly different (more diffused) than the constructed $I_{d+k}^{d+1:}$ in equation (5) used in Theorem 2.1, implying that the learned transformer is utilizing a different cluster center update than Lloyd's algorithm.
> > >   - The learned value matrices $(V_3, V_4)$ in the cluster center update cross and self attention (respectively) aligns with the constructed ones. However, the value matrices for the cluster assignment updates $(V_1, V_2)$ appear different than the constructed ones, especially with the self-attention values matrix $V_2$ having a dominant diagonal form (close to $I_{d+k}^{:d}$) that is orthogonal to the constructed $-I_{d+k}^{d+1:}$.
> > >
> > > - [Attention maps](https://imgur.com/apQl1pn): We present the attention maps (denoted as $\mathcal A(Z, Z', Q, K)$ with projected queries $QZ$ and projected keys $KZ'$) from the learned transformer for a single update with 4 different clustering tasks (plots with prefix `[Lrn]`), and visualize them next to the attention maps that implement Lloyd's algorithm in Theorem 2.1 (plots with prefix `[Thm]`). Note that for the cross-attention maps, the attention maps would be $(512 \times 10)$ or $(10 \times 512)$, making heatmap visualization quite unclear. Thus, we subsample 30 points, and show the cross-attention maps for those 30 points only (leading to more visually clearer $(30 \times 10)$ and $(10 \times 30)$ heatmaps). We can make the following observations:
> > >   - First, we see that all the attention maps with the learned transformer are quite sparse even with a softmax attention with $\gamma = 1$, implying that the transformer learned to utilize sparse attention maps (as in Theorem 2.1) without explicitly requiring the limiting softmax attention.
> > >   - The self-attention maps from the learned transformer for both the updates in equation (4) seem to match the constructed one quite well with a diagonal attention map.
> > >   - The cross-attention maps differ significantly between the learned and the constructed transformer. The learned cluster assignment update attention maps $\mathcal A(X, C, Q_1, K_1)$ visually appear to match the constructed ones better, while the learned cluster center update attention map $\mathcal A(C, X, Q_3, K_3)$ differs significantly from the constructed one.

---

### Official Review · Reviewer_DT5H · 2026-03-16

**Soundness:** 3
**Presentation:** 3
**Significance:** 3
**Originality:** 3
**Overall Recommendation:** 4
**Confidence:** 4

**Summary:**

This paper establishes a formal correspondence between transformer circuit mechanisms and Lloyd’s algorithm for k means clustering.The authors construct a specific encoder decoder transformer architecture - the ‘kmeans transformer’ and prove that with prescribed weights using limiting softmax attention, negative squared euclidean distance attention scores, and residual connections, each layer exactly executes one iteration of Lloyd’s algorithm. A second construction achieves the same result using standard dot product attention augmented with lifted embeddings and a feed forward network with rectified quadratic activation. The model size is O(d+k) smaller than input. The authors then show that training this architecture on a distribution of clustering tasks yields a learned algorithm that outperforms Lloyd’s on new tasks, generalizing across problem sizes. Also, the paper demonstrates that modifying transformer components yields a family of known clustering algorithms and some novel ones, framing the transformer as an ‘algorithmic skeleton’ for clustering.

**Compliance With Llm Reviewing Policy:**

Affirmed.

**Key Questions For Authors:**

Can you evaluate the trained k means transformer on real world clustering benchmarks?
What happens with layer specific weights?
Can you please provide empirical results for the novel robust k-means and randomized kmedoids?
Can you formally bound the approximation error between the kmeans transformer with finite γ and Llyod’s algorithm, as a function of γ and the minimum separation between cluster centers?

**Limitations:**

The paper includes a limitations section, but it should also discuss the assumptions needed for exactness, the gap between the euclidean attention / quadratic-FFN constructions and standard transformer blocks, the attention time cost of the construction, and the synthetic, distribution specific nature of the learning experiments.

**Strengths And Weaknesses:**

Strengths:

The theoretical proofs are well written with explicit parameterizations and clean algebraic derivations.
The training experiments go beyond expressivity to show that the architecture, when trained with γ=1 on a distribution of clustering tasks, learns an algorithm that consistently outperforms Lloyd’s across different problem sizes, and initializations.
The systematic parameter sweep in Appendix A across γ, d, n, k, and scaling factors, including 2D joint parameter heatmaps, provides comprehensive empirical support for the theoretical claims.
The transformer implementing Lloyd’s is smaller than the input O(d(n+k)) and even the output O(dk).

Weaknesses:

The learning experiments are restricted to Gaussian mixtures with known k. There is no evaluation on real world clustering benchmarks.
The training uses a single layer applied iteratively but the paper does not explore multi layer training with different weights per layer.
The comparison is only against Lloyd’s and not kmeans ++, mini-batch kmeans or other modern methods.
The robust kmeans (Algorithm 50 and randomized k-medoids are presented as theoretical constructions with no experimental validation.
The construction shares weights across all T layers, which means the transformer implements the same algorithm at every iteration.

---

> ### Author Rebuttal · Authors · 2026-03-31
>
> > ### DT5H-1 *Evaluate trained transformer on benchmarks*
>
> Please refer to response **LJkr-1** for new experiments.
>
> ---
> > ### DT5H-2 *Layer-specific weights in transformer*
>
> We present a 1-to-1 mapping between a 1-layer transformer and a single Lloyd's iteration, allowing repeated use of this layer to emulate iterative updates. With layer-specific weights, the choice of the maximum number of layers is unclear, as is the procedure for extending the number of iterations beyond the number of layers. While interesting, we consider this a nontrivial question beyond the current scope.
>
> ---
> > ### DT5H-3 *Compare to Lloyd's, kmeans++ or others*
>
> Our goal here is not a state-of-the-art algorithm, but rather to demonstrate that a transformer can exactly realize Lloyd's algorithm, and when trained on a distribution of clustering tasks, can learn to outperform Lloyd's (on that same distribution). We do consider kmeans++, which usually seeds clustering algorithms (including Lloyd's), and we show in Fig 4(c) that the trained model also benefits from kmeans++.
>
> ---
> > ### DT5H-4 *Evaluate Alg 5 (robust kmeans)*
>
> We present some results here. However, we qualify that evaluation of robust clustering typically considers datasets with outliers. In real datasets, the outliers (just like the clusters) are unknown. Thus, various evaluations rely on synthetic datasets. We compare Alg 4 (trimmed kmeans) vs Alg 5 (robust kmeans) on 10 OpenML datasets with $k=10$ for varying degrees of robustness ($\tau$ in Alg 4, and the nonlinear activation inverse temperature $\gamma$ in Alg 5), identifying situations where they achieve better kmeans objective than Lloyd's. We report the average log kmeans objective (10 trials). Across all settings, Alg 4 outperforms Lloyd's (*italics* in table below) on 4/10 datasets, while the new Alg 5 (A5) does so for 6/10 datasets, demonstrating competitive behavior.
>
> |Dataset|Initial|Lloyd|A4 $\tau=0.95$|A4 $\tau=0.99$|A5 $\gamma=0.01$|A5 $\gamma=0.001$|
> |-|-|-|-|-|-|-|
> |GermanCredit|9.1393|8.6342|8.6345|*8.6339*|8.6364|*8.6333*|
> |twonorm|8.1141|7.6126|7.6130|7.6127|7.6129|7.6127|
> |pc3|6.2494|5.6776|5.7997|5.6823|5.6802|5.6777|
> |ada-agnostic|9.6838|9.1439|9.1448|9.1440|9.1882|*9.1428*|
> |ringnorm|7.7387|7.3020|7.3023|*7.3019*|7.3021|*7.3020*|
> |pc4|6.3693|5.8479|5.8558|5.8493|*5.8395*|*5.8476*|
> |pendigits|8.7020|8.0354|*8.0313*|*8.0265*|8.1473|*8.0279*|
> |sylvine|8.4709|7.9196|7.9268|7.9209|7.9316|7.9249|
> |optdigits|10.269|9.6075|9.6092|9.6079|9.7229|9.6075|
> |mfeat-karhunen|8.2904|7.6923|7.6937|*7.6922*|*7.6911*|7.6924|
>
> ---
> > ### DT5H-5 *Evaluate Alg 6 (randomized kmedoids)*
>
> Here we evaluate Alg 6, varying values of $\delta \in \lbrace 0, 0.01, 1, 100, 1e4, 1e8, 1e15 \rbrace$. We use 10 OpenML datasets with $k=10$, and report the average log kmedoids objective (10 trials). The best objectives (*italics* in the table below) are found by values of $\delta>0$, thereby lowering the selection probability from outside the current cluster, but not too large $\delta$ which enforces within cluster sampling. The results highlight that it is critical to balance (i) the ability to sample from outside the current cluster, with (ii) exploiting strong centroid candidates within the current cluster. This tradeoff can be tuned with $\delta$ in Alg 6, which is obtained via architectural modifications to the base kmeans transformer.
>
> |Dataset|Init|$\delta$:0|$\delta$:1e-2|$\delta$:1|$\delta$:1e2|$\delta$:1e4|$\delta$:1e8|$\delta$:1e15|
> |-|-|-|-|-|-|-|-|-|
> |GermanCredit|9.1393|9.1021|*9.0945*|9.0993|9.0950|9.1029|9.1027|9.1147|
> |twonorm|8.1141|7.9807|7.9807|*7.9741*|7.9910|7.9902|8.0070|8.0215|
> |pc3|6.2322|6.0533|*6.0201*|6.0489|6.0278|6.0264|6.0438|6.0217|
> |ada-agnostic|9.7136|9.6223|9.6057|*9.6023*|9.6128|9.6313|9.6317|9.6341|
> |ringnorm|7.7808|*7.6342*|7.6393|7.6428|7.6603|7.6643|7.6691|7.6691|
> |pc4|6.4307|6.2480|6.2309|6.2239|*6.2102*|6.2113|6.2312|6.2412|
> |pendigits|8.7588|8.5852|8.5812|8.5747|8.4872|8.4844|*8.4672*|8.4799|
> |sylvine|8.4547|8.3895|8.3857|8.3861|*8.3542*|8.3788|8.3804|8.3580|
> |optdigits|10.268|10.172|10.173|10.170|10.172|*10.137*|*10.137*|10.140|
> |mfeat-karhunen|8.2856|8.2412|8.2423|8.2392|8.2374|8.2312|*8.2295*|8.2424|
>
> ---
> > ### DT5H-6 *Bounding approximation in transformer with finite $\gamma$*
>
> Assuming $n$ points in $[0,1]^d$ and $\Delta>0$ as the smallest margin between squared Euclidean distances of points to their nearest and second nearest centers, the difference between the cluster center updates in equation (4) for a finite $\gamma$ and Lloyd's update is bounded by a $O(\exp(-\gamma \Delta) (n + k)\sqrt{d})$ term. This gap increases with $n$ and $k$, but decreases very quickly with increase in $\gamma$ and $\Delta$, where $\Delta$ usually increases with dimensionality $d$. This theoretical behavior is also demonstrated in Fig 5, Appendix A (specifically Fig 5E,F,G,H) where $\gamma = 10$ generally suffices for $d \geq 10$, and $\gamma = 100$ suffices across the board unless $d$ is very small.

---

### Decision · Program_Chairs · 2026-04-30

**Decision:**

Accept (spotlight)

**Comment:**

This paper shows that transformer circuits can exactly realize Lloyd’s algorithm for k-means clustering. The authors construct a precise encoder-decoder transformer that implements one Lloyd iteration per layer, give a second construction using standard dot-product attention plus an FFN, show that the learned architecture can outperform Lloyd’s on task distributions of clustering problems, and demonstrate that interpretable architectural modifications recover both existing and new clustering variants. Overall, the study's central contribution pertains to a concrete and rigorous bridge between transformer circuit mechanisms and exact algorithmic computation, and Overall, the authors examine a fundamental question about what transformers can represent and learn beyond statistical sequence modeling.

I recommend strong acceptance. The paper is unusually strong in both conceptual clarity and technical substance: it provides exact constructive results, clean proofs, numerical validation of the exact implementation, and learning experiments showing that the same architecture can be trained into an effective clustering procedure that generalizes across task sizes and improves over Lloyd’s on the target task family. Beyond this, the paper uses the construction as an algorithmic scaffold for deriving meaningful clustering variants, which substantially broadens its impact.

The main reviewer concerns were largely about scope and additional empirical breadth rather than flaws in the core contribution, and the rebuttal strengthened the paper further by adding real-data benchmark evidence on OpenML datasets, empirical results for the proposed robust and randomized variants, additional interpretability analyses, and a finite-(\gamma) approximation bound. Reviewers explicitly noted that their concerns were resolved and raised or maintained positive scores accordingly.

Overall, this is a rare paper that combines exact algorithmic expressivity results, interpretable circuit constructions, and meaningful learning behavior in a single coherent story. It should be of broad interest to researchers in transformers, in-context algorithm learning, and unsupervised learning, and I recommend strong acceptance.